# Immediate early splicing controls translation in activated T-cells and is mediated by hnRNPC2 phosphorylation

Mateusz Dróżdż[1], Luíza Zuvanov [iD][1], Gopika Sasikumar[2], Debojit Bose[1], Franziska Bruening[3], Maria S Robles[3], Marco Preußner [iD][1], Markus Wahl [iD][2] & Florian Heyd [iD][1✉]

## Abstract

The fast and transient induction of immediate early genes orchestrates the cellular response to various stimuli. These stimuli trigger phosphorylation cascades that promote immediate early gene transcription independent of de novo protein synthesis. Here we show that the same phosphorylation cascades also target the splicing machinery, inducing an analogous splicing switch that we call immediate early splicing (IES). We characterize hnRNPC2-controlled IES, which depends on the MEK-ERK pathway and the T cell-specific kinase PKCθ. This splicing switch mainly targets components of the translation machinery, such as mRNAs encoding ribosomal proteins and eIF5A. Inducing the eIF5A IES protein variant is by itself sufficient to reduce global translation, and consistently, we observe reduced de novo protein synthesis early after T cell activation. We suggest that immediate early splicing and the ensuing transient decrease in translation efficiency help to coordinate the extensive changes in gene expression during T cell activation. Together, these findings set a paradigm for fast and transient alternative splicing in the immediate cellular response to activation, and provide evidence for its functional relevance during T-cell stimulation.

**Keywords** Immediate Early Genes (IEG); Immediate Early Splicing (IES); T Cell Activation; hnRNPC; PKCθ
**Subject Categories** Chromatin, Transcription & Genomics; Immunology

## Introduction

Immediate early genes (IEGs) were first described in 1984, when *c-Fos* was shown to be rapidly induced by growth factor stimulation (Cochran et al, 1984; Greenberg and Ziff, 1984; Müller et al, 1984). Since then, IEGs have been defined as being rapidly and transiently induced upon cellular stimulation/activation, with the induction being independent of de novo protein synthesis. Instead, induction of IEGs relies on phosphorylation cascades that alter the activity of existing proteins, ultimately transcription factors that bind to specific motifs in IEG promoters. IEGs are themselves often transcription factors or involved in regulating signaling cascades and thus represent the first response to coordinate secondary events that establish the longer-term cellular effects, for example proliferation or differentiation (O'Donnell et al, 2012). In addition to their role in mitogenic stimulation, IEGs are involved in a variety of more specific processes, prominent examples include immune cell activation (Simon et al, 2006; Waldrip et al, 2021), synaptic plasticity and long-term memory (Minatohara et al, 2015) or setting of the circadian clock (Ginty et al, 1993). The MEK-ERK pathway plays a crucial role in the regulation of IEGs as ERK signaling controls chromatin structure at IEG promotors and directly phosphorylates and activates transcription factors, which together induces IEG transcription (O'Donnell et al, 2012). MEK-ERK signaling is induced within seconds after stimulation and the activity of this signaling pathway returns to baseline levels within hours (see Fig. EV1A for ERK1/2 kinetics in our model system), in part due to the induction of phosphatases that act in a negative feedback loop (Lake et al, 2016). This transient activity forms the basis for transient expression of IEGs, as their mRNAs are short lived (Bramham et al, 2008), and their presence therefore directly correlates with the kinetics of de novo transcription.

While components of the transcription machinery that are targets of MEK-ERK signaling have been studied in much detail, other levels of gene regulation that could be regulated in a concerted manner have been overlooked in previous research. In particular, splicing and alternative splicing are processes that are known to be controlled by phosphorylation of core components of the spliceosome (Agafonov et al, 2011) and also of auxiliary splicing regulatory proteins such as SR proteins (Aubol et al, 2016). Alternative splicing plays a fundamental role in increasing the genome's coding capacity (Sinitcyn et al, 2023) especially in mammals (Barbosa-Morais et al, 2012) and contributes to establish cell type and tissue-specific functionality (Marasco and Kornblihtt, 2023). In addition, alternative splicing has long been known to dynamically adapt the transcriptome to changing conditions,

[1]Institute of Chemistry and Biochemistry, Laboratory of RNA Biochemistry, Freie Universität Berlin, Takustr. 6, 14195 Berlin, Germany. [2]Institute of Chemistry and Biochemistry, Laboratory of Structural Biochemistry, Freie Universität Berlin, Takustr. 6, 14195 Berlin, Germany. [3]Institute of Medical Psychology and Biomedical Center, Faculty of Medicine, LMU, Munich, Germany. ✉E-mail: florian.heyd@fu-berlin.de

including altered (body) temperature (Haltenhof et al, 2020; Los et al, 2022; Preußner et al, 2017, 2023), circadian time (Preußner et al, 2014) or immune cell activation and differentiation (Liao and Garcia-Blanco, 2021). Furthermore, alternative splicing has always been considered to enable a fast response to diverse stimuli (Liu et al, 2022) as it can alter mRNA coding sequences and abundance independent of the transcription machinery. Dynamically adapting alternative splicing of nascent transcripts therefore allows a fast response also in very long genes whose transcription can take several hours (Carrillo Oesterreich et al, 2016; Sánchez-Escabias et al, 2022). Alternative splicing would thus be ideally suited to contribute to the immediate response upon cellular activation/stimulation. However, whether there is a transient change in alternative splicing, controlled by transient phosphorylation of components of the splicing machinery, has not been investigated.

T cell activation induces vast changes in gene expression that alter functionality such as proliferation, migration and the production and secretion of effector molecules. The well-defined changes in functionality and available cell culture models have led to frequent use of T cell activation as a model system to study how external signals are transmitted to the nucleus to alter gene expression and impact on diverse aspects of cellular functionality. T cell activation is also one of the best-studied systems to characterize mechanisms and functional consequences of signal-induced alternative splicing. Signaling cascades connecting the T cell receptor with altered activity of the splicing machinery, the regulation of individual exons and their functionality as well global changes in alternative splicing during activation and differentiation have been analyzed in different cell culture and primary systems in detail (Blake et al, 2022; Ip et al, 2007; Martinez and Lynch, 2013). The full development of effector cell function takes one to several days, which might be one reason, why studies so far have focused on characterizing alternative splicing at later timepoints post activation, in many cases after 24 h or 48 h (Blake et al, 2022; Ip et al, 2007; Martinez and Lynch, 2013). It therefore remains unknown, whether, like for IEGs, an early and transient splicing switch might occur, and if so, how this could contribute to T cell activation. T cell activation induces massive changes in gene expression, with hundreds if not thousands of genes being up- or downregulated in a concerted effort. The early hours after T cell activation are key to coordinate this process, which at later time points allows the acquisition of effector cell function. Earlier studies have shown fast induction of IEGs upon T cell activation (see Fig. EV1B for *c-Fos* kinetics in our model system) followed by increasing transcriptomic changes, which peak after 1 day or later (Rade et al, 2023). Proteomic studies have suggested that global de novo protein synthesis is reduced in the early hours after T cell activation by an as of yet unknown mechanism. In the first hours post activation proteomic changes are mainly changes in phosphorylation of preexisting proteins, thereby remodeling protein-protein interaction networks and functionality. A strong increase in de novo synthesized proteins is only observed 16 h post activation (Tan et al, 2017), which is close to the timepoint where transcriptomic changes reach their peak level (Rade et al, 2023). Reducing de novo protein synthesis in the early phase post activation, where still mainly the naïve transcriptome is present, likely accelerates turnover of the proteome from the naïve to the activated state, as translation of naïve mRNAs is reduced, and replenishment of decaying proteins is slowed down. T cells may therefore benefit from increasing de novo translation only once transcriptomic changes have been implemented.

In the present study, we establish the concept of immediate early splicing (IES). We show that a group of splicing events, mostly retention of introns, shows a fast and transient splicing switch upon T cell activation. This concerted splicing switch is independent of de novo translation, requires MEK-ERK-mediated phosphorylation of hnRNPC2 and is rendered T cell specific by the additional involvement of the kinase PKCθ, which is mainly expressed in T cells. hnRNPC2-controlled IES affects many components of the translation machinery, amongst others several transcripts encoding ribosomal proteins and translation factors including eukaryotic initiation factor eIF5A. Furthermore, we observe an hnRNPC2-dependent reduction in de novo translation in the hours immediately following T cell activation. We suggest that this is mediated by the IES switch, as inducing the IES protein variant in eIF5A is alone sufficient to globally reduce translation. Altogether, we present a new paradigm for fast and transient alternative splicing regulation and suggest a fundamental role in coordinating different layers of gene expression upon T cell activation.

## Results

### Immediate early splicing upon T cell activation is controlled by hnRNPC

The induction of immediate early genes (IEGs) upon diverse stimuli is mechanistically well understood and known to be dependent on phosphorylation cascades, amongst others the MEK-ERK pathway (Murphy et al, 2004; O'Donnell et al, 2012; Ojea Ramos et al, 2022). We hypothesized that components of the splicing machinery could be similarly targeted, to then induce immediate early splicing (IES) changes. To start investigating this hypothesis, we used PMA-stimulation of Jurkat T cells, which leads to a robust and reversible activation of ERK1/2 (Fig. EV1A) and a fast and strong transient induction of *c-Fos* (Fig. EV1B), as expected (Cochran et al, 1984; Greenberg and Ziff, 1984; Müller et al, 1984). We performed RNA-Seq of chromatin-associated RNA 0-, 30- and 150-min post stimulation and investigated differential alternative splicing using rMATS (Shen et al, 2014). We focused our analysis on splicing patterns expected for immediate early events, i.e., a change between 0 and 30 min, which then goes back towards baseline levels 150 min post stimulation (Fig. 1A, see "Methods" for details). We analyzed all types of alternative splicing (Dataset EV1), but intron retention stood out as most strongly affected (Figs. 1A,B and EV1C). A principal component analysis of significantly changed IR events between 0 and 30 min revealed clustering of the biological duplicates and more similarity of unstimulated cells with 150 min stimulation, whereas cells stimulated for 30 min were different from the other conditions (Fig. 1A). A more detailed analysis revealed that most of these introns are retained after 30 min of stimulation but return to basal levels after 150 min (Fig. 1B). Many IR events were found within pre-mRNAs encoding for components of the translation apparatus, which is also reflected in translation-related GO terms being highly enriched (Fig. 1C).

As the vast majority of ~6000 rMATS-quantified introns is not affected, we rule out a general reduction of splicing efficiency 30 min post stimulation and suggest that this splicing switch is specific for some selected introns. Interestingly, introns that are more efficiently spliced 30 min after stimulation are longer than

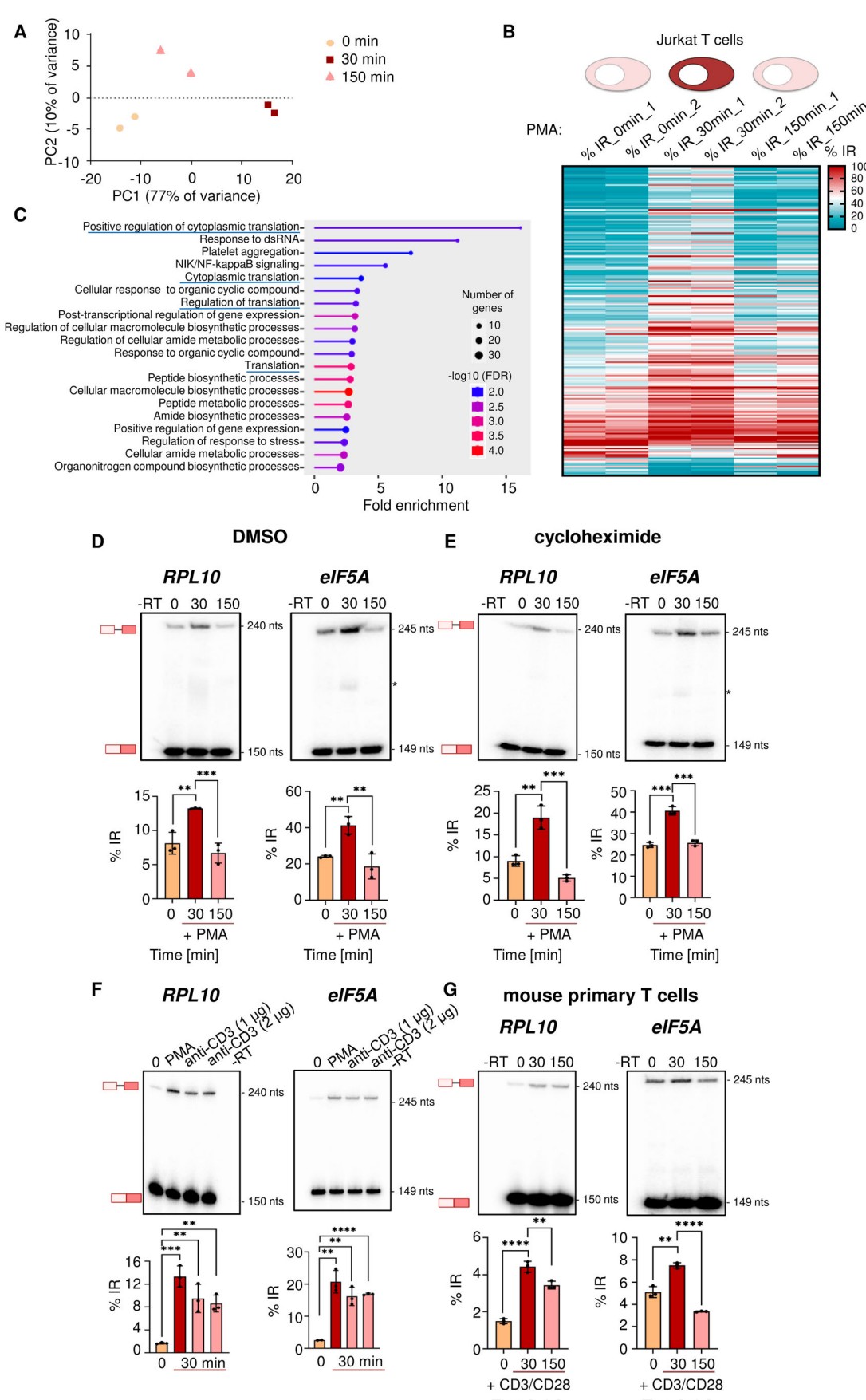

**Figure 1.  Characterization of immediate early splicing (IES) upon T cell activation.**

(A) Principle component analysis of the investigated duplicate samples of 0-, 30- and 150-min PMA-stimulated Jurkat T cells after sequencing of chromatin-associated RNA. Based on significantly changed IR events comparing 0 and 30 min. The percent variance is indicated. (B) Heat map of IR events as in (A). % IR (percent intron retention) values are shown for biological duplicates, each individual column corresponds to one individual sample. Sorted by Δ% IR between 0 and 30 min, each row corresponds to a different intron retention event. (C) The strongest GO term enrichments of genes containing retained introns after 30 min of PMA stimulation. GO terms containing the term "translation" are underlined in blue. (D) Validating reversible IR in chromatin-associated RNAs by radioactive, splicing-sensitive RT-PCR (top: left: *RPL10*; right: *eIF5A*). Representative gels show increased intron retention (IR) only after 30 min of PMA activation (-RT; without reverse transcriptase, * - degradation product). Splicing products are indicated on the left; size in nucleotides is indicated on the right. Bottom: corresponding quantifications (data are presented as % IR, mean ± SD, number of biological replicates: $n = 3$ (left: *RPL10*: 0 vs. 30 min, $P = 0.0054$, 30 vs. 150 min, $P = 0.0004$, right: *eIF5A*: 0 vs. 30 min, $P = 0.0037$, 30 vs. 150 min, $P = 0.0098$, Student's unpaired *t* test). (E) Reversible IR is maintained in the absence of translation (cycloheximide). RT-PCRs, gels and quantifications as in (D) (data are presented as % IR, mean ± SD, number of biological replicates: $n = 3$ (left: *RPL10*: 0 vs. 30 min, $P = 0.0043$, 30 vs. 150 min, $P = 0.001$, right: *eIF5A*: 0 vs. 30 min, $P = 0.0003$, 30 vs. 150 min, $P = 0.0002$, Student's unpaired *t* test). (F) Jurkat cells were stimulated for 30 min either by PMA or anti-CD3 Ab (with indicated amounts). Top: radioactive, splicing-sensitive RT-PCR confirms IES in *RPL10* (left) and *eIF5A* (right), -RT; without reverse transcriptase. Bottom: corresponding quantifications as in (D) (data are presented as % IR, mean ± SD, number of biological replicates: $n = 3$ (left: *RPL10*: 0 vs. 30 min (PMA), $P = 0.0004$, 0 vs. 30 min (anti-CD3 Ab—1 μg), $P = 0.0055$, 0 vs. 30 min (anti-CD3 Ab—2 μg), $P = 0.0013$, right: *eIF5A*: 0 vs. 30 min (PMA), $P = 0.0062$, 30 vs. 150 min (anti-CD3 Ab—1 μg), $P = 0.0077$, 0 vs. 30 min (anti-CD3 Ab—2 μg), $P < 0.0001$, Student's unpaired *t* test). (G) IES in mouse primary T cells stimulated with CD3/CD28. RT-PCRs, gels and quantifications as in (D) (data are presented as % IR, mean ± SD, number of biological replicates: $n = 3$ (left: *RPL10*: 0 vs. 30 min, $P < 0.0001$, 30 vs. 150 min, $P = 0.0086$, right: *eIF5A*: 0 vs. 30 min, $P = 0.0016$, 30 vs. 150 min, $P < 0.0001$, Student's unpaired *t* test). Source data are available online for this figure.

retained introns pointing to intron size as one factor that plays a role in determining splicing outcome. Splice site strength of affected introns was not different between the differentially regulated introns (Fig. EV1D), suggesting that other features are involved in controlling retention/splicing early after activation. In terms of potential functionality, we note that most of the 173 introns retained 30 min after stimulation (at least 15% more IR compared to the 0 timepoint with most of these introns displaying a change between 15% and 30%, Fig. EV1E) are located within the coding region of protein coding genes, with the large majority introducing a frame shift or directly encoding a stop codon (Fig. EV1F). In ten of these cases, the stop codon occurs in the last intron of the ORF, potentially producing a stable mRNA encoding an alternative C-terminus (Preußner et al, 2020). However, there are also 4 cases of in-frame introns that encode alternative protein isoforms (Fig. EV1F and see below). We therefore conclude that NMD and a reduction in the expression of the respective genes is the dominant effect of the retention of these introns, with some notable exception where retained introns have coding potential.

We used radioactive, splicing-sensitive RT-PCR to validate bioinformatic predictions and were able to validate various targets (Fig. 1D; Appendix Fig. S1A), confirming the validity of our analysis pipeline (see methods). We chose two IR events in *RPL10* and *eIF5A*, both components of the translation machinery, as exemplary targets throughout the study. Both show an around two-fold induction of IR 30 min post stimulation, which goes completely back to baseline levels after 150 min (Fig. 1D), whereas their mRNA abundance in total RNA was not altered (Appendix Fig. S1B). We observed the same splicing switch if de novo protein synthesis was inhibited by the presence of cycloheximide (Fig. 1E; Appendix Fig. S1C). These experiments show a fast and transient splicing switch that happens upon PMA stimulation of T cells independent of de novo protein synthesis thereby establishing the concept of immediate early splicing. Of note, a similar pattern of *RPL10* and *eIF5A* IES (Fig. 1F) was observed upon stimulation of Jurkat cells with anti-CD3, showing that this response is not confined to PMA stimulation. To rule out a cell line specific effect, we performed similar experiments in CD3/CD28-stimulated mouse primary T cells. We again observed IR after 30 min of stimulation (Fig. 1G), which went back towards baseline levels after 150 min,

validating IES in primary cells and showing evolutionary conservation of short introns and their regulation during T cell activation.

We then went on to identify components of the splicing machinery, whose transient phosphorylation could mediate IES. To this end, we performed label-free quantitative phosphoproteomics in Jurkat cells 0, 15, 30 and 90 min after PMA stimulation and identified reversible changes in phosphorylation after 15 and 30 min that went back towards baseline levels after 90 min. We focused on splicing regulatory proteins and indeed identified many candidates with a transient change in phosphorylation (Dataset EV2), among them the well-known splicing regulator hnRNPC. In our phosphoproteomics data, hnRNPC shows a peak in phosphorylation at Ser115 already 15 min post stimulation, which then slightly decreases towards baseline levels at the later time points (Fig. 2A). hnRNPC controls alternative splicing in diverse settings, amongst others at later time points upon T cell activation (Schultz et al, 2017; Zarnack et al, 2013). hnRNPC is expressed in two splice variants, hnRNPC1 and hnRNPC2, as the result of alternative usage of a 5' splice site in exon 4 (Burd et al, 1989; McAfee et al, 1996). hnRNPC2 contains an additional 13 amino acids including the transiently phosphorylated Ser115 (Fig. 2A,B). Using western blotting we confirmed the expression of both hnRNPC isoforms in unstimulated Jurkat cells (Fig. 2C), with hnRNPC1 being the main variant. This corresponds to the isoform expression ratio observed at the mRNA level (Fig. EV2A). Fifteen minutes after PMA stimulation, we observed an almost complete shift in the size of the hnRNPC2 isoform, which reverted back towards the apparent size of the unstimulated sample after 90 and 240 min. In contrast, the size of hnRNPC1 was not affected (Fig. 2C). We interpret this apparent shift in size in hnRNPC2 as transient phosphorylation of the (almost) entire hnRNPC2 population, which is consistent with our phosphoproteomics data and is supported by the finding that phosphatase treatment reverses the apparent size shift (Fig. EV2B). This phosphorylation pattern is quite remarkable, as in many cases phosphorylation affects only a fraction of the protein population (Nishi et al, 2014; Zhang et al, 2023). Furthermore, a similar pattern of hnRNPC2 phosphorylation was observed upon stimulation of Jurkat cells with anti-CD3, showing that this response is not confined to PMA stimulation but

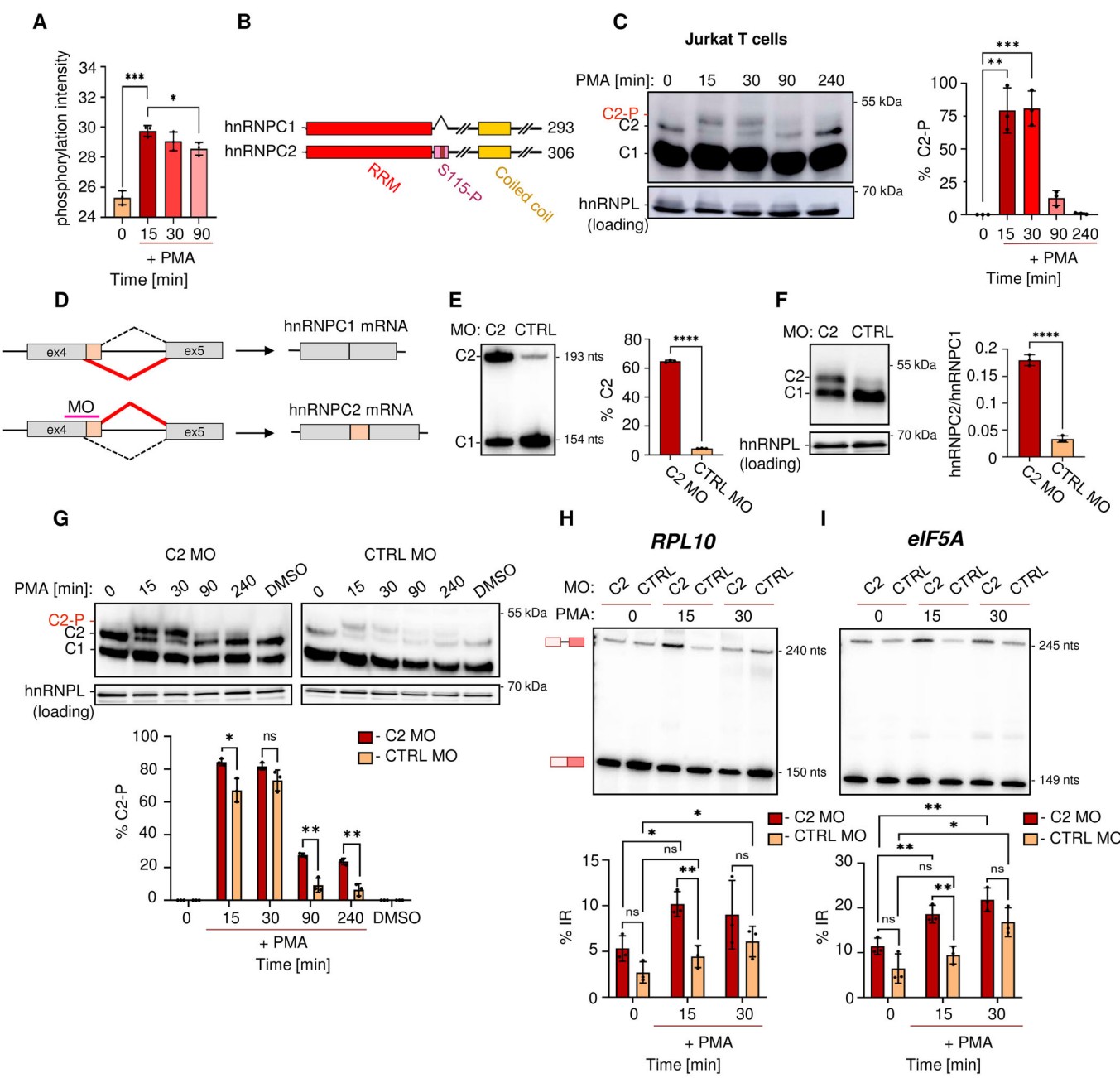

## hnRNPC2 controls immediate early splicing

To directly connect hnRNPC2 with IES, we designed an anti-sense Morpholino (MO) that blocks specifically the 5' splice site leading to hnRNPC1 and thus increases the production of hnRNPC2 (Fig. 2D). Transfecting cells with this MO indeed strongly increased hnRNPC2 at RNA and protein levels (Fig. 2E,F). This experiment also validated our western blot assignment of the hnRNPC1 and C2 isoforms, as the MO decreased the C1 band while increasing the C2 isoform, thus confirming the identity of the respective bands. Increasing hnRNPC2

is also triggered by stimuli mimicking the endogenous situation (Fig. EV2C).

also strongly increased the amount of phosphorylated hnRNPC2 upon PMA stimulation (Fig. 2G). We then investigated IES upon PMA stimulation in cells with MO-induced increased hnRNPC2 expression. We observed slightly increased basal IR in *RPL10* and *eIF5a* and increased IES 15- and 30-min post stimulation when compared to control MO transfected cells (Fig. 2H,I). These data clearly implicate hnRNPC2 in IES regulation, as increasing hnRNPC2 led to increased and faster accumulation of IR variants.

## hnRNPC is required for efficient RPL10 and eIF5A splicing

To further connect hnRNPC with IES we knocked down hnRNPC (using a siRNA that targets both isoforms) in Jurkat cells and

**Figure 2.  hnRNPC2 is phosphorylated upon T cell activation and controls IES of *RPL10* and *eIF5A*.**

(A) Transient phosphorylation pattern of hnRNPC2 in PMA-stimulated Jurkat T cells, assessed by mass spectrometry (label-free quantification of TiO2-enriched phospho-peptides performed in four replicates per condition, data are presented as log2 phosphorylation intensity, mean ± SD, 0 vs. 15 min, $P = 0.0002$, 15 vs. 90 min, $P = 0.042$, Student's unpaired $t$ test). (B) Schematic representation of human hnRNPC isoforms hnRNPC1 and hnRNPC2 (red: RRM—amino-terminal RNA recognition motif, pink—13 amino acids exclusively present in hnRNPC2, containing Ser115, yellow: coiled coil). (C) Reversible phosphorylation of hnRNPC2 was confirmed by western blot (left) and quantified (right, data are presented as % phosphorylated hnRNPC2 of total hnRNPC2, mean ± SD, number of biological replicates: $n = 3$, 0 vs. 15 min, $P = 0.013$, 0 vs. 30 min, $P = 0.0005$, Student's unpaired $t$ test). hnRNPL acting as loading control. (D) Schematic view of the morpholino (MO)-induced manipulation of hnRNPC splicing. The MO (pink bar) blocks the proximal 5′-splice site leading to the usage of the distal 5′splice thereby increasing the hnRNPC2 isoform. (E) Jurkat cells were electroporated with the hnRNPC2-inducing MO (hnRNPC2 MO) or control MO (CTRL MO). Cells were harvested and total RNA was extracted. Left: the efficiency of hnRNPC2 MO was analyzed by radioactive, splicing-sensitive PCR and quantified (right, data are presented as % C2, mean ± SD, number of biological replicates: $n = 3$, $P < 0.0001$, Student's unpaired $t$ test). (F) Jurkat T cells were electroporated as in (E) and total protein was extracted. Left: the efficiency of hnRNPC2 MO was analyzed by western blot. hnRNPL acting as a loading control. Right: corresponding quantification; data are presented as hnRNPC2/hnRNPC1 ratio, mean ± SD, number of biological replicates: $n = 3$, $P < 0.0001$, Student's unpaired $t$ test). (G) Jurkat cells were electroporated as in (E). After 48 h of indicated MOs transfections, cells were stimulated by PMA for the indicated times or DMSO-treated as control. Total protein was extracted. Top: Western blot shows strongly increased amount of hnRNPC2 and phosphorylated C2 protein (C2-P) after stimulation in the hnRNPC2 MO-electroporated cells. hnRNPL serves as loading control. Bottom: corresponding quantification (data are presented as % hnRNPC2 phosphorylation of total hnRNPC2, mean ± SD, number of biological replicates: $n = 3$, C2 MO vs. CTRL MO (15 min), $P = 0.0163$, C2 MO vs. CTRL MO (30 min), $P = 0.0887$, C2 MO vs. CTRL MO (90 min), $P = 0.002$, C2 MO vs. CTRL MO (240 min), $P = 0.002$, Student's unpaired $t$ test). (H, I) Jurkat cells were treated as in (E) and chromatin-associated RNA was extracted. Top: radioactive, splicing-sensitive RT-PCR shows increased IR in *RPL10* (H) and *eIF5A* (I) in hnRNPC2 MO-electroporated cells. All experiments are representative of three biological replicates ($n = 3$). Bottom: corresponding quantifications (data are presented as % IR, mean ± SD, number of biological replicates: $n = 3$, *RPL10* (H): C2 MO vs. CTRL MO (0 min), $P = 0.0678$, C2 MO vs. CTRL MO (15 min), $P = 0.0056$, C2 MO vs. CTRL MO (30 min), $P = 0.0867$, 0 vs. 15 min (C2 MO), $P = 0.0126$, 0 vs. 15 min (CTRL MO), $P = 0.1505$, 0 vs. 30 min (CTRL MO), $P = 0.0453$, *eIF5A* (I): C2 MO vs. CTRL MO (0 min), $P = 0.0696$, C2 MO vs. CTRL MO (15 min), $P = 0.0076$, C2 MO vs. CTRL MO (30 min), $P = 0.0503$, 0 vs. 15 min (C2 MO), $P = 0.0087$, 0 vs. 30 (C2 MO), $P = 0.0056$, 0 vs. 15 min (CTRL MO), $P = 0.0786$, 0 vs. 30 min (CTRL MO), $P = 0.0118$, Student's unpaired $t$ test). Source data are available online for this figure.

investigated IR in *RPL10* and *eIF5A*. Knockdown efficiency of 30-40% was confirmed by western blot (Fig. EV2D). Despite the mild reduction in hnRNPC levels, both IES events were completely abrogated (Fig. 3A,B), as was another IES event in *TRAF4* (Fig. EV2E), demonstrating that these splicing switches are hnRNPC-dependent. For these analyses total RNA was used, which showed a similar fold change as nascent RNA but reduced absolute percent IR, as IES in this setup is measured in the context of pre-existing mRNA. For all three investigated target introns, knockdown of U2AF1 or hnRNPK did not interfere with IES, indicating specificity for hnRNPC. U2AF1 and hnRNPK were chosen as specificity control as U2AF1 exhibits transient phosphorylation upon T cell activation and hnRNPK functions as another polyC binding protein (Figs. 3A,B and EV2E). Interestingly, hnRNPC knockdown led to increased basal intron retention in *RPL10*, *eIF5A* and *TRAF4*, suggesting that hnRNPC is required for efficient splicing of these introns. This conclusion was confirmed in HEK293 cells, where knockdown of hnRNPC, validated by RT-qPCR (Fig. EV3F), also increased retention of these introns (Fig. 3C,D). We then aimed to address a direct regulation of these IR events by hnRNPC2 phosphorylation. To this end, we tested direct binding of recombinant, purified wt and phosphomimetic (S115D) hnRNPC2 (Fig. EV2G) to a part of the *RPL10* or *eIF5A* intron that includes the polypyrimidine tract and the 3′ splice site, as hnRNPC is known to bind pyrimidine stretches (Görlach et al, 1994). The RNA-hnRNPC2 interactions were tested by EMSA, which revealed the binding of the wt protein to the *RPL10* and *eIF5A* introns. In contrast, binding of the phosphomimetic mutant was strongly reduced (Figs. 3E,F and EV2H for similar results using the *TRAF4* intron), while no difference in binding to a poly-U RNA was observed (Fig. 3G, note reduced hnRNPC2 concentrations in the poly-U EMSA, as binding affinity is higher for this RNA). This finding leads to a model in which hnRNPC1/2 promotes splicing of these target introns by direct binding. Upon T cell activation, hnRNPC2 becomes phosphorylated on Ser115, which, according to the phosphomimetic mutant, reduces binding to the target introns,

thereby transiently increasing intron retention (Fig. 3H). This model is also consistent with the MO experiment (Fig. 2), as increasing the hnRNPC2:C1 ratio shifts the balance towards the isoform that is phosphorylated and shows reduced RNA binding upon activation, thereby increasing intron retention. Interestingly, altering the ratio between hnRNPC1 and hnRNPC2 using splice-site blocking MOs in HEK293 or HeLa cells had negligible effects on alternative splicing or gene expression as determined by RNA-Seq (Fig. EV2I–K). Thus, hnRNPC1 and hnRNPC2 isoforms appear to fulfill similar and redundant functions, and differences become only apparent upon hnRNPC2-specific phosphorylation, as this reduces binding to selected RNAs and controls alternative splicing.

## MEK-ERK signaling is necessary but not sufficient for hnRNPC2 phosphorylation and IES

To characterize the signaling cascade that is required for IES, we stimulated Jurkat cells with PMA in the presence of several well-established kinase inhibitors. We then prepared total RNA and assayed for *RPL10* and *eIF5A* IES. In control cells, we again observed increased IES with the same fold change as in nascent RNA (Fig. 4A,B). For both, *RPL10* and *eIF5A* IES was completely blocked in the presence of a MEK inhibitor (Fig. 4A,B), whereas inhibiting AKT, JNK, p38 or JAK did not alter IES, which was also the case for an additional IES event in *TRAF4* (Appendix Fig. S2A). We also investigated modification of hnRNPC2 in the presence of the MEK inhibitor. The appearance of the slower migrating hnRNPC2 isoform was completely abrogated in the presence of the MEK inhibitor, demonstrating that the appearance of this band is, as IES, dependent on the MEK-ERK signaling pathway (Fig. 4C). These data, together with the data in Fig. 2, confirm that the slower migrating band is a phosphorylated variant of hnRNPC2. Combined with our mass spectrometry data (Fig. 2A) we suggest that the apparent shift in molecular weight of hnRNPC2 is due to the effect of ERK-mediated phosphorylation of Ser115, which is in

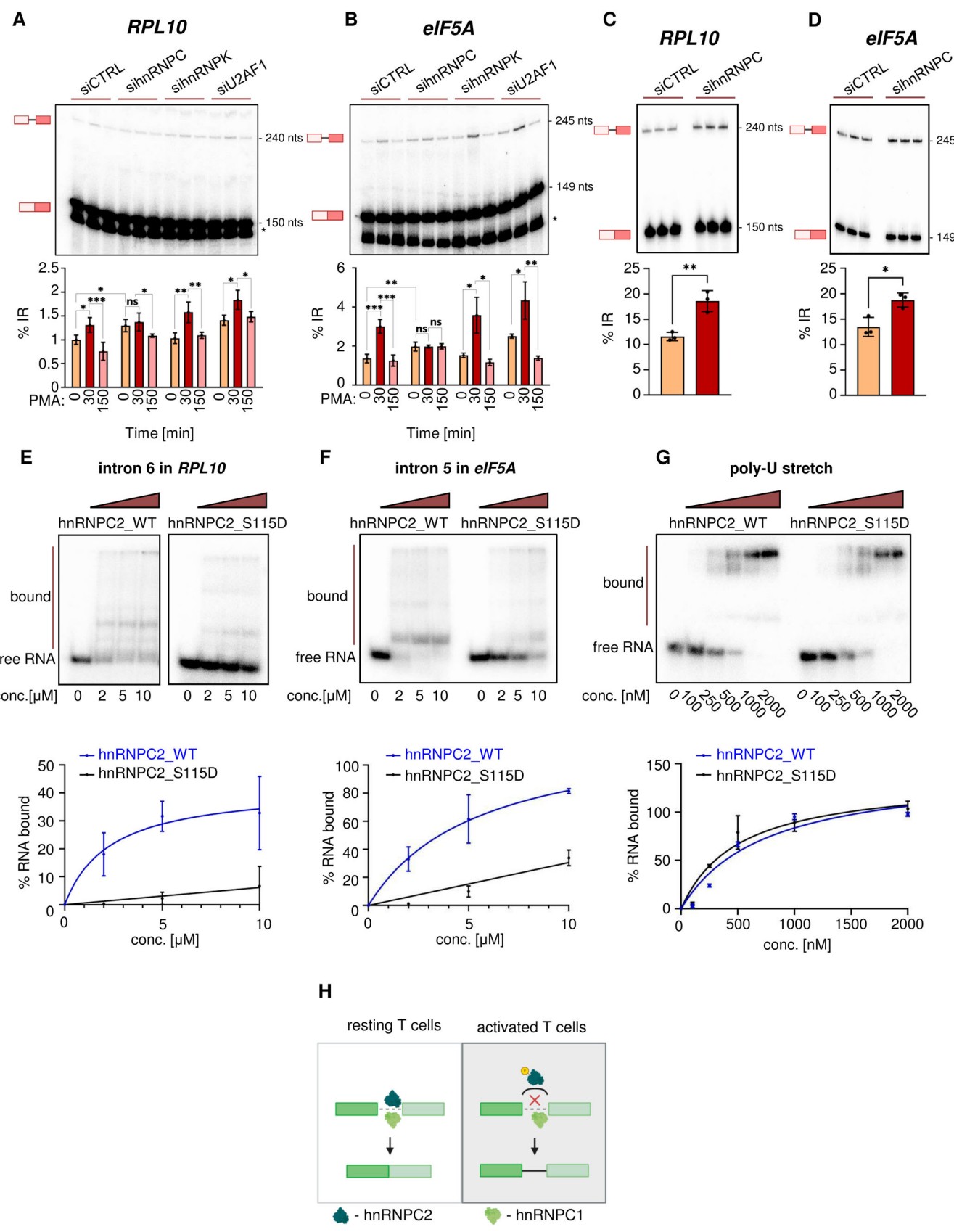

**Figure 3. hnRNPC is required for efficient *RPL10* and *eIF5A* splicing and a phosphomimetic S115D hnRNPC2 mutant shows reduced binding to intronic regions of IES targets.**

(A, B) IES depends on hnRNPC. Jurkat cells were transfected with siRNA against hnRNPC, hnRNPK or U2AF1 and 48 h post-transfection PMA-stimulated for the indicated times. *RPL10* (A) and *eIF5A* (B) IR were analyzed by radioactive, splicing-sensitive RT-PCR in total RNA. * - degradation product. Bottom: corresponding quantifications, data are presented as % IR, mean ± SD, number of biological replicates: $n = 3$, *RPL10* (A): 0 vs. 30 min (siCTRL), $P = 0.0114$, siCTRL vs. sihnRNPC (0 min), $P = 0.0181$, 30 vs. 150 min (siCTRL), $P = 0.0001$, 0 vs. 30 min (sihnRNPC), $P = 0.4986$, 30 vs. 150 min (sihnRNPC), $P = 0.0125$, 0 vs. 30 min (sihnRNPK), $P = 0.0012$, 30 vs. 150 min (sihnRNPK), $P = 0.0055$, 0 vs. 30 min (U2AF1), $P = 0.0247$, 30 vs. 150 min (U2AF1), $P = 0.0303$, *eIF5A* (B): 0 vs. 30 min (siCTRL), $P = 0.0006$, siCTRL vs. sihnRNPC (0 min), $P = 0.008$, 30 vs. 150 min (siCTRL), $P = 0.0008$, 0 vs. 30 min (sihnRNPC), $P = 0.4462$, 30 vs. 150 min (sihnRNPC), $P = 0.8858$, 0 vs. 30 min (sihnRNPK), $P = 0.0186$, 30 vs. 150 min (sihnRNPK), $P = 0.0109$, 0 vs. 30 min (U2AF1), $P = 0.0297$, 30 vs. 150 min (U2AF1), $P = 0.0059$, Student's unpaired $t$ test). (C, D) hnRNPC promotes splicing of *RPL10* (C) and *eIF5A* (D) introns in HEK293 cells. HEK293 cells were transfected with siRNA against hnRNPC and control. After 48 h, cells were harvested, and chromatin-associated RNA was analyzed by radioactive, splicing-sensitive RT-PCR. Bottom: corresponding quantifications (data are presented as % IR, mean ± SD, number of biological replicates: $n = 3$, *RPL10* (C): $P = 0.0055$, *eIF5A* (D): $P = 0.0175$, Student's unpaired $t$ test). (E, F) EMSAs were performed using 10 pmol of 30 nt RNA spanning the polypyrimidine tract and the 3′ splice-site of intron 6 in *RPL10* (E) and intron 5 in *eIF5A* (F) (see "Methods"). Increasing amounts of either hnRNPC2_WT or S115D were complexed with radioactively labeled RNA. Top: representative native gels. Data are representative of at least three independent experiments. Bottom: quantification of EMSAs performed in triplicates (mean ± SD, number of biological replicates: $n = 3$). (G) EMSA and corresponding quantification as in (E, F) using 10 pmol of radioactively labeled poly-U (45nt) RNA and lower concentrations of hnRNPC2_WT and S115D proteins (from 0 nM to 2000 nM, representative gel, mean ± SD, $n = 3$). (H) Schematic model based of our findings. Upon T cell activation, hnRNPC2 gets phosphorylated, leading to decreased binding to intronic sequences in *RPL10*, *eIF5A* and *TRAF4*, and, in consequence, reduced splicing efficiency. If the hnRNPC1:C2 balance is shifted towards the C2 isoform, e.g., in the MO experiment in Fig. 2, a higher proportion of total hnRNPC is phosphorylated and shows reduced binding, leading to increased intron retention. Created using Biorender.com. Source data are available online for this figure.

line with the sequence context around Ser115 (112-PSPSPLL-118) and ERK being a proline-directed kinase (Roskoski, 2012). As MEK-ERK signaling is a ubiquitous pathway that has been implicated in splicing regulation before (Weg-Remers, 2001) and hnRNPC2 is broadly expressed (e.g., Fig. EV2A), we assumed that hnRNPC2 phosphorylation and IES upon PMA stimulation would be a general response. However, treating HEK293 or HeLa cells with PMA did neither induce significant IES of *RPL10* or *eIF5A* nor did we observe substantial amounts of phosphorylated hnRNPC2 (Fig. 4D–F). In contrast, we did observe substantial IES in Jurkat cells treated in parallel with HEK293 and HeLa cells, and in another T cell line, mouse EL4 cells (Fig. 4E,F). These data show that hnRNPC2-controlled IES splicing is cell type-specific and happens in two T cell lines of mouse and human origin and in primary mouse T cells, whereas it is not observed in two other human cell lines. Based on these data we suggest that hnRNPC2-controlled IES is T cell specific, which is further supported by the finding that IES is not observed in a mouse macrophage cell line (RAW264.7) stimulated with LPS (Appendix Fig. S2B,C) or neuronal (N2a) cells upon KCl depolarization (Appendix Fig. S2D,E).

## PKCθ is necessary for T cell specific hnRNPC2 phosphorylation and IES

To address the mechanistic basis for T cell specificity, we considered additional kinases activated by PMA as potential mediators. Such kinases could for example induce a priming phosphorylation on hnRNPC2 that then allows ERK-mediated phosphorylation of Ser115 (also see discussion). Previous work has revealed that the PKC isoform PKCθ is mainly expressed in T cells, where it plays a role in controlling signaling upon T cell receptor engagement and T cell differentiation (Brezar et al, 2015; Nicolle et al, 2021). RT-qPCR analysis showed expression of PKCθ in human Jurkat and mouse EL4 cells, whereas it was almost undetectable in HEK293, HeLa and N2a cells (Appendix Fig. S3A,B), making PKCθ a promising candidate to mediate T cell specific IES. To start addressing this hypothesis we used pharmacological inhibition of PKCθ. Pretreating Jurkat cells with

a PKCθ inhibitor before PMA stimulation almost completely abolished IES of *RPL10* and *eIF5A*, indicating that PKCθ activity is required for this process (Fig. 5A,B). Furthermore, phosphorylation of hnRNPC2 was reduced in the presence of the PKCθ inhibitor, indicating a direct impact of PKCθ on this phosphorylation event (Fig. 5C). Importantly, inhibiting PKCθ had no impact on ERK activation, as evidenced by PMA-induced ERK phosphorylation (Appendix Fig. S3C). This finding is consistent with a PKCθ-mediated priming phosphorylation, as we observe reduced hnRNPC2 phosphorylation despite full ERK activity. As inhibiting PKCθ abolished IES almost completely, whereas residual hnRNPC2 phosphorylation was still detected, PKCθ might act by additional mechanisms to control hnRNPC2 activity or other events controlling IES. To further corroborate these findings, we used knockdown of PKCθ as another means to reduce its activity. Although the knockdown reduced PKCθ to only 50% (Appendix Fig. S3D), this was sufficient to substantially and significantly reduce IES of *RPL10* and *eIF5A* and hnRNPC2 phosphorylation in PKCθ siRNA-treated cells when compared to control siRNA (Appendix Fig. S3E–G), which is consistent with the results obtained by pharmacological PKCθ inhibition. Together, these data show that PKCθ activity is necessary for PMA-induced hnRNPC2 phosphorylation and IES of *RPL10* and *eIF5A*, and very likely other hnRNPC-dependent IES targets.

## Expression of PKCθ in HEK293 or HeLa cells is sufficient to induce IES in an hnRNPC2-dependent manner

Having established that PKCθ is necessary for IES upon PMA stimulation in Jurkat cells, we wished to test whether introducing PKCθ in other cell lines is sufficient to induce IES after PMA treatment. To this end, we used transient overexpression of PKCθ in HEK293 or HeLa cells and compared these cells with cells transfected with empty vector (Fig. EV3A). Consistent with the hypothesis that PKCθ is required for IES in *RPL10* and *eIF5A*, HEK293 cells overexpressing PKCθ clearly show IES similar to Jurkat cells (Fig. 5D,E). To show that this effect is mediated through hnRNPC2, we used CRISPR/Cas-9 and removed the

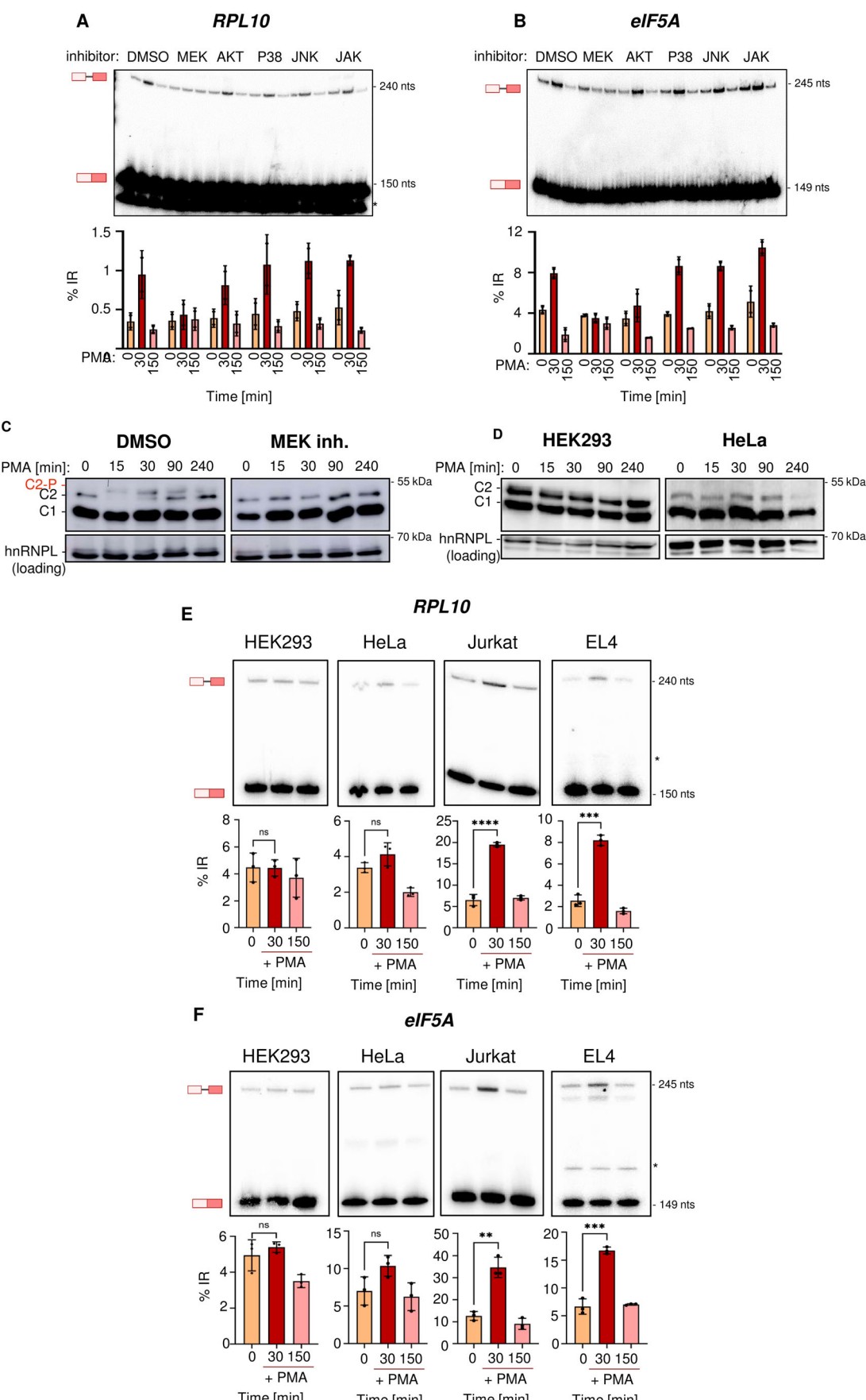

**Figure 4.  IES is triggered by the RAF/MEK/ERK signaling pathway.**

(**A, B**) MEK inhibition prevents IES. Jurkat cells were treated for 30 min with the indicated small molecule inhibitors and then stimulated with PMA for the indicated time points. Cells were harvested, and total RNA was extracted. Top: representative gel from radioactive, splicing-sensitive RT-PCR for *RPL10* (**A**) and *eIF5A* (**B**) (*-degradation product). Bottom: corresponding quantification (data are presented as mean % IR; individual data points are shown, the range in values is indicated by bars, number of biological replicates: n = 2). (**C**) MEK inhibition prevents transient phosphorylation of hnRNPC2. Cells were treated for 30 min either with DMSO (left) or MEK inhibitor (MEK inh.; right) and then stimulated by PMA for the indicated times. Cells were harvested, and total protein was investigated for hnRNPC2 phosphorylation (number of biological replicates: n = 3). hnRNPL serves as loading control. (**D**) Absence of reversible hnRNPC phosphorylation in non-immune cells. HEK293 (left) and HeLa (right) cells were stimulated by PMA for the indicated time points and protein lysates were investigated for hnRNPC2 phosphorylation (number of biological replicates: n = 3). hnRNPL serves as loading control. (**E, F**) Absence of IES in non-immune cells. HEK293, HeLa, Jurkat and EL4 cells were stimulated in parallel by PMA for the indicated time points and chromatin-associated RNA was investigated by splicing-sensitive RT-PCR for *RPL10* (**E**) and *eIF5A* (**F**). Bottom: corresponding quantifications (data are presented as % IR, mean ± SD, number of biological replicates: n = 3, *RPL10* (**E**): 0 vs. 30 min (HEK293): P = 0.964, HeLa: P = 0.1399, Jurkat: P < 0.0001, EL4 P = 0.0002, *eIF5A* (**F**): 0 vs. 30 min (HEK293): P = 0.0738, HeLa: P = 0.1267, Jurkat: P < 0.0001, EL4: P = 0.0003, Student's unpaired t test). Source data are available online for this figure.

alternative 5' splice site (Fig. EV3B) that leads to the generation of hnRNPC2, creating HEK293 and HeLa cells that only express the hnRNPC1 isoform. The exclusive expression of hnRNPC1 was confirmed on protein and RNA level (Fig. EV3C,D). Overexpression of PKCθ in these cells showed a complete lack of IES after PKCθ overexpression, confirming that hnRNPC2 is required for IES and that PKCθ acts through hnRNPC2 to induce IES upon PMA stimulation (Fig. 5D,E). We obtained very similar results in HeLa cells overexpressing PKCθ (Fig. 5F,G), validating this conclusion in an additional cell line.

To further validate that phosphorylation of Ser115 hnRNPC2 is required to induce IES, we generated a phosphomimetic hnRNPC2 S115→D mutant (Fig. EV3E) and overexpressed it, alongside wt hnRNPC2, in HEK293 cells. Expressing hnRNPC2 S115D alone was sufficient to increase IR in *RPL10* and *eIF5A* (Fig. EV3F,G) pointing to a central role of this phosphorylation event in controlling IES and suggesting that the phosphorylated/phospho-mimetic version acts in a dominant negative manner, as the effect on IR is seen with the endogenous wt protein still present. Furthermore, we mutated potential phosphorylation sites in and surrounding the hnRNPC2-specific part of the protein, including Ser115 (Fig. EV3E), to alanine, to prevent phosphorylation. We coexpressed PKCθ along with wt or mutant hnRNPC2 S/Y→A in HEK293 cells and assayed for IES upon PMA stimulation. Notably, we observed significantly stronger IES when expressing wt hnRNPC2 as compared to the non-phosphorylatable version (Fig. EV3H,I). We obtained the same result in HeLa cells, further validating the hypothesis that phosphorylation of hnRNPC2 is required to mediate IES (Fig. EV3J,K). Together, these data provide evidence that PKCθ mediates the cell type specificity of IES upon PMA stimulation and that PKCθ acts through hnRNPC2, likely through enabling phosphorylation of Ser115.

## hnRNPC and IES control de novo protein synthesis early after T cell activation

To start addressing the functionality of IES, we considered, based on GO term enrichment, a potential role in globally controlling translation upon T cell activation (Fig. 1C). We first assessed de novo translation at different time points after stimulation of Jurkat cells using incorporation of puromycin followed by detection with an α-puromycin antibody (a modified WB-SuNSET assay (Schmidt et al, 2009) or [35]S-methionine incorporation followed by autoradiography. Both methods showed reduced de novo protein

synthesis 2 and 4 h after stimulation, which returned to (or was slightly above) base line levels after 8 h and an increase was only observed at 16 and 24 h post stimulation (siCTRL in Figs. 6A and EV4A,B). This is consistent with earlier observations that proteomic changes directly following T cell activation are mainly mediated by altering the phosphorylation status of existing proteins, whereas de novo protein synthesis takes over at later time points (16 h) post-stimulation (Tan et al, 2017, also see "Discussion"). This drop in de novo protein synthesis correlates very well with the timing of IES. To address this connection experimentally, we first knock down hnRNPC, which we had shown to prevent IES (Figs. 3A,B and EV2E) and investigated the impact on de novo protein synthesis. Notably, both assays, WB-SuNSET and incorporation of [35]S-methionine showed that the immediate drop in de novo protein synthesis was completely abrogated in conditions with reduced hnRNPC expression (sihnRNPC in Fig. 6A, Fig. EV4A,B). To show that it is the hnRNPC2 isoform that impacts on de novo protein synthesis, we used our MO approach to increase the expression of hnRNPC2, which we had shown to facilitate the accumulation of IES variants (Fig. 2H,I). Under these conditions we observed a stronger decrease of de novo protein synthesis than in control conditions (Figs. 6B and EV4C), providing direct evidence for the involvement of the hnRNP2 isoform in controlling translation upon T cell stimulation. While these data strongly suggest a connection between hnRNPC2-controlled IES and the regulation of de novo protein synthesis after T cell activation, the mechanistic connection remains unclear. We speculated that individual protein coding IES IR isoforms could directly control translation. We chose the eIF5A IES variant as an intron that is retained, does not contain a stop codon, and thus encodes for a protein containing additional 32 amino acids (Fig. 6C, also see Fig. EV1F). When analyzing cytoplasmic RNA, we clearly detected the EIF5A IES variant, showing export from the nucleus (Fig. EV5A) and indicating translation. Consistently, both wt and IES variants of eIF5A were expressed when corresponding cDNA was inserted in an expression vector and transfected into HEK293 cells (Fig. EV5B). We then used WB-SuNSET to address the impact on de novo protein synthesis. While overexpression of the eIF5A wt had no effect compared to the empty vector control, overexpression of the IES variant significantly decreased de novo protein synthesis (Figs. 6D and EV5C). This result shows that already a single IES variant, if overexpressed, can have a profound impact on global de novo translation. To confirm these data at endogenous expression levels, we used a splice-site blocking MO to induce eIF5A intron

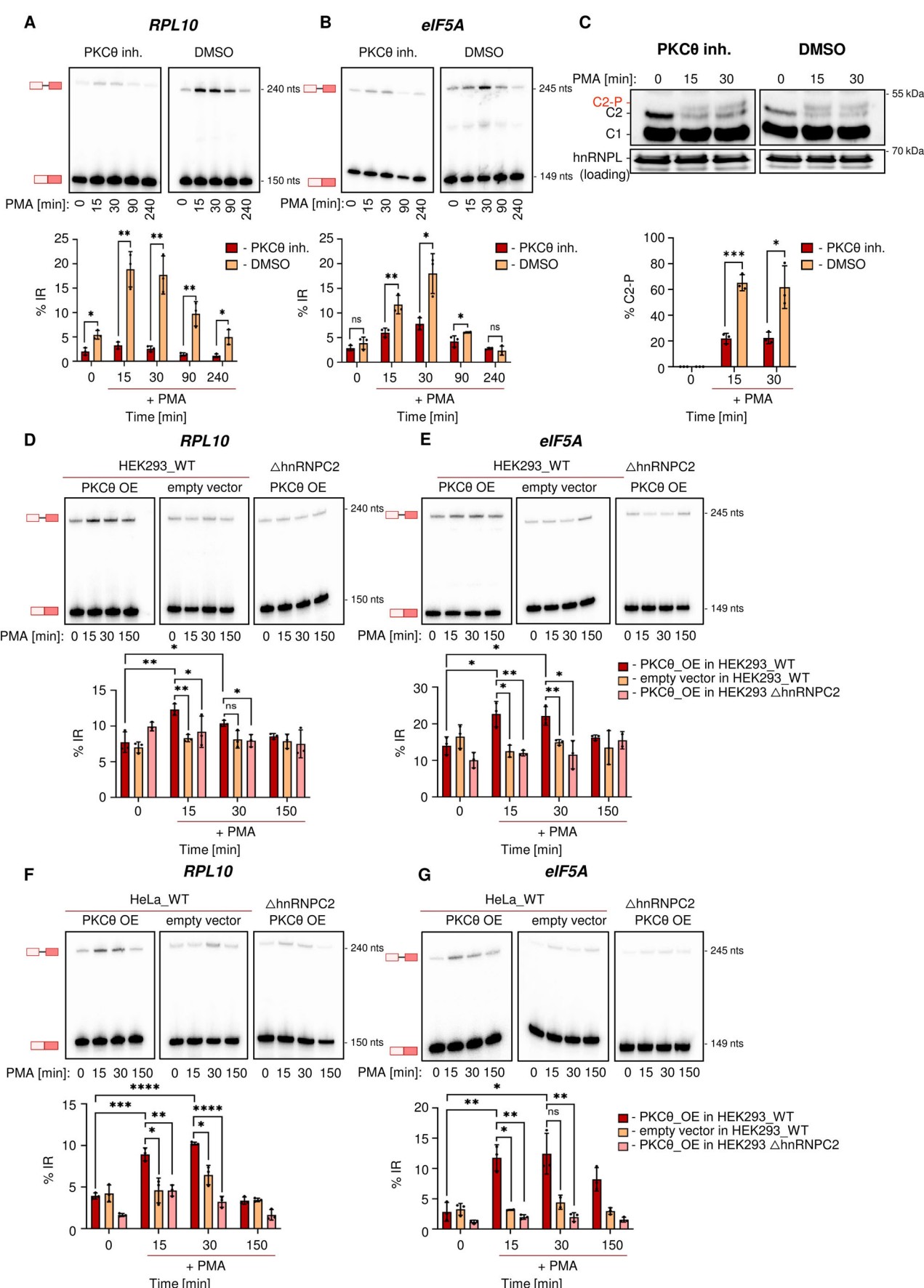

**Figure 5.  T cell-specific IES requires PKCθ.**

(A, B) PKCθ inhibition abolishes IES. After Jurkat cell treatment with PKCθ inhibitor for 30 min (left) or DMSO (right), cells were stimulated with PMA for the indicated times. Chromatin-associated RNA was investigated for *RPL10* (A) and *eIF5A* (B) IES by radioactive, splicing-sensitive RT-PCR and quantified (bottom, data are presented as % IR, mean ± SD, number of biological replicates: $n = 3$, *RPL10* (A): PKCθ inh. vs. DMSO (0 min): $P = 0.0249$, 15 min: $P = 0.0019$, 30 min: $P = 0.0026$, 90 min: $P = 0.0046$, 240 min: $P = 0.0131$, *eIF5A* (B): PKCθ inh. vs. DMSO (0 min): $P = 0.1109$, 15 min: $P = 0.0081$, 30 min: $P = 0.0162$, 90 min: $P = 0.0462$, 240 min: $P = 0.2038$, Student's unpaired *t* test). (C) PKCθ inhibition reduces hnRNPC2 phosphorylation. After Jurkat cell treatment with PKCθ inhibitor (left) or DMSO (right), cells were stimulated with PMA for the indicated times and protein lysates were investigated for phosphorylation of hnRNPC2 by western blot (top) and quantified (bottom, data are presented as % hnRNPC2 phosphorylation of total hnRNPC2, mean ± SD, number of biological replicates: $n = 3$, PKCθ inh. vs. DMSO (15 min): $P = 0.0004$, 30 min: $P = 0.0103$, Student's unpaired *t* test). (D, E) PKCθ overexpression induces IES in an hnRNPC2-dependent manner. HEK293 cells were transfected with an empty FLAG vector or an overexpression (OE) vector for PKCθ. Additionally, HEK293 cells with hnRNPC2-deletion were transfected with an overexpression vector for PKCθ. After 48 h, cells were stimulated by PMA for the indicated time points and chromatin-associated RNA was investigated by radioactive, splicing-sensitive RT-PCR for IR in *RPL10* (D) and *eIF5A* (E). Bottom: data are presented as % IR, mean ± SD, number of biological replicates: $n = 3$, *RPL10* (D): 0 vs. 15 min (PKCθ_OE in HEK293_WT), $P = 0.0088$, 0 vs. 30 min (PKCθ_OE in HEK293_WT), $P = 0.0383$, PKCθ_OE in HEK293_WT vs. empty vector in HEK293_WT (15 min): $P = 0.0018$, PKCθ_OE in HEK293_WT vs. empty vector in HEK293_WT (30 min): $P = 0.0911$, PKCθ_OE in HEK293_WT vs. PKCθ_OE in HEK293_ΔhnRNPC2 (15 min): $P = 0.01$, PKCθ_OE in HEK293_WT vs. PKCθ_OE in HEK293_ΔhnRNPC2 (30 min): $P = 0.0119$, *eIF5A* (E): 0 vs. 15 min (PKCθ_OE in HEK293_WT), $P = 0.024$, 0 vs. 30 min (PKCθ_OE in HEK293_WT), $P = 0.016$, PKCθ_OE in HEK293_WT vs. empty vector in HEK293_WT (15 min): $P = 0.0115$, PKCθ_OE in HEK293_WT vs. empty vector in HEK293_WT (30 min): $P = 0.009$, PKCθ_OE in HEK293_WT vs. PKCθ_OE in HEK293_ΔhnRNPC2 (15 min): $P = 0.001$, PKCθ_OE in HEK293_WT vs. PKCθ_OE in HEK293_ΔhnRNPC2 (30 min): $P = 0.0322$, Student's unpaired *t* test). (F, G) Analysis as in (D, E) using HeLa cells (*RPL10* (F): 0 vs. 15 min (PKCθ_OE in HeLa_WT), $P = 0.0005$, 0 vs. 30 min (PKCθ_OE in HeLa_WT), $P < 0.0001$, PKCθ_OE in HeLa_WT vs. empty vector in HeLa_WT (15 min): $P = 0.011$, PKCθ_OE in HeLa_WT vs. empty vector in HeLa_WT (30 min): $P = 0.0163$, PKCθ_OE in HeLa_WT vs. PKCθ_OE in HeLa_ΔhnRNPC2 (15 min): $P = 0.0018$, PKCθ_OE in HeLa_WT vs. PKCθ_OE in HeLa_ΔhnRNPC2 (30 min): $P < 0.0001$, *eIF5A* (G): 0 vs. 15 min (PKCθ_OE in HeLa_WT), $P = 0.0044$, 0 vs. 30 min (PKCθ_OE in HeLa_WT), $P = 0.011$, PKCθ_OE in HeLa_WT vs. empty vector in HeLa_WT (15 min): $P = 0.0128$, PKCθ_OE in HeLa_WT vs. empty vector in HeLa_WT (30 min): $P = 0.0531$, PKCθ_OE in HeLa_WT vs. PKCθ_OE in HeLa_ΔhnRNPC2 (15 min): $P = 0.0015$, PKCθ_OE in HeLa_WT vs. PKCθ_OE in HeLa_ΔhnRNPC2 (30 min): $P = 0.0063$ (Student's unpaired *t* test). Source data are available online for this figure.

retention in Jurkat T cells (Fig. 6E). Using these conditions, we also confirmed the presence of the EIF5A IES product in the cytoplasm (Fig. EV5D). WB-SuNSET showed reduced de novo protein synthesis in cells with increased eIF5A intron retention (Fig. 6F). This confirms that altering IES of a single event can globally control translation efficiency. As the level of eIF5A intron retention in this MO experiment was higher than that observed through IES during T cell activation, we titrated the MO down to a level that induced IR levels comparable to that observed in activated T cells (Fig. EV5E). This experiment shows a gradual decrease in de novo protein synthesis which already starts at low MO concentrations that only cause a moderate effect on IR (Fig. EV5F), confirming the conclusion also for endogenous IR levels. In addition, we suggest that the accumulating effect of several or many smaller IES changes in components of the translation machinery can have a substantial overall effect, either by creating inhibiting, potentially dominant negative protein variants (eIF5A) or by reducing the amount of expressed protein through inclusion of PTCs (*RPL10*, Fig. EV5G for *RPL10*). These data are consistent with a model in which a concerted IES switch, controlled by hnRNPC2, targets the translation machinery to transiently reduce de novo protein synthesis in the early hours after T cell activation.

## Discussion

Mechanism and functionality of immediate early genes (IEGs) have been analyzed for decades and their essential role in establishing cellular responses to diverse stimuli is well established (Fowler et al, 2011). Here we show that one of the major signaling cascades controlling IEGs, MEK-ERK signaling, also targets the splicing machinery. The response is a fast and transient splicing switch independent of de novo protein synthesis that we therefore call immediate early splicing (IES). It has been a common assumption that alternative splicing is a mechanism that allows a very fast adaptation of the transcriptome in response to changes in the cellular environment (Habib et al, 2021; Kucherenko and Shcherbata,

2018), but an immediate change in the splicing program and ensuing functionality has remained unexplored.

Here we use T cell activation as a model to characterize the mechanistic basis and functionality of an IES switch. T cell activation requires major transcriptomic/proteomic adaptation to allow fundamental changes in cellular functionality. Many studies have analyzed different phases of T cell activation at different levels, from signaling cascades initiated within seconds of T cell receptor engagement to the formation of long-lasting memory T cells upon infection in in vivo models (Hope et al, 2019; Hwang et al, 2020; Jankowska et al, 2018; Sánchez-Escabias et al, 2022). Alternative splicing is a mechanism that acts largely independent of changes in de novo transcription to induce transcriptomic changes upon T cell activation (Ip et al, 2007). While activation induced alternative splicing has been analyzed at later timepoints upon stimulation, i.e. 24 h or 48 h post-induction (Martinez and Lynch, 2013), its role in the early phase of T cell activation has remained elusive. Immediately upon T cell activation, phosphorylation cascades alter activity and interactions of existing proteins that prepare the cell for long-term functional changes. For example, T cell activation induces IEGs that then contribute to fundamentally alter the transcriptome at later stages of activation. At the same time, de novo protein synthesis is reduced in the hours following T cell activation. This likely is a coordinated reaction to allow faster protein turnover and the formation of an 'activated proteome'. The preexisting mRNAs from the resting phase are translated with reduced efficiency and only once major transcriptomic changes have been implemented (16–24 h post-stimulation), translation efficiency is strongly increased (Rade et al, 2023; Tan et al, 2017). Our data suggest that hnRNPC2 leading to IES plays a fundamental role in coordinating this response, as it is instrumental to transiently decrease translation efficiency. The IES event in eIF5A is on its own sufficient to reduce translation efficiency when manipulated using an anti-sense MO, also when MO-induced splicing changes are within the magnitude of endogenous IR levels. This suggests that this mRNA variant is translated, with the

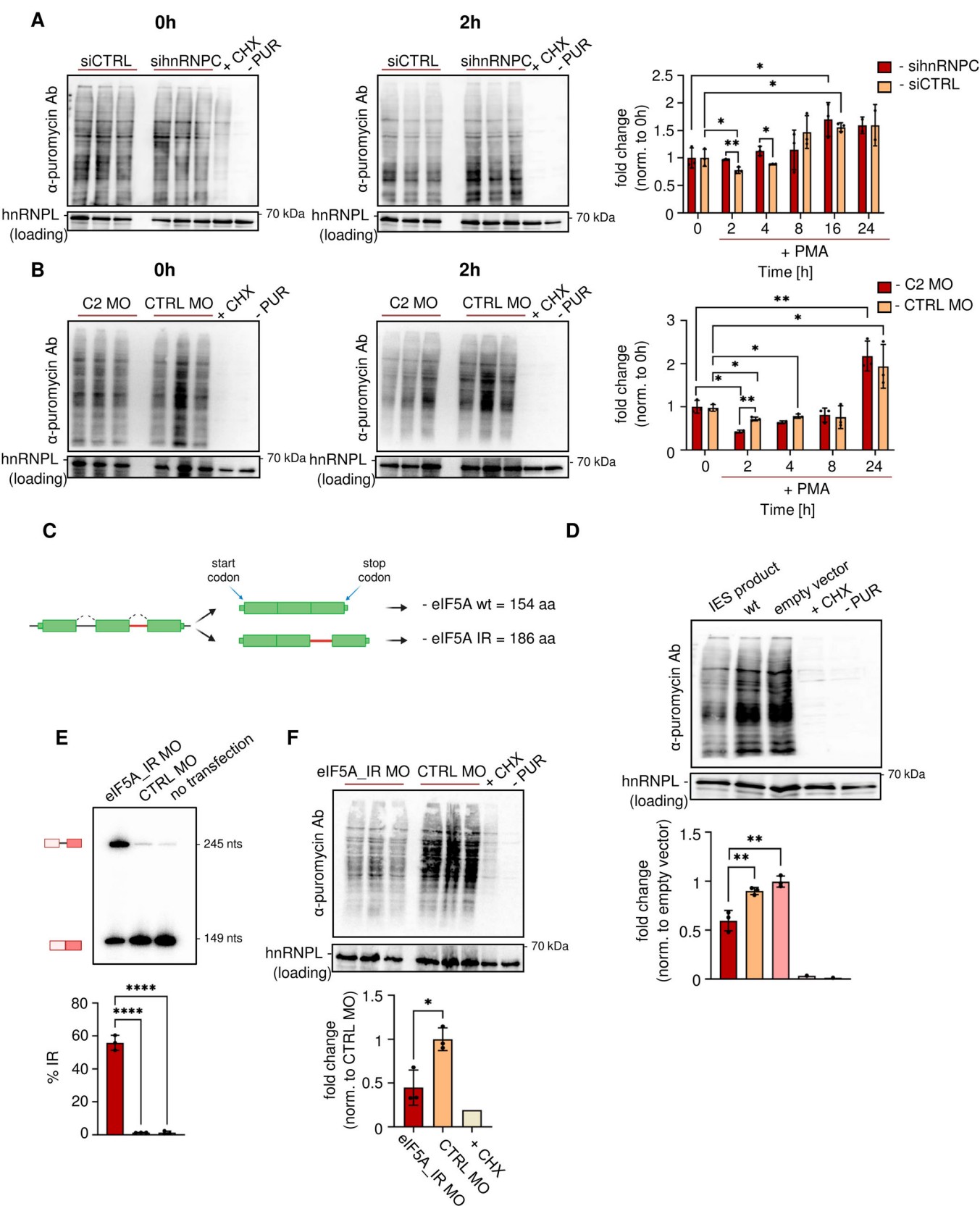

◄ **Figure 6. hnRNPC-mediated IES reduces global translation at the early stage of T cell activation.**

(A) De novo translation in Jurkat cells after PMA activation in control and hnRNPC knockdown conditions at indicated time points. Cells were treated with 10 µg/ml of puromycin 10 min before harvesting. Total protein was extracted, and WB-SUnSET was performed. Left: representative gels showing decreased global translation at early stages of T cell activation (0 h vs. 2 h) only in siCTRL-electroporated cells. Cycloheximide was added 1 h before harvesting as a control of treatment. -PUR—no puromycin treatment. hnRNPL serves as a loading control. Right: corresponding quantification (data are presented as fold change normalized to 0 h, mean ± SD, number of biological replicates: $n = 3$, 0 vs. 2 h (siCTRL), $P = 0.0485$, 0 vs. 16 h (siCTRL), $P = 0.0121$, 0 vs. 16 h (sihnRNPC), $P = 0.0269$, sihnRNPC vs. siCTRL (2 h), $P = 0.0081$, sihnRNPC vs. siCTRL (4 h), $P = 0.0319$, Student's unpaired $t$ test. Gels from other time points are presented in Fig. EV4A. (B) De novo translation in Jurkat cells after PMA activation in hnRNPC2-inducing MO (C2 MO) and CTRL MO conditions at indicated time points. Cells were treated as in (A). Left: representative gels showing decreased global translation at early stages of T cell activation (0 h vs. 2 h) in C2 MO-electroporated cells. Right: corresponding quantification (data are presented as fold change normalized to 0 h, mean ± SD, number of biological replicates: $n = 3$, 0 vs. 2 h (C2 MO), $P = 0.0146$, 0 vs. 2 h (CTRL MO), $P = 0.0156$, 0 vs. 4 h (CTRL MO), $P = 0.0447$, 0 vs. 24 h (C2 MO), $P = 0.0058$, 0 vs. 24 h (CTRL MO), $P = 0.0224$, C2 MO vs. CTRL MO (2 h), $P = 0.0039$, Student's unpaired $t$ test. Gels from other time points are presented in Fig. EV4C. (C) A schematic representation of eIF5A pre-mRNA with (eIF5A_IR) and without retained intron (eIF5A_WT). Green boxes represent exons, dashed lines represent introns. Red connecting line represents the IES event. Created with BioRender.com. (D) EIF5A_IR inhibits translation. HEK293 cells overexpressing eIF5A wt or IR variants for 48 h were treated with 10 µg/ml of puromycin 10 min before harvesting. Total protein was extracted. Top: representative blot of the WB-SUnSET experiment. hnRNPL serves as a loading control. Bottom: corresponding quantification (data are presented as fold change normalized to empty vector, mean ± SD, number of biological replicates: $n = 3$, IES product vs. wt, $P = 0.0088$, IES product vs. empty vector, $P = 0.0043$). (E) Jurkat cells were electroporated with either eIF5A_IR MO to induce intron retention or control MO for 48 h. Cells were harvested and total RNA was extracted. Additionally, non-transfected cells were used as control. Top: the efficiency of eIF5A_IR MO was analyzed by radioactive, splicing-sensitive PCR and quantified (bottom, data are presented as % IR, mean ± SD, number of biological replicates: $n = 3$, eIF5A_IR MO vs. CTRL MO, $P < 0.0001$, eIF5A_IR MO vs. no transfection, $P < 0.0001$, Student's unpaired $t$ test). (F) Jurkat cells as in (D) were treated with 10 µg/ml of puromycin 10 min before harvesting. Total protein was extracted. Top: representative blot of the WB-SUnSET experiment shows decreased global translation in Jurkat cells with electroporated eIF5A_IR MO. Cycloheximide (+ CHX) was added 1 h before harvesting as a control. -PUR—no puromycin treatment. hnRNPL serves as a loading control. Bottom: corresponding quantification (data are presented as fold change normalized to CTRL MO, mean ± SD, number of biological replicates: $n = 3$, $P = 0.0162$, Student's unpaired $t$ test). Source data are available online for this figure.

resulting protein acting to globally reduce translation. This argues for this intron to be an in-frame, retained intron, rather than a detained intron, as we find it to be exported to the cytoplasm. However, we can not exclude that some of the IES variants are also detained introns that hold their partially spliced RNAs in the nucleus and that are either degraded or spliced and exported at later time points after stimulation (Boutz et al, 2015). We suggest that concerted IES events in many components of the translation machinery can have a substantial inhibiting effect on de novo translation, especially as some of the IES variants may act in a dominant negative manner.

Our data implicate hnRNPC2 as essential mediator of IES during T cell activation. We suggest that hnRNPC1/2 are required for efficient splicing of the introns that are affected by IES and that phosphorylation of hnRNPC2 at Ser115 reduces binding affinity to these introns thereby reducing splicing efficiency. It will be interesting to analyze specific requirements that render an intron susceptible to regulation through hnRNPC2-mediated IES. This likely depends on specific sequences, as binding to a polyU RNA is not affected by the phosphomimetic hnRNPC2 S115D mutation, whereas binding to IES targets is, and may be additionally influenced by intron length. hnRNPC has rather been associated with reduced splicing efficiency, for example by competing with U2AF65 for binding to poly-pyrimidine tracts (Zarnack et al, 2013). However, a splice-activating role of hnRNPC has also been described (Moon et al, 2019), which is in line with the model we propose here. The two hnRNPC isoforms, hnRNPC1 and hnRNPC2, have been known for a long time, but functional differences have not been analyzed. Our data suggest that phosphorylation of hnRNPC2 alters RNA binding, but somewhat surprisingly, altering the hnRNPC1:C2 ratio did not result in major changes in the transcriptome of HEK293 or HeLa cells. Given the role of hnRNPC heterotetramers (C1:C2 = 3:1) in forming an 'RNA nucleosome' these data suggest redundancy of the two hnRNPC isoforms in this function (McAfee et al, 1996). This also indicates that the precise C1:C2 ratio in these tetramers is not an essential prerequisite for their functionality.

Phosphorylation of hnRNPC2 required MEK-ERK signaling as well as the activity of PKCθ. While the precise role of PKCθ remains unknown, several scenarios are possible. PKCθ could be required for a priming phosphorylation on hnRNPC2 that then allows ERK-mediated phosphorylation of S115. PKCθ may also phosphorylate ERK or another adaptor molecule to bring hnRNPC and ERK together in one protein complex, which could also be achieved by a scaffolding function of PKCθ, independent of its catalytic activity. PKC-mediated phosphorylation of hnRNPC has been described (Martino et al, 2022), which may point to a priming phosphorylation, but this remains speculative. While the precise mode of action remains unknown, the involvement of PKCθ renders this particular IES program cell type specific (Brezar et al, 2015). This indicates that the coordinated regulation of transcription and translation that is achieved through hnRNPC2-controlled IES is particularly important during T cell activation.

Our mass spectrometry data suggest that several other splicing regulatory proteins show a transient phosphorylation upon T cell activation that would be consistent with controlling IES events. In particular interesting is U2AF35, the protein that directly binds the AG at the 3' splice site (Wu et al, 1999). Phosphorylation could alter RNA affinity or binding specificity, which could contribute to regulate some IR events and may also result in changes in cassette exon inclusion or 3' splice site selection. This may affect additional exons, introns or splice sites under different conditions and in other cell lines. For example, neurons and the immediate response to depolarization is a model system that will be very interesting to be analyzed (Minatohara et al, 2015). We did not observe hnRNPC2-mediated IES in N2a cells, but other targets controlled by other RBPs may contribute to control transcriptomic changes and cellular functionality in this setting. Altogether, we show that the same signaling cascades that regulate IEGs can also target the splicing machinery to control IES. Given this fundamental yet intuitive concept, we consider it likely that IES plays an essential role in controlling the cellular response in diverse conditions, which, however, remains to be analyzed in future research.

# Methods

### Reagents and tools table

| Reagent/resource | Reference or source | Identifier or catalog number |
|---|---|---|
| **Experimental models** | | |
| Top10 competent *E. coli* | This study | N/A |
| BL21 RIL *E. coli* | Novagen | 70954 |
| Jurkat | Kristen Lynch, UPenn | N/A |
| Primary mouse T cells | This study | N/A |
| EL4 | ATCC, USA | TIB-39™ |
| HEK293 | ATCC, USA | CRL-1573™ |
| HeLa | ATCC, USA | CCL-2™ |
| RAW264.7 | ATCC, USA | TIB-71™ |
| N2a | MDC, Berlin | N/A |
| **Recombinant DNA** | | |
| pCDNA | Thermo Fisher Scientific | V79020 |
| TwinStrep-SUMO tagged hnRNPC2 construct | Thermo Fisher Scientific | |
| pBS-PKCθ | Mischak et al, 1993 | #8426 |
| pFLAG-N3-PKCθ | This study | N/A |
| pTWIST CMV_EIF5A_WT | Twist Bioscience | N/A |
| pTWIST CMV_EIF5A_IR | Twist Bioscience | N/A |
| pFLAG-N3-hnRNPC2_WT | This study | N/A |
| pFLAG-N3- hnRNPC2_S115D | This study | N/A |
| pFLAG-N3- hnRNPC2_S→A/Y | This study | N/A |
| pSpCas9(BB)-2A-GFP (pX458) | Addgene | #48138 |
| **Antibodies (dilution)** | | |
| hnRNP C1/C2 (1:1000) | Santa Cruz Biotechnology | sc-32308 |
| p-ERK (1:1000) | Santa Cruz Biotechnology | sc-7383 |
| ERK ½ (1:1000) | Santa Cruz Biotechnology | sc-514302 |
| DYKDDDDK Tag (1:2000) | Cell Signaling Technology | #14793 |
| GAPDH (1:2000) | Santa Cruz Biotechnology | sc-32233 |
| hnRNPL (1:2000) | Santa Cruz Biotechnology | sc-32317 |
| puromycin (1:5000) | Sigma-Aldrich | 12D10 |
| anti-mouse IgG HRP, (1:5000) | Cell Signaling Technology | #7076s |
| anti-rabbit IgG HRP, (1:5000) | Cell Signaling Technology | #7074s |
| **Oligonucleotides and other sequence-based reagents** | | |
| Anti-sense morpholinos | Gene Tools | Table 1 |
| siRNA sequences | Eurofins Scientific | Table 2 |
| gRNAs #1 5'-CTCTAC TCAGGTCCGGAAC | Eurofins Scientific | N/A |
| #2: 5' -CTGCATTGT GTCCATCAGT | Eurofins Scientific | N/A |
| RT-PCR primers | Eurofins Scientific | Table 3 |
| RNA sequences | Eurofins Scientific | Table 4 |

| Reagent/resource | Reference or source | Identifier or catalog number |
|---|---|---|
| **Chemicals, enzymes and other reagents** | | |
| DMEM High Glucose | Biowest | L0104 |
| Trypsin | Gibco | 25200-056 |
| RPMI 1640 | Biowest | L0500 |
| Phorbol 12-myristate 13-acetate | Sigma-Aldrich | P8139 |
| anti-CD3 antibody (1 µg/ml) | BioLegend | OKT3 |
| Lipopolysaccharide (LPS) | Thermo Fisher Scientific | 00-4976-03 |
| FastAP Thermosensitive Alkaline Phosphatase | Thermo Fisher Scientific | EF0654 |
| Potassium chloride | Carl Roth. Germany | 6781.1 |
| cycloheximide | Sigma-Aldrich | 239763-M |
| PD0325901 (MEK1/2 inhibitor) | New England Biolabs | 9903S |
| MK2206 (AKT1/2/3 inhibitor) | Biomol | SYN-1162-M001 |
| VX-745 (p38 inhibitor) | Biomol | Cay18075-1 |
| Ruxolitinib (JAK1/2 inhibitor) | Biomol | Cay11609-1 |
| CD28 | Thermo Fisher Scientific | AB_468926 |
| PKCθ inhibitor | Selleck | S6577 |
| DNase I | Biozym | 170500 |
| ROTIFect | Carl Roth, Germany | P001.1 |
| Lipofectamine 2000 | Thermo Fisher Scientific | 11668019 |
| Opti-MEM (Opti-MEM® I (1X) + GlutaMAX™-I - Reduced Serum Medium | Thermo Fisher Scientific | 51985091 |
| PBS | Biowest | L0615-500 |
| BpiI | Life Technologies | FD1014 |
| puromycin (1 µg/ml) | InvivoGen | ant-pr-1 |
| proteinase K (20 mg/mL) | Serva | 33755.01 |
| The Quick-Load® 1 kb Plus DNA Ladder | New England Biolabs Inc. | N0550S |
| ROTI®Aqua-P/C/I | Carl Roth | X985.1 |
| RNA-se inhibitor, RiboLock | Life Technologies | EO0382 |
| MMuLV reverse transcriptase | Enzymatics Inc. | 280550 |
| T4 PNK (10 U/ml) | Life Technologies | EK0032 |
| γ-$^{32}$P-dATP (10 µCI/µl, 6000 Ci/mmol, | Hartmann Analytic | SRP-501 |
| Absolute QPCR SYBR Green Mix | Biozym | 331416 |
| XhoI | Life Technologies | FD00695 |
| NheI | Life Technologies | FD0974 |
| PMSF -1:100 | Carl Roth | 6367.2 |
| Aprotinin-1:200 | Carl Roth | A162.3 |
| Leupeptin-1:200 | Carl Roth | CN33.3 |
| Vanadate-1:500 | Sigma-Aldrich | S6508-50G |
| Coomassie Brilliant Blue G250 | Carl Roth | 9598.1 |

| Reagent/resource | Reference or source | Identifier or catalog number |
|---|---|---|
| BSA bovine serum albumin | Carl Roth | 8076.4 |
| PageRuler Plus Pre-stained protein ladder | Life Technologies | 26620 |
| $^{35}$S-Met ( > 1,000 Ci (37.0TBq)/mmol | Hartmann Analytic | KSM-01 |
| 0.4 mM IPTG | Carl Roth | 2316.4 |
| RNA Tri-flüssig | BIO&SELL | BS67.211.0100 |
| Trichloromethane/ Chloroform | Carl Roth | 7331.2 |
| The marker pBR322-Mspl Digest | New England Biolabs Inc. | N3032 S |
| Pierce™ ECL Western Blotting Substrate | Life Technologies | 32106 |
| fetal bovine serum | BIO&SELL | FBS.S0613 |
| penicillin/streptomycin | Biowest | L0022 |
| BME | Sigma-Aldrich | M6250-10ML |
| Desthiobiotin | IBA Lifesciences | 2-1000-002 |
| **Software** | | |
| Sanger sequencing | https://www.microsynth.com/home-de.html | N/A |
| GraphPad Prism | https://www.graphpad.com | 10.3.0 |
| ImageQuant | Provided by GE Healthcare | TL 8.1 |
| Python | https://www.python.org/ | v3.10.12 |
| Pandas | https://pandas.pydata.org/ | v2.2.1 |
| Biopython | https://biopython.org/ | v.1.78 |
| STAR | https://github.com/alexdobin/STAR | v2.7.9a |
| rMTAS turbo | https://github.com/Xinglab/rmats-turbo | v4.1.1 |
| RSeQC | https://rseqc.sourceforge.net/ | v4.0.0 |
| DESeq2 | https://bioconductor.org/packages/release/bioc/html/DESeq2.html | v1.28.1 |
| ShinyGO | http://bioinformatics.sdstate.edu/go/ | v0.80 |
| bedtools | https://bedtools.readthedocs.io/en/latest/ | N/A |
| NGC Quest 10 Plus System, ChromeLab software | https://www.bio-rad.com/de-de/product/chromlab-software?ID=MFCVPXIVK | N/A |
| **Other** | | |
| Gene Pulser electroporation system | Bio-Rad | N/A |
| Nucleofector II | Amaxa Biosystems | N/A |
| S1000 Thermal Cycler | Bio-Rad | N/A |
| PCR Clean-up Kit | Macherey-Nagel | 740609.250 |
| NucleoBond® Xtra Midi Kit | Macherey-Nagel | 740410.100 |

| Reagent/resource | Reference or source | Identifier or catalog number |
|---|---|---|
| BD FACS Melody™ Cell Sorter | BD Biosciences | 664341 |
| UVsolo touch system | Analytik Jena, Germany | 849-00502-2 |
| heating block | Biozym | 551010 |
| Nanophotometer | Serva | 055308522 |
| Al600 RGB GEL Imaging System | GE Healthcare | N/A |
| Illustra MicroSpin G-25 columns | GE Healthcare | 27-5325-01 |
| Stratagene Mx3000P instrument | Bioer Technology Co | N/A |
| EASY-nLC 1000 system | Thermo Fisher Scientific | LC120 |
| Amersham Typhoon 9200 | GE Healthcare | N/A |
| Q Exactive HF-X mass spectrometer (MS) | Thermo Fisher Scientific | N/A |
| StrepTrap™ 5 mL column | GE Healthcare | N/A |
| Superdex 200 16/60 size-exclusion column | GE Healthcare | N/A |
| Coulter Avanti J-25 High-Speed Centrifuge | Beckman | 367501 |
| Sonopuls ultrasonic homogenizer | Bandelin | HD-500 |

## Cell culture

HEK293, HeLa, EL4, RAW264.7, N2a and Jurkat cell stocks were maintained in liquid nitrogen, and early passage aliquots were thawed periodically. HEK293, HeLa, RAW264.7 and N2a cells were maintained in DMEM High Glucose (Biowest), supplemented with 10% fetal bovine serum (FBS; Biochrom) and 1% penicillin/streptomycin (Biowest). Jurkat and EL4 cells were maintained in RPMI 1640 (Biowest) supplemented with 10% FBS and 1% penicillin/streptomycin. The cells were tested for mycoplasma contamination monthly using a PCR-based assay. All cell lines were maintained at 37 °C and 5% $CO_2$. Mouse primary T cells were isolated from lymph nodes. They were cultured in RPMI medium (containing 2 mM glutamine, 1 mM pyruvate, nonessential amino acids, and 50 μM 2-mercaptoethanol) supplemented with 10% FBS and 1% penicillin-streptomycin.

Cells were stimulated with 20 ng/ml of phorbol myristate acetate (PMA, Sigma-Aldrich) or dimethyl sulfoxide (DMSO) as solvent control for different time points depending on the experimental setup. As the alternative for PMA, coated anti-CD3 antibody (1 μg/ml or 2 μg/ml) and anti-CD28 antibody (1 μg/ml) were used to activate Jurkat cells. Furthermore, RAW264.7 cells were stimulated by 0.1 μg/ml of LPS (Thermo Fisher Scientific, #00-4976-03) and N2a cells were stimulated by 60 mM of KCl. For translation inhibition, cycloheximide (Sigma-Aldrich) was used at 40 ng/ml final concentration or DMSO as solvent control. Small molecule inhibitors were used at the following concentrations: PD0325901 (MEK1/2), MK2206 (AKT1/2/3), VX-745 (p38), Ruxolitinib (JAK1/2), JNK inhibitor VIII (JNK1/2/3) (10 μM).

**Table 1. Anti-sense morpholinos used in this study.**

| Anti-sense morpholino | Sequence (5′ → 3′) |
|---|---|
| hnRNPC2 | GGTGTTCTGTTACTGACCCGTACAT |
| eIF5A_IR | AGGGAGGCACCATACCAGGATCTCT |
| Standard control | CCTCTTACCTCAGTTACAATTTATA |

**Table 2. Sequence of the siRNAs against genes of interest.**

| Name | Target sequence (5′ → 3′) |
|---|---|
| hnRNPC | CUUAAAUAGGAGAGGCUCA |
| | GUAAGUAACCCGUGACUAG |
| | UUUCAUAGCAUGCGGCACU |
| | CCUAGGCGCUUGUCUAAGA |
| hnRNPK | UAAACGCCCUGCAGAAGAU |
| | GGUCGUGGCUCAUAUGGUG |
| | UGACAGAGUUGUUCUUAUU |
| | GCAAGAAUAUUAAGGCUCU |
| U2AF35 | CGCCGUCGCAAGAAGCAUA |
| | UGACCAAACCAGUUCAUAA |
| | UAGAAAGUGUUGUAGUUGA |
| | CAAGUUUCGCCGUGAGGAA |
| PKCθ | AAUUGACAUGCCACACAGA |
| | UUAGAAUUCCCAACCAUACA |
| siAllstar (control) | UUCUCCGAACGUGUCACGU |

For PKCθ inhibition experiment, Jurkat cells were pretreated with PKCθ inhibitor (C20, Selleck, Houston, TX, USA) or DMSO for 30 min followed by the indicated procedures. C20 was dissolved in DMSO at 5 mM and diluted to the working concentration (5 μM).

## RNA sequencing

For RNA-seq, Jurkat cells were seeded on T175 flasks and grown for ~48 h 37 °C. Cells were then stimulated by PMA for 0 min (as a control), 30 min and 150 min. Chromatin-associated RNA was extracted as described below. Sequencing was performed on an Illumina HiSeq 2500 system with V4 sequencing chemistry, generating around 40 million 125 bp paired-end reads per sample. For morpholino treated HeLa and HEK293 cells (see below), triplicate DNase I-digested RNA samples of CTRL or hnRNPC2 inducing morpholino samples were used for library preparation. Libraries were prepared using the mRNA enrichment method at BGI Genomics and sequenced using the Eukaryotic Strand-specific Transcriptome Resequencing PE150 technique. This yielded around 40 million paired-end 150 nt reads for all samples. Sequencing data are available under GSE271051 (reviewer access token: mzmpqeqgflkjvch). The raw sequenced reads were aligned to the human hg38 reference genome with STAR (v2.7.9a) default parameters, resulting in ~65% (Jurkat) or ~95% (HeLa or HEK293) unique alignment rate. Alternative splicing analysis was done with rMTAS turbo (v4.1.1), including variable-read-length criteria. We focused on alternative splicing events of SE, RI, A5SS and A3SS. Events were considered significant between two conditions if the

mean absolute difference in percent spliced in (PSI) was larger than +/− 0.15, if the FDR was smaller than 0.01 and if a splicing event had at least 100 combined junction reads across the tested samples.

For the PCA plot and heatmap PSI values of significant IR events comparing 0- and 30-min PMA treatment were used. PCA and PCA variance were calculated using a Python script. In the heatmap, events are sorted by fold change in mean ΔPSI values between 0 and 30 min. STAR (v2.7.9a) was run in GenCounts mode to allow gene-level quantification. Gene counts provided by STAR were tested for strandness using RSeQC (v4.0.0). A Python script, with the help of Pandas library (v2.2.1), was used to create the input files for downstream differential gene expression analysis with DESeq2 (v1.28.1). Significantly differential genes were identified by filtering on Benjamini-Hochberg adjusted p-value (Padj) $\leq 0.001$ and absolute $\log_2 FC \geq 1$ (or $\leq -1$), which did not reveal any significant changes for morpholino treated HeLa or HEK293 samples. Gene ontology (GO) term enrichment was performed using ShinyGO (v0.80) for GO biological processes, comparing genes with retained introns after 30 min with all genes containing IR events quantified in Jurkat cells.

Intron length was calculated with Python using the rMATS coordinates of the upstream and downstream exons. To analyze the splice site strength of introns, the respective coordinates were extracted from the rMATS output table in bed format ($-3$ to $+6$ for 5′-ss and $-20$ to $+3$ for 3′-ss). Bedtools getfasta was then used to extract splice site sequences. Sequences were input to MaxEntScan (Yeo and Burge, 2004), using the Maximum Entropy Model to score. To calculate the magnitude of the intron retention effect, we calculated the maximal PSI change between 0 and 30 min or 30 and 150 min of all 173 IR events significantly increased at 30 min in either of the two comparisons. Events were then clustered into bins centred between $-0.15$ and $-0.65$ (in 0.05 steps) using GraphPad Prism. Results are presented as a histogram of frequency distributions. To calculate the protein-coding potential of these 173 introns, we next retrieved information on these introns from the human annotation genome gtf file and sequencing fasta file (version GRCh38.111) using a Python script with loaded Pandas v.2.2.1 and Biopython v.1.78 libraries. First, it was confirmed whether the introns are within protein-coding genes. If yes, we tested whether the introns were positioned within an open reading frame (ORF) of the encoded gene. If yes, we checked whether the intron encodes a PTC by introducing a frameshift or directly harbouring a stop codon. We considered introns within the ORF of protein-coding genes as having coding potential if they are (i) frame-preserving and do not introduce a stop codon or (ii) if the introduced stop (directly or by frameshift) lies within the last intron.

## Transfections

Transfections of HEK293 and HeLa cells using either ROTIFect (Carl Roth) or Lipofectamine 2000, respectively were performed according to the manufacturer's instructions. Transfections of Jurkat cells were performed using a Gene Pulser electroporation system (Bio-Rad) for morpholino experiments and Nucleofector II from Amaxa Biosystems for siRNAs transfections. For morpholino experiments, HEK293 cells were seeded and transfected 1 day later using Endo-Porter following the manufacturer's manual. Jurkat cells were washed twice with Opti-MEM® I (1X) + GlutaMAX™-I –

**Table 3. A list of PCR primer sequences.**

| Name | Target sequence (5′ → 3′) | Goal |
|---|---|---|
| hnRNPC2_F | ACTTTGTAGTTTGTTTTACCCGG | RT-qPCR |
| hnRNPC2_R | GGGTGTGGGGAGGTTTAAGT | RT-qPCR |
| mHPRT_qF | CAACGGGGGACATAAAAGTTATTGGTGGA | RT-qPCR |
| mHPRT_qR | TGCAACCTTAACCATTTTGGGGCTGT | RT-qPCR |
| hHPRT_qF | CCTGGCGTCGTGATTAGTGA | RT-qPCR |
| hHPRT_qR | TCTCGAGCAAGACGTTCAGT | RT-qPCR |
| mPKCθ_F | CAGGGACCTGAAGCTTGATAAT | RT-qPCR |
| mPKCθ_R | GCATCTCCTAGCATGTTCTCTT | RT-qPCR |
| hPKCθ_F | GAGGACAAGTGGAAAGTGAGAG | RT-qPCR |
| hPKCθ_R | CTTTCCATCCACCCATTCTCA | RT-qPCR |
| NheI_PKCθ_F | AATTGCTAGCATGTCACCGTTTCTTCGGTC | Cloning |
| XhoI_PKCθ_R | AATTTCTCGAGTCAGGAGCAAATGAGAGTCT | Cloning |
| c-FOS_F | CAA GCG GAG ACA GAC CAA CT | RT-qPCR |
| c-FOS_R | GTG AGC TGC CAG GAT GAA CT | RT-qPCR |
| hTraf4_F | CACCTCTGAGTGCCCCAAG | Splicing PCR |
| hTraf4_R | AGCCGGAGTCTTTGAATGGG | Splicing PCR |
| mTraf4_F | AGGTCCAGGTGTTAGGCTTGG | Splicing PCR |
| mTraf4_R | CCACTGAAGTCACAGCCACAG | Splicing PCR |
| RPL7A_F | TTAACACCGTCACCACCTTG | Splicing PCR |
| RPL7A_R | CAGTCTTGCCTTTCCCTTGA | Splicing PCR |
| hRPL10_F | GTAAGAACAAGCCGTACCCA | Splicing PCR |
| hRPL10_R | AGGACAGCTGCTCATATTCATCT | Splicing PCR |
| mRPL10_F | GGTATTGTAAGAACAAGCCATACC | Splicing PCR |
| mRPL10_R | CCACAAAGTGGGAATTCATCAAC | Splicing PCR |
| RPL13_F | GTTCGGTACCACACGAAGGT | Splicing PCR |
| RPL13_R | GTTGGCCTGCAGGGACTC | Splicing PCR |
| hEIF5A_F | CAGGACAGCGGGGAGGTA | Splicing PCR |
| hEIF5A_R | ATGGCCTTGATTGCAACAGC | Splicing PCR |
| mEIF5A_F | TACCTATCCCTGCTCCAGGAC | Splicing PCR |
| mEIF5A_R | GCCTTGATTGCAACAGCTGC | Splicing PCR |
| UBXN1_F | GCAATTGCTCAGTGGCTTC | Splicing PCR |
| UBXN1_R | GGACATTTCTTGGCCACAAT | Splicing PCR |
| USP11_F | CCTGGTCAGCTGGTATGGTC | Splicing PCR |
| USP11_R | ATTGTGCCGGACAAGCAG | Splicing PCR |
| HindIII_hnRNPC_F | ATATAAGCTTATGGCCAGCAACGTTACCA | Cloning |
| BamHI_hnRNPC_R | ATATGGATCCTTAAGAGTCATCCTCGCCATTGGC | Cloning |
| hnRNPC_S/Y-A_F | GCCGGGGCAGTAACAGAACACCCTGC | Cloning |
| hnRNPC_S/Y-A_R | AGCGTCCAAGTCAAAAGCGGCGGGGA | Cloning |
| BamHI_hnRNPC_FLAG_R | ATATGGATCCAGAGTCATCCTCGCCATTGG | Cloning |
| C2FL_S115D_FP | GAAGAGGACAGCAGAGGATCGGGAGAGGGGTGCTCAG | Cloning |
| C2FL_S115D_RP | CTGAGCACCCCTCTCCCGATCCTCTGCTGTCCTCTTC | Cloning |
| hnRNPC _F | TAGCAGGAGAGGATGGCAGA | RNA genotyping |
| hnRNPC _R | TCCCGTTGAAAGTCATAGTCCA | RNA genotyping |

**Table 4. Sequences of RNA oligonucleotides.**

| Name | Sequence (5′→ 3′) |
| --- | --- |
| RPL10 | ACCCCCUGCACACUUACCCAAUCCUUUUAG |
| eIF5A | AUCUCUUGGCUAUCCCUCUUGCUUCUCCAG |
| TRAF4 | ACUCCUGCCUCUCUACUUCUGUGGCCCCAG |

Reduced Serum Medium, transferred to cuvettes and mixed with 10 nmol of morpholinos. Samples were electroporated using Gene Pulser electroporation system (Bio-Rad). Electroporated cells were transferred into 12-well plates containing 500 ul of the fresh media and incubated at 37 °C for 24 h unless otherwise indicated. Antisense morpholinos (hnRNPC2 MO: 5′-GGTGTTCTGTTACT-GACCCGTACAT-3′, eIF5A_IR MO: 5′-AGGGAGGCACCATAC-CAGGATCTCT-3′ and standard control; Table 1) and Endo-Porter transfection reagent were purchased from Gene Tools. For the siRNA transfection, either siRNAs (20 µM/mL) against PKCθ or a pool of 4 individual siRNAs against the respective target gene (Table 2) were transfected in Jurkat cells using Nucleofector II from Amaxa Biosystems, and at 48 h post-transfection, cells were stimulated with PMA for different times.

## Generation of CRISPR/Cas9-modified cell lines

For CRISPR/Cas9-mediated deletion of the alternative 5′ splice site required for the generation of hnRNPC2 (without effecting the upstream hnRNPC1) in HEK293 and HeLa cells, sgRNA candidates were designed in silico using the Benchling tool (Ran et al, 2013). A pair of oligonucleotides for the highest ranked candidate sgRNA cutting between the alternative 5′ splice sites in exon 4 and in the downstream intron (Fig. EV3B) were synthesized and subcloned into pSpCas9(BB)-2A-GFP (px458) plasmid (Addgene, USA). sgRNA sequences are #1: 5′-CTCTACTCAGGTCCGGAAC-3′; and #2: 5′-CTGCATTGTGTCCATCAGT-3′. Cells were cotransfected with a pair of sgRNAs in six-well plates using ROTIFect (Roth) according to the manufacturer's instructions. Forty-eight hours after transfection, cells were selected with 1 µg/ml puromycin and clonal cell lines were isolated by dilution. RNA was extracted as described below and radioactive, splicing sensitive PCR was performed using gene-specific primers (forward primer (FP): 5′-TAGCAGGAGAGGATGGCAGA-3′ and reverse primer (RP): 5′-TGGACTATGACTTTCAACGGGA-3′ Table 3) to confirm the removal of the alternative hnRNPC2 specific 5′ splice site in exon 4 on RNA level. In potentially positive clones, the hnRNPC2 knockout was additionally confirmed by western blot.

## RNA extraction, reverse transcription, RT-PCR, RT-qPCR

RNA isolation, reverse transcription and radioactive, splicing-sensitive RT-PCRs were done as described previously (Preußner et al, 2014). For the RNA fractionation analysis, chromatin-associated RNA and cytoplasmic RNA were isolated as described previously (Los et al, 2022). All primers are listed in Table 3. Briefly, total and nascent RNA were isolated using RNATri (Bio&Sell) followed by DNAse I digestion to remove DNA contamination. In all, 1 µg of RNA was used for a gene-specific RT reaction. For alternative splicing quantification, a low-cycle PCR using a $^{32}$P-labeled forward primer was performed and PCR

products were separated by denaturing PAGE and quantified using a Phosphoimager (Typhoon 9200, GE Healthcare) and Image-QuantTL software. For RT-qPCR analyses, up to 4 gene-specific primers were combined in one RT reaction and qPCRs were performed using Absolute QPCR SYBR Green Mix (Biozym) in a 96-well format on a Stratagene Mx3000P instrument. The reactions were performed in at least three biological replicates, and mean values were normalized to the expression of a housekeeping gene (human hHPRT; mouse mHPRT) (ΔCT) and Δ(ΔCT) were calculated for different conditions. *P* values were calculated using Student's unpaired *t* test.

## Western blot

Whole-cell extracts were prepared in RIPA buffer (5-mM Tris [pH 8.0], 150 mM sodium chloride, 1% IGEPEL, 0.5% sodium deoxycholate, 0.1% sodium dodecyl sulfate) supplemented with protease inhibitor mix (Aprotinin, Leupeptin, Vanadate, and PMSF). Concentrations were determined using Roti Nanoquant (Serva), according to the manufacturer's instructions. FastAP Thermosensitive Alkaline Phosphatase (1 U/µL, Thermo Scientific) was used to allow dephosphorylation of hnRNPC2; lysates were incubated for 1 h in 37 °C. SDS-PAGE and western blotting were done using standard protocols. The primary antibodies used for western blotting were as follows: hnRNP C1/C2 antibody (4F4: sc-32308, Santa Cruz Biotechnology, 1:1000), p-ERK antibody (E-4): sc-7383 Santa Cruz Biotechnology, 1:1000), ERK 1/2 antibody (C-9): sc-514302, Santa Cruz Biotechnology, 1:1000), DYKDDDDK Tag antibody (Cell Signaling Technology, 1:2000), GAPDH antibody (6C5): sc-32233; 1:2000), hnRNP-L antibody: sc-32317, Santa Cruz Biotechnology, 1:2000). Primary Antibodies were detected with the secondary antibody linked to horseradish peroxidase (anti-mouse IgG HRP, 1:5000, #7076 s, Cell Signaling Technology; anti-rabbit IgG HRP, 1:5000, #7074 s, Cell Signaling Technology). Western blots were quantified using the ImageQuant TL software, shown is the percentage of phosphorylated hnRNPC2 over total hnRNC2, using the following equation 100*(C2P/(C2P + C2)).

## Western blot SUnSET (WB-SUnSET)

For measuring protein synthesis by WB-SUnSET, HEK293 and Jurkat cells were treated with 10 µg/ml puromycin (P8833, Sigma) for 10 min before harvesting. To confirm that WB-SUnSET was suitable to detect changes in protein synthesis, control cells were treated with cycloheximide (Sigma) at a final concentration (2 µg/ml) for 1 h before the puromycin treatment. The lysates were then normalized for equal amounts of protein using the Bradford method and western blot was performed using standard protocols. The membranes were then incubated with anti-puromycin antibody, followed by secondary antibody linked to horseradish peroxidase (anti-mouse IgG HRP, 1:5000, #7076 s, Cell Signaling Technology) and detected as above. The results were quantified using the GE ImageQuant TL 8.1 software.

## $^{35}$S-Met Incorporation

Jurkat cells were seeded at $1 \times 10^5$/well in 12-well plates and cultured overnight prior to transfection with a pool of 4 siRNA against

hnRNPC and control as described above. At 1-day post transfection, cells were washed twice with methionine-free DMEM supplemented with 10% dialysed FBS (labeling medium) and stimulated by PMA for different times (from 0 min to 24 h). During PMA stimulation, 1 µl of [35]S-Met (> 1000 Ci (37.0 TBq)/mmol; Hartmann Analytic) was added to each well for 30 min before harvesting. Labeled cells were washed twice with PBS, and protein lysates were prepared as described above. Equal amounts of protein were separated on 15% SDS-PAGE. The gel was then stained with Coomassie blue, dried, autoradiographed and quantified using a Phosphoimager (Typhoon 9200, GE Healthcare) and ImageQuantTL software.

## Molecular cloning

The PKCθ overexpression construct was obtained from addgene (Plasmid #8426)[41]. Cloning was performed with the use of NheI and XhoI restriction sites introduced through PCR primers. PCR products were digested and ligated directly into a pCMV-N3-FLAG expression vector to yield an N-terminally FLAG-tagged protein. The construct was identified by western blot and verified by sequencing.

EIF5A_WT and IR codon optimized-inserts were obtained from Twist Bioscience in the pTWIST CMV expression vectors containing a FLAG-tag at the N-terminal end.

To obtain a non-phosphorylatable hnRNPC2 mutant, all potential phosphorylation sites in the hnRNPC2-specific amino acids and surrounding residues in hnRNPC1 were substituted to alanine (Fig. EV3E). Furthermore, we generated a single mutation in Ser115 of hnRNPC2 to obtain a phosphomimetic version of hnRNPC2 (S115D). hnRNPC2 substitution mutants were generated by PCR with a WT hnRNPC2 expression plasmid as a template and verified by sequencing.

## Phosphoprotomic analysis

### Sample preparation and phosphopeptides enrichment

Jurkat pellets were lysed (0.1 mM Tris-HCl [pH 7.6], and 4% SDS), sonicated in a bioruptor (4 °C for 15 min) and boiled at 95 °C for 7 min. A total of 1 mg protein lysate was treated first with 1 µl DTT (1 M) followed by 10 µl 2-Chloroacetamide (0.5 M); both treatments were performed at room temperature (RT, 22 °C) for 20 min. The lysates were precipitated with acetone and phosphopeptides enriched using the EasyPhos method as described (Humphrey et al, 2015). In detail, pellets were resuspended in 500 µl TFE digestion buffer following the addition of digestion enzymes (trypsin and LysC) 1:100 (protein:enzyme). After overnight incubation (37 °C, 1500 rpm) peptide samples were diluted with a buffer containing 150 µl 3.2 M KCl, 55 µl of 150 mM $KH_2PO_4$, 800 µl 100% acetonitrile (ACN), and 95 µl 100% trifluoroacetic acid (TFA) and incubated at RT for 5 min at 1600 rpm prior to centrifugation. The peptide supernatant was moved to a clean 2 ml tube, $TiO_2$ beads subsequently added at a ratio of 10:1 beads/protein, and incubated at 40 °C for 5 min at 2000 rpm. Beads with bound phosphopeptides were pelleted by centrifugation for 1 min at $3500 \times g$ and the supernatant was discarded. Pelleted beads were washed with a buffer containing 60% ACN and 1% TFA and transferred to a clean 2 ml tube, followed by a further four washes with 1 ml of the same buffer. After the last wash beads were resuspended in transfer buffer (80% ACN and 0.5 acetic acid) and added on top of C8 StageTips. After centrifugation phosphopeptides were eluted with

60 µl elution buffer (40% ACN and 15% NH4OH [25%, HPLC grade]) and collected in clean PCR tubes, concentrated in a SpeedVac for 15 min at 45 °C and acidified with 10 µl of 10% TFA. Peptides were then desalted using stageTips with two layers of styrenedivinylbenzene-reversed phase sulfonated (SDB-RPS; 3 M Empore), washed twice with wash buffer (0.2% TFA) and one with isopropanol containing 1% TFA. Peptides were eluted by adding 60 µl SDB-RPS elution buffer (80% ACN, 1.25% $NH_4OH$ (25% HPLC grade)) and immediately concentrated in a SpeedVac for 30 min at 45 °C, after that peptides were then resuspended in a buffer containing 2% ACN and 0.1% TFA prior to chromatography-tandem mass spectrometry (LC-MS/MS) analysis.

### LC-MS/MS analysis and data processing

Phosphopeptides were loaded onto a 50 cm reversed-phase column (diameter 75 µM; packed in-house with 1.9 µM C18 ReproSil particles [Dr. Maisch GmbH]). The temperature of the homemade column oven was maintained at 60 °C. The column was mounted to the EASY-nLC 1000 system (Thermo Fisher Scientific) and the peptides were eluted with a binary buffer system consisting of buffer A (0.1% formic acid) and buffer B (80% ACN and 0.1% formic acid) over a gradient of 140 min (5%–65% buffer B for 130 min followed by 10 min 80% buffer B) with a flow rate of 300 nl/min. Peptides were analyzed in a Q Exactive HF-X mass spectrometer (MS) (Thermo Fisher Scientific) coupled to the nLC, obtaining full scans (300–1600 $m/z$, $R = 60,000$ at 200 $m/z$) at a target of 3e6 ions, the 10 most abundant ions were selected and fragmented with higher-energy collisional dissociation (HCD) (target 1e5 ions, maximum injection time 120 ms, isolation window 1.6 $m/z$, normalized collision energy 25% underfill ratio 40%) followed by the detection in the Orbitrap ($R = 15,000$ at 200 $m/z$). Raw MS data files were processed using MaxQuant (version 1.5.5.6) with the Andromeda search engine with FDR < 0.01 at protein, peptide, and modification level with default settings (variable modifications: methionine (M), acetylation (protein N-term), as well as phospho (STY) and the fixed modification carbamidomethyl (C)); peptides minimal length of seven amino acids; and "match between run" of 0.7 min time window. For protein and peptide identification the UniProt database from human (September 2014) including 51,210 entries was used. Each raw file was treated as one experiment and replicates were combined in the same fraction (no numerical ordering).

### Bioinformatic and statistical analyses

Processed data was uploaded in the Perseus software for further bioinformatical analyses (Tyanova et al, 2016). Reverse sequences and potential contaminants were removed and phosphopeptide intensities were log2 transformed and filtered for at least one valid value. After expanding the site table, the data was again filtered for one valid value. Two sample $t$ test analyses were compared to identify statistically significant (FDR < 0.05) phosphopeptides between control and treated conditions for each time point.

### hnRNPC2 protein purification

The hnRNPC2 full-length construct was designed codon-optimized with an N-terminal Twin-Strep-tag® and SUMO tag for recombinant production from GeneArt custom gene synthesis service (Thermo Fisher Scientific) and cloned into pETM-11 vector using primers shown in Table 3. hnRNPC2 WT and S115D mutant were expressed in *E. coli* BL21 RIL cells in terrific broth media. Following normal growth,

cells were induced at an $OD_{600}$ nm of 0.8–1.0 with 0.4 mM IPTG followed by protein expression for 16 h at 18 °C. Cells were collected by centrifugation, lysed by sonication in the presence of DNase, lysozyme, and EDTA-free complete protease inhibitor (Roche Applied Science), then resuspended in binding buffer consisting of 50 mM Tris (pH 8.0), 500 mM NaCl, and 2 mM β-mercaptoethanol (BME). The cleared lysate was passed over a StrepTrap™ 5 mL column (GE Healthcare). Elution with 50 mM Tris (pH 8.0), 500 mM NaCl, 2.5 mM desthiobiotin, and 2 mM BME was followed by tag (TwinStrep-SUMO) removal with PreScission protease (1:20) overnight. The eluate was concentrated using Amicon Ultra (10 kDa), Millipore centrifugal filter unit and passed over Superdex 200 16/60 size-exclusion column (GE Healthcare) in a buffer containing 50 mM Tris-HCl pH 8.0, 200 mM NaCl, and 2 mM DTT. The peak fractions were pooled, concentrated, and aliquots were flash-frozen, and stored at −70 °C.

### EMSA

RNA electrophoretic mobility shift assays were carried out in binding buffer (20 mM Tris-HCl pH 8.0, 0.25 M NaCl, 1 mM DTT, 10% glycerol and 0.2 mM EDTA). RNA probes containing parts of regulated introns, including the splice site AG (30 nucleotides), were ordered from Eurofins Scientific. (RPL10: 5'-ACCCCCUGCACACUUACC-CAAUCCUUUUAG-3', eIF5A: 5'-AUCUCUUGGCUAUCCCU-CUUGCUUCUCCAG-3' and TRAF4: 5'- ACUCCUGCCUCUCUAC UUCUGUGGCCCCAG-3', Eurofins Scientific, Table 4). Furthermore, a 45-nucleotide poly-U was used as a positive control (Thermo Fisher Scientific). 5 nmol of each RNA was $^{32}$P-labeled at the 5' end using PNK (T4 PNK (10 U/μL, Thermo Fisher Scientific, USA) and subsequently diluted. Excess [γ-32P]-ATP was separated from the labeled RNA by purification on a G25 spin column (GE Healthcare). An amount of 10 pmol of radiolabeled RNAs was incubated with either hnRNPC2 FL or S115D mutant at the following concentrations: 0 μM, 2 μM, 5 μM, 10 μM (unless otherwise indicated) at 37 °C for 20 min. Samples were separated on a 5% polyacrylamide gel and visualized using autoradiography. To determine the level of hnRNPC2 FL and S115D mutant binding to introns of interest, intensities of binding were quantified relative to free RNA (mean ± SD, number of biological replicates: $n = 3$).

### Statistical analysis

Data represent mean values of at least 3 biological replicates (exact numbers are given in the figure legends), and error bars represent standard deviation (SD). Statistical significance was determined using GraphPad Prism version 10.3.03, calculated by Student's unpaired $t$ test, and accepted at $P \leq 0.05$. In the figure legends, "ns" indicates $P \geq 0.05$—non-significant, * indicates $P \leq 0.05$, ** indicates $P \leq 0.01$, *** indicates $P \leq 0.001$ and **** indicates $P \leq 0.0001$.

## Data availability

RNA-Seq data are available as GSE271051. The source data are available online for this paper. Any additional information required to reanalyze the data reported in this paper is available from the corresponding author upon request.

The source data of this paper are collected in the following database record: biostudies:S-SCDT-10_1038-S44318-025-00374-8.

## Peer review information

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

## Acknowledgements

We thank members of the Heyd and Wahl labs for critical comments and interesting discussions. High-performance computing for analysis of RNA-Seq data was provided by FUB-IT. This work was supported by the DFG, through the TRR186, project A15 to MCW and FH. Further support came from the LMU Munich's Institutional Strategy LMUexcellent within the framework of the German Excellence Initiative and DFG INST 86/1800-1 FUGG (to MSR).

## Author contributions

**Mateusz Dróżdż**: Formal analysis; Validation; Investigation; Methodology; Writing—original draft; Writing—review and editing. **Luíza Zuvanov**: Software; Formal analysis; Investigation. **Gopika Sasikumar**: Formal analysis; Investigation. **Debojit Bose**: Formal analysis; Investigation. **Franziska Bruening**: Formal analysis; Investigation. **Maria S Robles**: Formal analysis; Supervision; Investigation. **Marco Preußner**: Conceptualization; Data curation; Software; Formal analysis; Supervision; Investigation; Visualization; Methodology; Writing—review and editing. **Markus Wahl**: Conceptualization; Formal analysis; Supervision; Funding acquisition; Visualization; Methodology; Project administration. **Florian Heyd**: Conceptualization; Data curation; Formal analysis; Supervision; Funding acquisition; Methodology; Writing—original draft; Project administration; Writing—review and editing.

Source data underlying figure panels in this paper may have individual authorship assigned. Where available, figure panel/source data authorship is listed in the following database record: biostudies:S-SCDT-10_1038-S44318-025-00374-8.

## Funding

## Disclosure and competing interests statement

The authors declare no competing interests.

# Expanded View Figures

**Figure EV1.   IES during T cell activation is independent of de novo protein synthesis.**

(**A**) Jurkat cells were PMA-stimulated for the indicated times. Cells were harvested, total protein was extracted, and western blot was performed. A representative graph shows the activation of ERK1/2 upon PMA stimulation. (**B**) Jurkat cells were stimulated as in A. Cells were harvested and total RNA was extracted. Gene expression was analyzed by RT-qPCR (number of biological replicates: n = 3, mean ± SD). (**C**) rMATS analysis identifying significant changes (see methods) in alternative splicing of the types skipped exon (SE), retained introns (RI) alternative 5′ splice site (A5SS) or 3′ splice site (A3SS). Pairwise comparisons for 0 vs 30, 30 vs 150 and 0 vs 150 min are shown (from left to right). Increased RI events were observed when non-stimulated cells (0 min) were compared with PMA-stimulated cells for 30 min. (**D**) Box-whisker plots comparing intron length (left), MaxEntScan 5′ss (middle) and 3′ss (right) for introns retained after 30 min (IR_30) and introns more efficiently spliced (spliced_30) with all introns quantified by rMATS (all introns). Statistical significance was analyzed by Student's unpaired $t$ tests and is indicated by asterisks (ns: non-significant, ***$P < 0.001$). Line represents median, box covers the interquartile interval and whiskers min to max. (**E**) Frequency distribution of IR changes. The maximal IR change (0 vs 30 or 30 vs 150 min) was calculated for all 173 significant introns. The histogram summarizes the frequency of IR changes in bin's differing by 5% (the number on the x-axis represents the center of each Bin). (**F**) Protein coding potential of IR isoforms. Our analysis shows that 169 of the 173 IR events occur in protein-coding genes, with 140 of these situated within the open reading frame (ORF). Of the 140 introns within the ORF, 136 introduce stop codons, either through frameshifts (93 cases) or by directly introducing a stop codon (43 cases). In 10 of these cases, the stop codon occurs in the last intron of the ORF, potentially producing an alternative C-terminus. In the remaining 126 cases, the introns would encode premature termination codons (PTCs). This leaves 4 cases that maintain in-frame sequences, including the translation initiation factor eIF5A. The other three genes are IRF3 (Interferon Regulatory Factor 3), MFSD10 (Major Facilitator Superfamily Domain-containing Protein 10), and TNFRSF1A (Tumor Necrosis Factor Receptor Superfamily Member 1A). Notably, IRF3, a key regulator of interferon alpha/beta transcription and other interferon-induced genes and has an alternative intron close to its DNA-binding domain, which might affect its transcriptional activity. TNFRSF1A, with an IR isoform, could potentially alter its TNF-alpha binding domain.

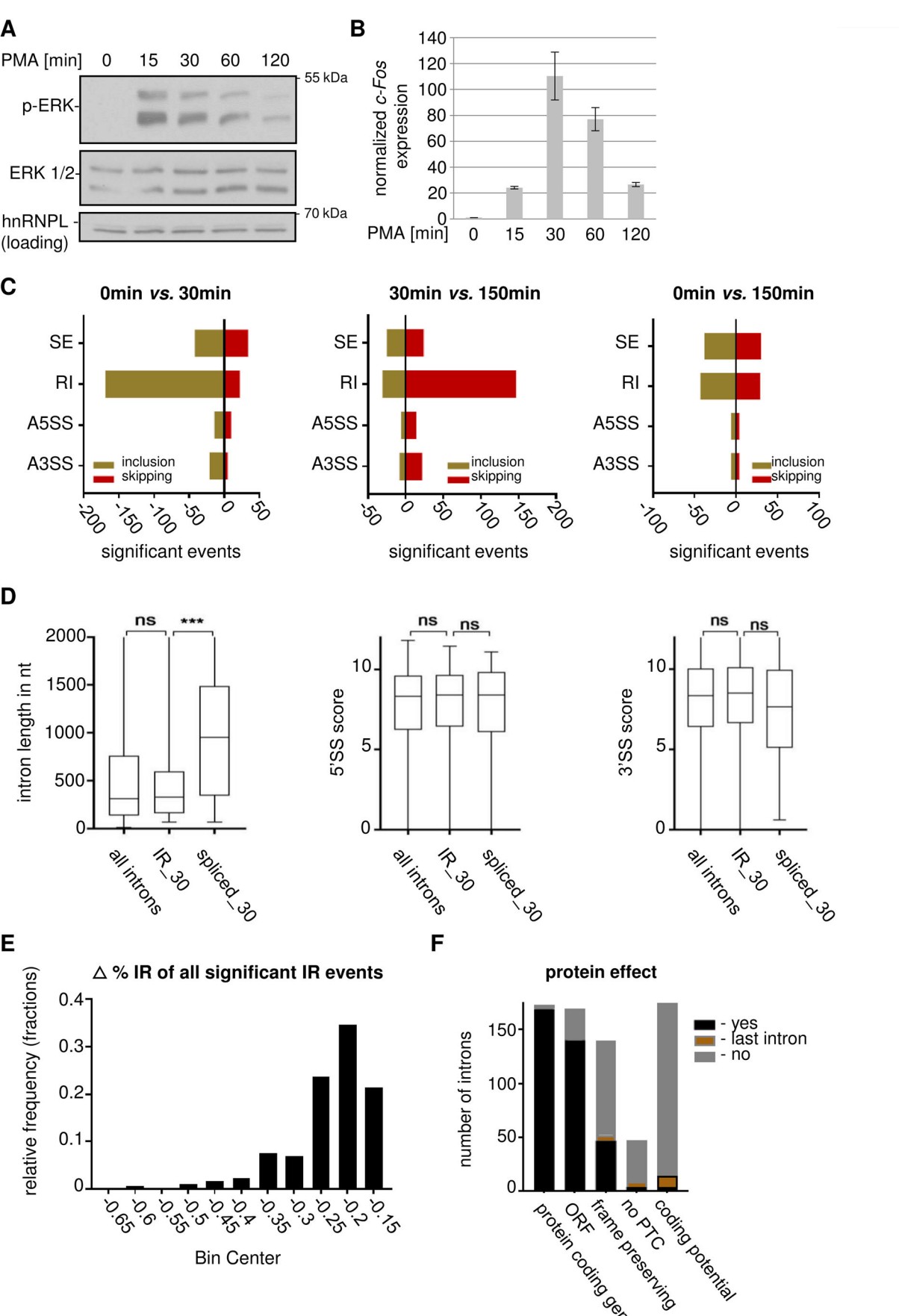

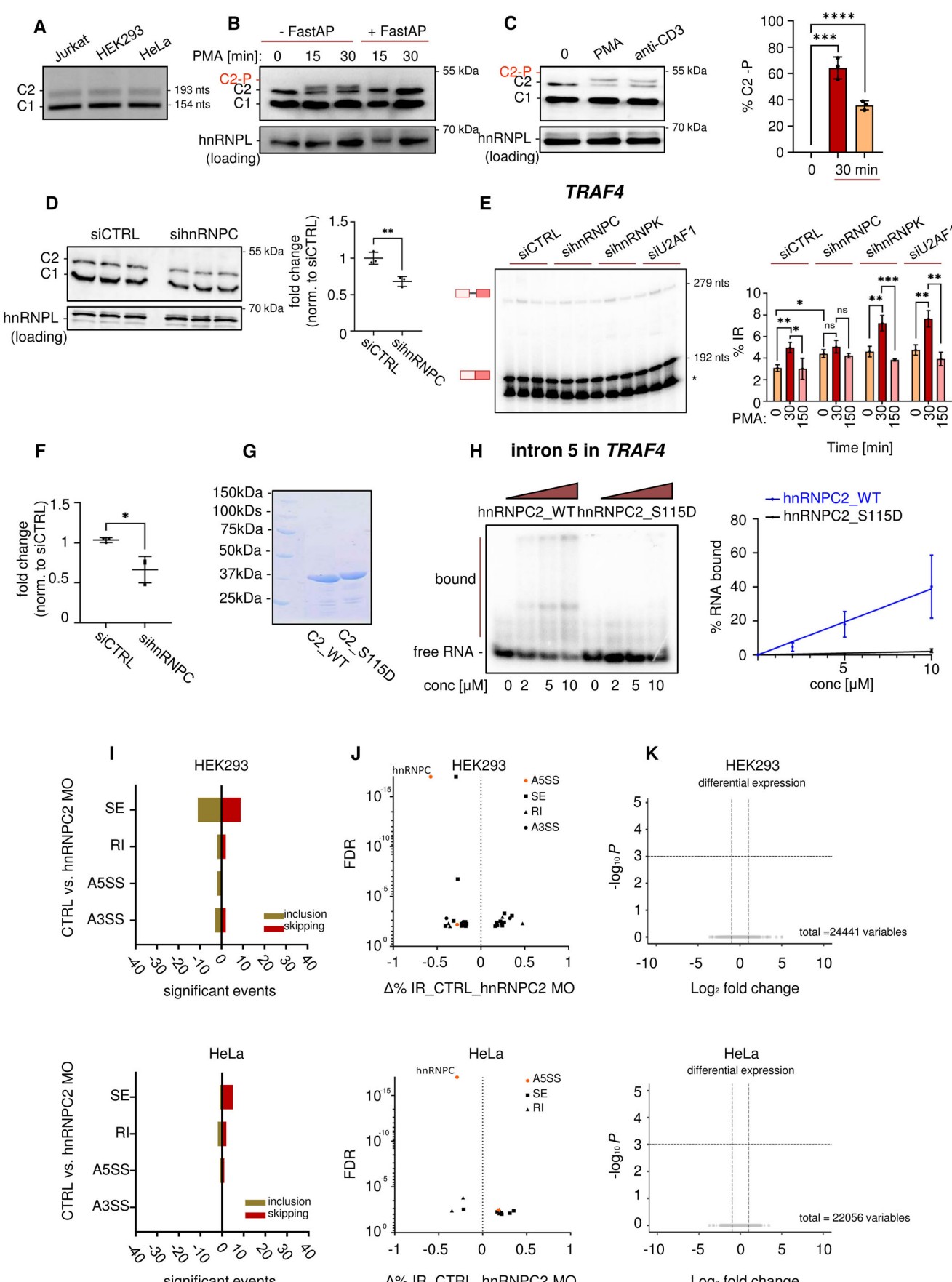

◀ **Figure EV2.   IES is regulated by hnRNPC2.**

(**A**) hnRNPC1/C2 isoform expression was detected in Jurkat cells, HEK293, and HeLa cells by standard splicing sensitive RT-PCR. (**B**) Jurkat cells were stimulated by PMA for the indicated times. FastAP was added to the indicated protein lysate for 1 h at 37 °C and lysates were analyzed by western blot. hnRNPL serves as loading control (number of biological replicates: $n = 3$). (**C**) Jurkat T cells were stimulated for 30 min by PMA and 5 µg anti-CD3 Ab. Left: Western blot presents increased phosphorylation of hnRNPC2 that was detected in PMA and anti-CD3 stimulated cells. hnRNPL serves as a loading control. Right: corresponding quantification (data are presented as % hnRNPC2 phosphorylation of total hnRNPC2, mean ± SD, number of biological replicates: $n = 3$, 0 vs. 30 min (PMA: dark red), $P = 0.0002$ vs. 30 min (anti-CD3: bright yellow), $P < 0.0001$, Student's unpaired $t$ test). (**D**) Jurkat cells were transfected with siRNA against hnRNPC or control. Total protein was extracted. The efficiency of hnRNPC knockdown was analyzed by western blot (left, the gel shows triplicate samples) and quantified (right, data are presented as fold change normalized to siCTRL, mean ± SD, number of biological replicates: $n = 3$, $P = 0.0069$, Student's unpaired $t$ test). hnRNPL serves as loading control. (**E**) IES in *TRAF4* depends on hnRNPC. Jurkat cells were transfected with siRNA against hnRNPC, hnRNPK or U2AF1 and, 48 h later, stimulated with PMA. Left: *TRAF4* IR was analyzed by radioactive, splicing-sensitive RT-PCR and quantified (right, data are presented as % IR, mean ± SD, number of biological replicates: $n = 3$, 0 vs. 30 min (siCTRL), $P = 0.0078$, siCTRL vs. sihnRNPC (0 min), $P = 0,007$, 30 vs. 150 min (siCTRL), $P = 0.0074$, 0 vs. 30 min (sihnRNPC), $P = 0.558$, 30 vs. 150 min (sihnRNPC), $P = 0.654$, 0 vs. 30 min (sihnRNPK), $P = 0.008$, 30 vs. 150 min (sihnRNPK), $P = 0.0003$, 0 vs. 30 min (U2AF1), $P = 0.0068$, 30 vs. 150 min (U2AF1), $P = 0.0059$, Student's unpaired $t$ test). (**F**) HEK293 cells were transfected with siRNA against hnRNPC and control. After 48 h, cells were harvested, and RNA was extracted. The efficiency of hnRNPC knockdown was determined by RT-qPCR. mRNA expression is relative to hHPRT (data are presented as fold change normalized to siCTRL, mean ± SD, number of biological replicates: $n = 3$, $P = 0.0175$, Student's unpaired $t$ test). (**G**) Coomassie-stained SDS-gel showing the amount of purified hnRNPC2 proteins used for EMSAs. (**H**) Increasing amounts of either hnRNPC_WT or S115D (0 µM, 2 µM, 5 µM, 10 µM) were complexed with 10 pmol of radioactively labeled RNA corresponding to a part of the *TRAF4* intron 5 that includes the polypyrimidine tract and the 3' splice site. Left: representative native gel shows a reduction of RNA binding in the S115D phosphomimetic-mutant. Data are representative of at least three independent experiments. Right: corresponding quantification of experiment performed in triplicates. (**I**) HEK293 cells (top) and HeLa cells (bottom) were transfected with an hnRNPC2-inducing morpholino (hnRNPC2 MO), or control MO (CTRL MO). After 48 hours, cells were harvested, total RNA was extracted and RNA-seq was performed. Plots represent rMATs splicing analysis in HEK293 and HeLa cells after hnRNPC2 MO treatment. Only significant events are shown. Note that only very few splicing events are affected. (**J**) Volcano plots from HEK293 (top) and HeLa (bottom) cells show that hnRNPC alternative splicing is the most (and almost only) highly significant splicing event detected after MO treatment. This shows high efficiency of the MO, and basically no effect of shifting the hnRNPC1:C2 ratio on other splicing events. (**K**) Volcano plots as in J showing no difference in gene expression between hnRNPC MO samples and CTRL.

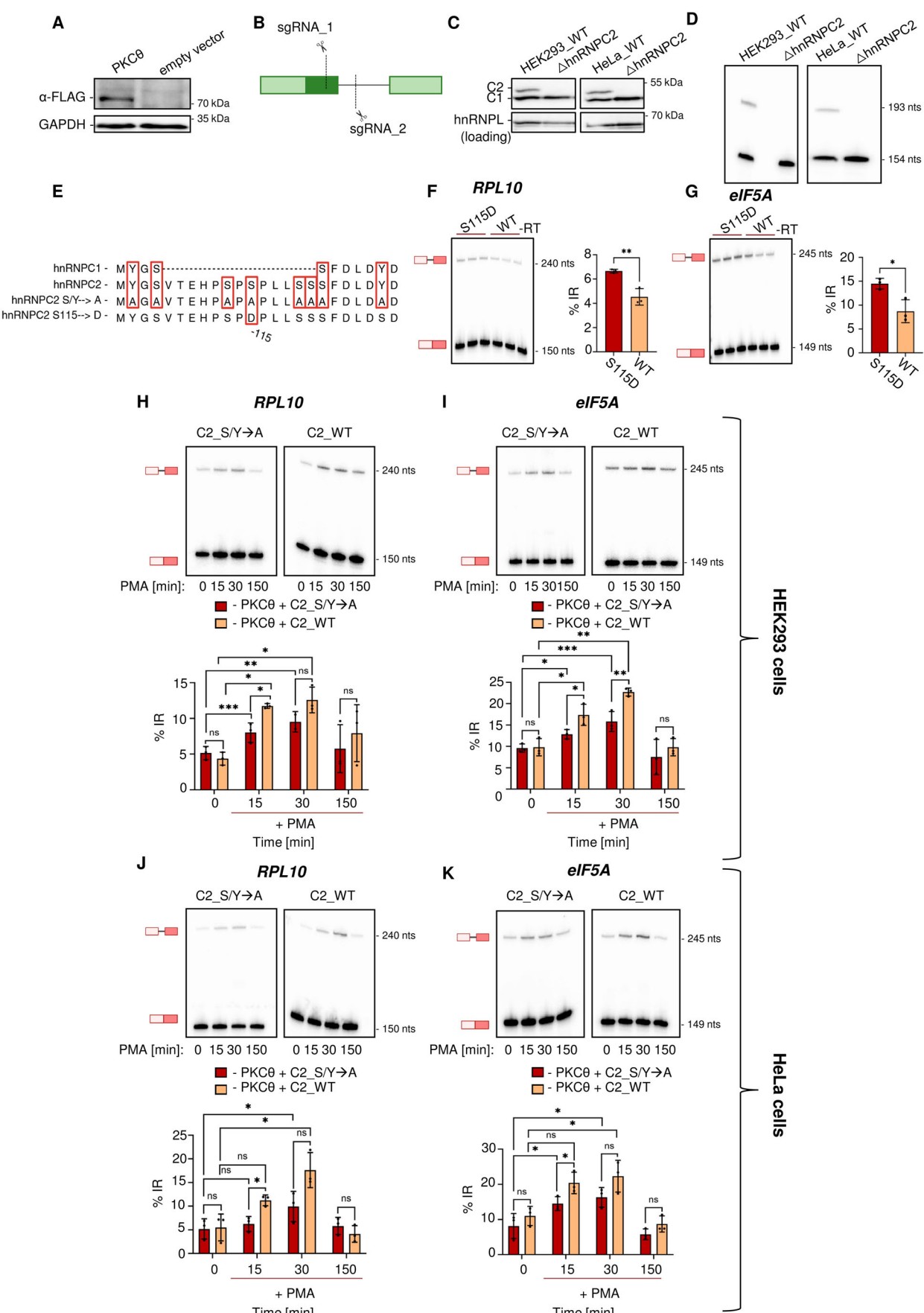

◀ **Figure EV3. PKCθ and hnRNPC2 are required for IES.**

(A) HEK293 cells were transfected with an overexpression vector for PKCθ and empty FLAG vector. After 48 h, total protein was extracted. Western blot confirms PKCθ (anti-FLAG) overexpression with GAPDH as loading control. (B) Schematic view of the deletion of the hnRNPC2-generating 5′ splice site of exon 4 using CRISPR/Cas9. Scissors show the position of the sgRNAs. Created with BioRender.com. (C) HEK293_WT and CRISPR/Cas9-edited HEK293 with a deletion of hnRNPC2 (left) were harvested and total protein was extracted. Western Blot shows the deletion of hnRNPC2. hnRNPL acts as loading control. The same approach was used in HeLa cells shown on the right. (D) Clones of HEK293_WT, HeLa_WT and CRISPR/Cas9-edited HEK293 and HeLa cells with a deletion of hnRNPC2 were harvested and RNA was extracted. Radioactive, splicing-sensitive RT-PCR confirms the absence of hnRNPC2. (E) Alignment of hnRNPC1, hnRNPC2, a nonphosphorylatable version of hnRNPC2 and phosphomimetic version of hnRNPC2 (S115→D). All potential phosphorylation sites in hnRNPC2, including Ser115 and some surrounding residues in hnRNPC1 were mutated. Mutated residues are highlighted in red frames. (F, G) hnRNPC2 WT and its phosphomimetic version S115D were overexpressed in HEK293 cells. After 48 h, cells were harvested, and chromatin-associated RNA was extracted. *RPL10* (F) and *eIF5A* (G) IR were analyzed by radioactive, splicing-sensitive RT-PCR (left) and quantified (right, data are presented as % IR, mean ± SD, number of biological replicates: $n = 3$, *RPL10* (F): $P = 0.006$, *eIF5A* (G): $P = 0.0192$, Student's unpaired *t* test). (H, I) PKCθ-induced IES depends on hnRNPC2 phosphorylation. HEK293 cells were co-transfected with PKCθ and either hnRNPC2_WT or the nonphosphorylatable version. Cells were treated and analyzed as in Fig. EV3F, G (*RPL10* (H) and *eIF5A* (I), data are presented as % IR, mean ± SD, number of biological replicates: $n = 3$, *RPL10* (H): PKCθ + C2_S/Y→A vs. PKCθ + C2_WT (0 min), p = 0.3541, PKCθ + C2_S/Y→A vs. PKCθ + C2_WT (15 min), $P = 0.0102$, PKCθ + C2_S/Y→A vs. PKCθ + C2_WT (30 min), $P = 0.141$, PKCθ + C2_S/Y→A vs. PKCθ + C2_WT (150 min), $P = 0.8609$, 0 vs. 15 min (PKCθ + C2_S/Y), $P = 0.0397$, 0 vs. 30 min (PKCθ + C2_S/Y), $P = 0.023$, 0 vs. 15 min (PKCθ + C2_WT), $P = 0.0002$, 0 vs. 30 min (PKCθ + C2_WT), $P = 0.0021$, *eIF5A* (I): PKCθ + C2_S/Y→A vs. PKCθ + C2_WT (0 min), $P = 0.881$, PKCθ + C2_S/Y→A vs. PKCθ + C2_WT (15 min), $P = 0.0309$, PKCθ + C2_S/Y→A vs. PKCθ + C2_WT (30 min), $P = 0.0016$, PKCθ + C2_S/Y→A vs. PKCθ + C2_WT (150 min), $P = 0.4382$, 0 vs. 15 min (PKCθ + C2_S/Y), $P = 0.0185$, 0 vs. 30 min (PKCθ + C2_S/Y), $P = 0.0079$, 0 vs. 15 min (PKCθ + C2_WT), $P = 0.0208$, 0 vs. 30 min (PKCθ + C2_WT), $P = 0.0006$, Student's unpaired *t* test). (J, K) Experiments as in (H, I) using HeLa cells. Data are presented as % IR, mean ± SD, number of biological replicates: $n = 3$, *RPL10* (J): PKCθ + C2_S/Y→A vs. PKCθ + C2_WT (0 min), $P = 0.8923$, PKCθ + C2_S/Y→A vs. PKCθ + C2_WT (15 min), $P = 0.1341$, PKCθ + C2_S/Y→A vs. PKCθ + C2_WT (30 min), $P = 0.1245$, PKCθ + C2_S/Y→A vs. PKCθ + C2_WT (150 min), $P = 0.3176$, 0 vs. 15 min (PKCθ + C2_S/Y), $P = 0.1341$, 0 vs. 30 min (PKCθ + C2_S/Y), $P = 0.0353$, 0 vs. 15 min (PKCθ + C2_WT), $P = 0.3416$, 0 vs. 30 min (PKCθ + C2_WT), $P = 0.0107$, *eIF5A* (K): PKCθ + C2_S/Y→A vs. PKCθ + C2_WT (0 min), $P = 0.327$, PKCθ + C2_S/Y→A vs. PKCθ + C2_WT (15 min), $P = 0.0488$, PKCθ + C2_S/Y→A vs. PKCθ + C2_WT (30 min), $P = 0.1224$, PKCθ + C2_S/Y→A vs. PKCθ + C2_WT (150 min), $P = 0.1382$, 0 vs. 15 min (PKCθ + C2_S/Y), $P = 0.0526$, 0 vs. 30 min (PKCθ + C2_S/Y), $P = 0.035$, 0 vs. 15 min (PKCθ + C2_WT), $P = 0.0163$, 0 vs. 30 min (PKCθ + C2_WT), $P = 0.0206$, Student's unpaired *t* test).

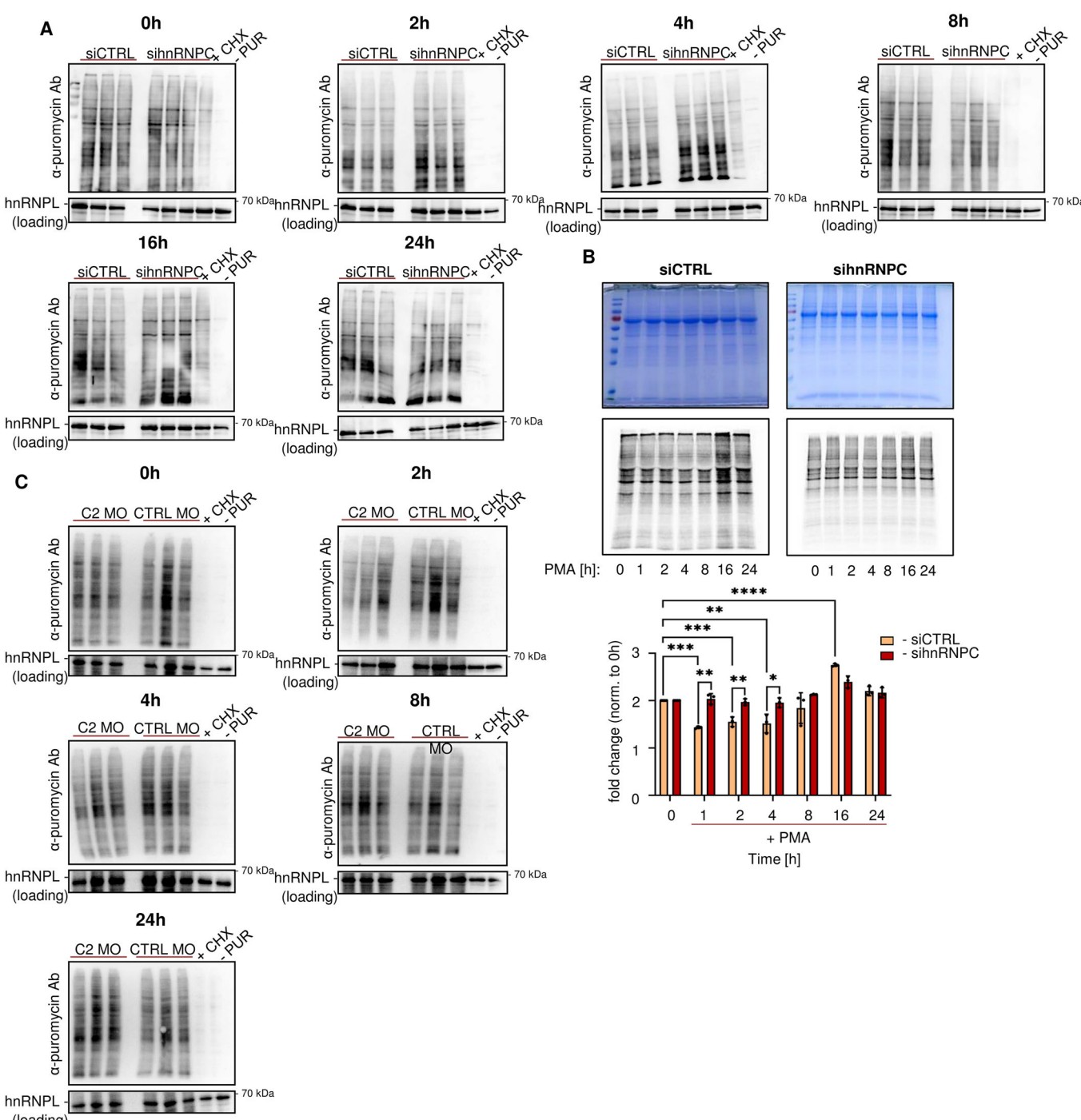

**Figure EV4. IES reduces global translation.**

(A) Triplicate samples for the analysis in Fig. 6A. (B) De novo translation in Jurkat cells after PMA activation in control and hnRNPC knockdown conditions at indicated time points analyzed using $^{35}$S-Met incorporation. Upper gels show commassie-loading control, bottom gels show autoradiographs (representative gels). Bottom: corresponding quantification (data are presented as fold change normalized to 0 h, mean ± SD, number of biological replicates: $n = 3$, 0 vs. 1 h (siCTRL), $P = 0.0005$, 0 vs. 2 h (siCTRL), $P = 0.0004$, 0 vs. 4 h (siCTRL), $P = 0.0035$, 0 vs. 16 h (siCTRL), $P < 0.0001$, sihnRNPC vs. siCTRL (1 h), $P = 0.002$, sihnRNPC vs. siCTRL (2 h), $P = 0.0043$, sihnRNPC vs. siCTRL (4 h), $P = 0.023$ (Student's unpaired $t$ test). (C) Triplicate samples for the analysis in Fig. 6B.

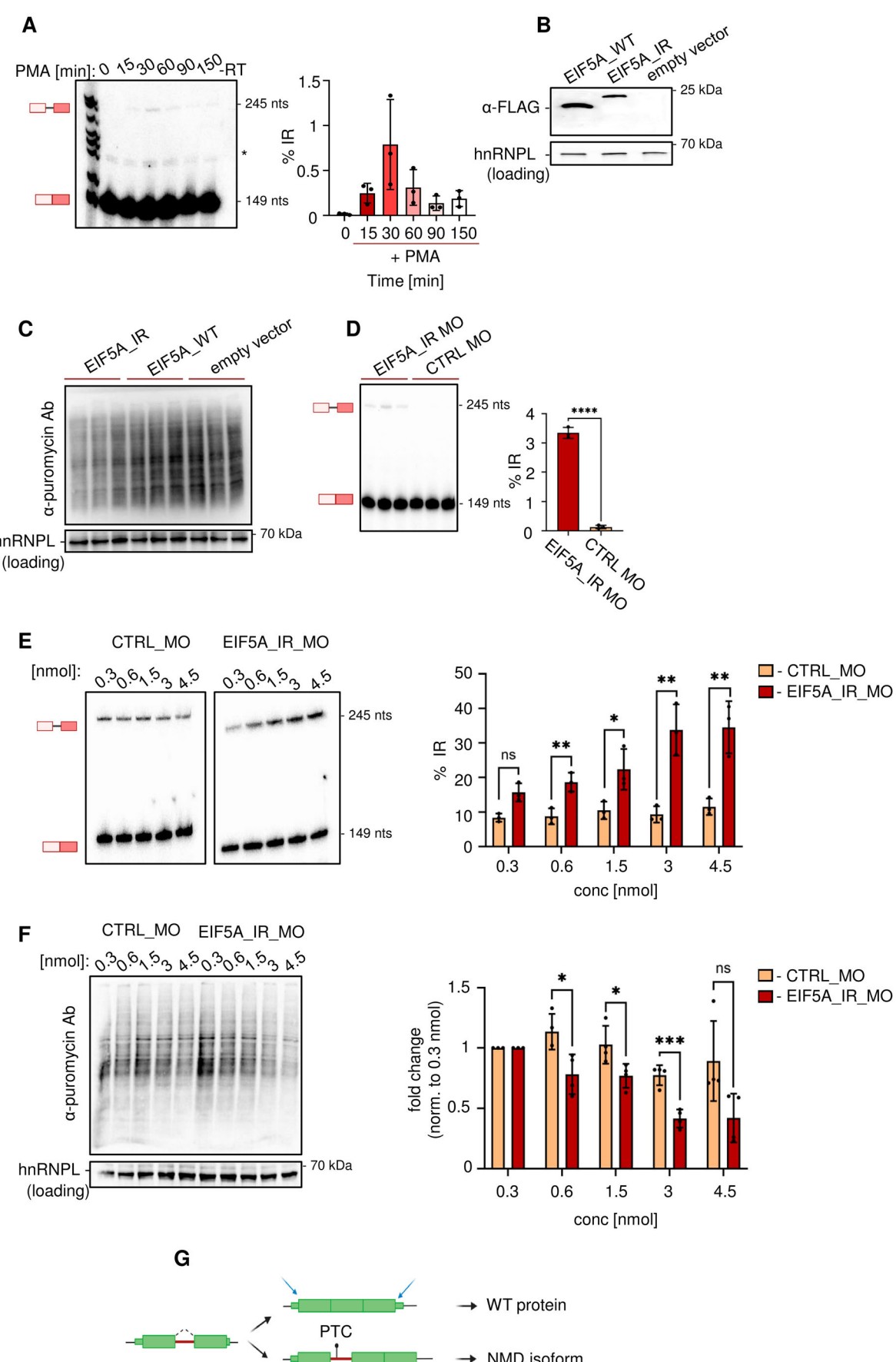

◀ **Figure EV5. The EIF5A IES isoform is sufficient to globally reduce translation.**

(A) The EIF5A IES product is exported to cytoplasm. Jurkat cells were PMA stimulated for the indicated times and cytoplasmic RNA was extracted. Left: representative gel from radioactive, splicing-sensitive PCR showing IR events in cytoplasm after T cell activation. * - degradation product, -RT; without reverse transcriptase. Right: corresponding quantification (data are presented as % IR, number of biological replicates: $n = 3$). (B) HEK293 cells were transfected with eIF5A expression constructs (WT and IR). After 48 h, cells were harvested, and total protein was extracted. Western blot confirms overexpression with hnRNPL as loading control. (C) Triplicate samples for experiments in Fig. 6D. (D) Jurkat cells were electroporated with either eIF5A_IR MO to induce intron retention or control MO for 48 h. Cells were harvested and cytoplasmic RNA was extracted. Left: gel from radioactive, splicing-sensitive PCR showing IR events in cytoplasm after T cell activation. Gel shows triplicate samples. Right: corresponding quantification (data are presented as % IR, mean ± SD, number of biological replicates: $n = 3$, $P < 0.0001$, Student's unpaired $t$ test). (E) Titration experiment with increasing concentrations of EIF5A_IR_MO. Jurkat cells were electroporated with increasing concentrations of EIF5A_IR_MO (from 0.3 nmol to 4.5 nmol). Left: representative gel from radioactive, splicing-sensitive PCR; * degradation product. Right: corresponding quantification (data are presented as % IR, mean ± SD, number of biological replicates: $n = 3$, CTRL_MO vs. EIF5A_IR_MO (0.3 nmol), $P = 0.0511$, CTRL_MO vs. EIF5A_IR_MO (0.6 nmol), $P = 0.0089$, CTRL_MO vs. EIF5A_IR_MO (1.5 nmol), $P = 0.0333$, CTRL_MO vs. EIF5A_IR_MO (3 nmol), $P = 0.0056$, CTRL_MO vs. EIF5A_IR_MO (4.5 nmol), $P = 0.0073$ (Student's unpaired $t$ test). (F) Titration experiment with increasing concentrations of EIF5A_IR_MO. Jurkat cells were electroporated with increasing concentrations of EIF5_IR_MO as in (E). After 48 h, cells were treated with 10 μg/ml of puromycin 10 min before harvesting. Total protein was extracted. Left: representative blot of the WB-SUnSET experiment. hnRNPL serves as loading control. Right: corresponding quantification (data are presented as fold change normalized to 0.3 nmol, mean ± SD, number of biological replicates: $n = 3$, CTRL_MO vs. EIF5A_IR_MO (0.6 nmol), $P = 0.0187$, CTRL_MO vs. EIF5A_IR_MO (1.5 nmol), $P = 0.0318$, CTRL_MO vs. EIF5A_IR_MO (3 nmol), $P = 0.0007$, CTRL_MO vs. EIF5A_IR_MO (4.5 nmol), $P = 0.0511$ (Student's unpaired $t$ test). (G) Schematic representation of the pre-mRNAs of *RPL10*. Due to the presence of a PTC, IR likely induces NMD. Green boxes represent exons, dashed lines show introns, blue arrows show start and stop codon. Created with BioRender.com.

