## [Peer Review File · The EMBO Journal]

Immediate early splicing controls translation in activated T-cells and is mediated by hnRNP2 phosphorylation

Mateusz Drozd, Luiza Zuvanov, Gopika Sasikumar, Debojit Bose, Franziska Bruening, Maria Robles, Marco Preußner, Markus Wahl, and Florian Heyd

Corresponding author(s): Florian Heyd (florian.heyd@fu-berlin.de)

Review Timeline:

Submission Date:	23rd Jul 24
Editorial Decision:	13th Sep 24
Revision Received:	22nd Nov 24
Editorial Decision:	2nd Jan 25
Revision Received:	13th Jan 25
Accepted:	20th Jan 25

Editor: William Teale

Transaction Report:

Dear Dr. Heyd,

Thank you again for the submission of your manuscript entitled "Immediate early splicing upon hnRNPC2 phosphorylation controls translation after T cell activation" and for your patience during the review process; in this regard, please accept my apologies for the usually long time this has taken. We have now received the reports from the referees, which I copy below.

As you can see from their comments, while the referees point to some specific areas in which the manuscript should be strengthened, all of them point to the significance and solid experimental foundation of your work.

The referees' concerns will require your attention before your manuscript can be published in The EMBO Journal; however, based on the overall interest expressed in the reports, I would like to invite you to address the comments of all referees in a revised version of the manuscript. I should add that it is The EMBO Journal policy to allow only a single major round of revision and that it is therefore important to resolve the main concerns at this stage. I believe the concerns of the referees are reasonable and addressable, but please contact me if you have any questions, need further input on the referee comments or if you anticipate any problems in addressing any of their points. I am available to discuss the referee reports with you over Zoom at any time; please do not hesitate to get in touch if you would like to go through them together.

Please, follow the instructions below when preparing your manuscript for resubmission.

I would also like to point out that as a matter of policy, competing manuscripts published during this period will not be taken into consideration in our assessment of the novelty presented by your study ("scooping" protection). We have extended this 'scooping protection policy' beyond the usual 3 month revision timeline to cover the period required for a full revision to address the essential experimental issues. Please contact me if you see a paper with related content published elsewhere to discuss the appropriate course of action.

Again, please contact me at any time during revision if you need any help or have further questions.

Thank you very much again for the opportunity to consider your work for publication. I look forward to your revision.

Best regards,

William

William Teale, Ph.D.
Editor
The EMBO Journal

When submitting your revised manuscript, please carefully review the instructions below and include the following items:

- 1) a .docx formatted version of the manuscript text (including legends for main figures, EV figures and tables). Please make sure that the changes are highlighted to be clearly visible.
- 2) individual production quality figure files as .eps, .tif, .jpg (one file per figure).
- 3) a .docx formatted letter INCLUDING the reviewers' reports and your detailed point-by-point response to their comments. As part of the EMBO Press transparent editorial process, the point-by-point response is part of the Review Process File (RPF), which will be published alongside your paper.
- 4) a complete author checklist, which you can download from our author guidelines ([https://wol-prod-cdn.literatumonline.com/pb-assets/embo-site/Author Checklist%20-%20EMBO%20J-1561436015657.xlsx](https://wol-prod-cdn.literatumonline.com/pb-assets/embo-site/Author%20Checklist%20-%20EMBO%20J-1561436015657.xlsx)). Please insert information in the checklist that is also reflected in the manuscript. The completed author checklist will also be part of the RPF.
- 5) Please note that all corresponding authors are required to supply an ORCID ID for their name upon submission of a revised manuscript.
- 6) We require a 'Data Availability' section after the Materials and Methods. Before submitting your revision, primary datasets

produced in this study need to be deposited in an appropriate public database, and the accession numbers and database listed under 'Data Availability'. Please remember to provide a reviewer password if the datasets are not yet public (see <https://www.embopress.org/page/journal/14602075/authorguide#datadeposition>). If no data deposition in external databases is needed for this paper, please then state in this section: This study includes no data deposited in external repositories. Note that the Data Availability Section is restricted to new primary data that are part of this study.

Note - All links should resolve to a page where the data can be accessed.

8) For data quantification: please specify the name of the statistical test used to generate error bars and P values, the number (n) of independent experiments (specify technical or biological replicates) underlying each data point and the test used to calculate p-values in each figure legend. The figure legends should contain a basic description of n, P and the test applied. Graphs must include a description of the bars and the error bars (s.d., s.e.m.).

9) We would also encourage you to include the source data for figure panels that show essential data. Numerical data can be provided as individual .xls or .csv files (including a tab describing the data). For 'blots' or microscopy, uncropped images should be submitted (using a zip archive or a single pdf per main figure if multiple images need to be supplied for one panel). Additional information on source data and instruction on how to label the files are available at .

10) We replaced Supplementary Information with Expanded View (EV) Figures and Tables that are collapsible/expandable online (see examples in <https://www.embopress.org/doi/10.15252/embj.201695874>). A maximum of 5 EV Figures can be typeset. EV Figures should be cited as 'Figure EV1, Figure EV2" etc. in the text and their respective legends should be included in the main text after the legends of regular figures.

12) Our journal encourages inclusion of *data citations in the reference list* to directly cite datasets that were re-used and obtained from public databases. Data citations in the article text are distinct from normal bibliographical citations and should directly link to the database records from which the data can be accessed. In the main text, data citations are formatted as follows: "Data ref: Smith et al, 2001" or "Data ref: NCBI Sequence Read Archive PRJNA342805, 2017". In the Reference list, data citations must be labeled with "[DATASET]". A data reference must provide the database name, accession number/identifiers and a resolvable link to the landing page from which the data can be accessed at the end of the reference. Further instructions are available at .

13) In order to increase the reproducibility and reach of your work, The EMBO Journal includes a table of reagents that were used in the study. Please provide this along with your revisions.

Further instructions for preparing your revised manuscript:

When assembling figures, please refer to our figure preparation guideline in order to ensure proper formatting and readability in

print as well as on screen:

We realize that it is difficult to revise to a specific deadline. In the interest of protecting the conceptual advance provided by the work, we recommend a revision within 3 months (12th Dec 2024). Please discuss the revision progress ahead of this time with the editor if you require more time to complete the revisions. Use the link below to submit your revision:

Referee #1:

Immediate early splicing upon hnRNPC2 phosphorylation controls translation after T cell activation
By Drózdź et al

Drózdź et al propose a model for immediate early splicing (IES), i.e., phosphorylation-dependent alternative splicing changes in response to T cell stimulation, in analogy to the well-studied transcriptional induction of immediate early genes (IEGs). Using RNA-seq on chromatin-associated RNA, the authors observe the transient regulation of alternative splicing, most prominently intron retention, in PMA-stimulated JURKAT cells and further validate exemplary events in stimulated primary T cells. Using phospho-proteomics, they observe transient phosphorylation of the longer hnRNPC isoform hnRNPC2. Indeed, increasing the relative hnRNPC2 abundance with a splice-switching morpholino leads to higher intron retention. In vitro binding assays with wt and phosphomimetic hnRNPC2 and inhibitor studies indicate that the phosphorylation results in decreased hnRNPC2 binding and is catalyzed by ERK kinase. The authors move on to show that the cell type-specific regulation by hnRNPC is achieved through the PKC θ kinase, which is restricted to T cells and may set a priming phosphorylation on hnRNPC2. Consistently, IES can be recapitulated in HEK293 cells through the ectopic overexpression of PKC θ . Finally, the authors show that IES IR in eIF5A mediates a global reduction in de novo protein synthesis.

The study of alternative splicing programs in immunology is both timely and relevant. The authors provide a detailed mechanistic dissection of the proposed IES response, offering convincing evidence from orthogonal assays to support their conclusions.

Comments:

1) Regarding the observation of increased intron retention upon 30 min of treatment, I wondered whether this could also be due to an increase in transcription. A burst in transcription could lead to an increase in yet unspliced nascent RNA. Have the authors checked this? Also related to this, could it be that splicing is rather slowed down at these introns and that they will eventually still be spliced?

2) Regarding the binding of hnRNPC2 to U tracts (Figure 3G), I think that from the presented data it cannot be concluded that there is no difference in binding between wt and the phosphomimetic mutant since binding is already saturated in the lowest protein concentration used (2 micromolar). The assay should be repeated with lower concentrations of hnRNPH2.

3) In the authors' model, phosphorylation leads to reduced RNA binding of hnRNPH2. Does this loss of binding result in a loss of function or can it also have a gain-of-function effect? For example, how would hnRNPH2 phosphorylation affect the binding of the hnRNPH tetramer? In this context, I wondered whether the authors have tested overexpression of wt and phosphomimetic hnRNPH2 in cells. It will be interesting to show and discuss how splicing is affected in these experiments.

4) The requirement of a priming phosphorylation for IES is not very well supported. Have the authors experimentally approached this? Are there alternative explanations? Does the priming need to be on HNRNPH2 or could it be on an adapter protein mediating the ERK-mediated phosphorylation of Ser115?

Other:

I find the use of "PSI (percent spliced in)" in combination with intron retention confusing. Maybe better to use "isoform frequency" instead?

Figure 2C, F and G: More meaningful labelling of the y-axis for the bar diagrams would be helpful. What was quantified in the bar chart? For instance, in panel G, is this phosphorylated hnRNPC2 over total hnRNPC2 or over total hnRNPC?

This sentence is complicated to read: "For all three investigated target introns IES occurred upon knock down of U2AF1 or hnRNPK, two other candidate trans-acting factors with transient phosphorylation upon T cell activation (U2AF1) or another polyC binding protein (hnRNPK) (Figs. 3A, B, Supp. Fig. 2D), showing specificity for hnRNPC."

The following paper would be good to cite:

Regulation of alternative pre-mRNA splicing by the ERK MAP-kinase pathway
<https://www.ncbi.nlm.nih.gov/pmc/articles/PMC149173/>

Figure 1A: Colors of 0 and 150 min are indistinguishable

Figure 1D, E: If the Y-axis shows "fold change (rel to 0 min)", why are the 0 min values not at 1? The authors should ensure that the quantification and normalization strategies are mentioned in the methods.

Referee #2:

This manuscript reports a fascinating new phenomenon that coordinates splicing programs with T cell activation through a phosphorylation cascade has the potential for broad applicability to cell cycle and related phenomena. Specifically, a specific kinase in model T cells (Jurkat) targets the hnRNPC2, which then induces through alternative splicing a palette of protein isoform changes relevant to translation. This coordinated use of an alternative splicing pathway - called by the authors immediate early splicing (IES) by analogy to the classical term immediate early genes (IEG) - can yield transient intron retention and is an exciting new function for alternative splicing in general. In Jurkat cells, the effect of IES is to reduce translation. In some ways the most important finding in the paper is that the cellular events including global inhibition of translation can be phenocopied by ASO-mediated induction of intron retention in the eIF5A pre-mRNA, and this is basically a sanity check. The strength of the paper is the rigorous testing along the arc of the paper providing clear evidence of eIF5A and RPL10 intron retention at 30 min of PMA or anti-CD3 stimulation, direct effects on RNA binding and this AS event by hnRNPC2 phosphorylation, identification of the kinase, and reliance on hnRNPC2 expression and activity of the kinase in vivo. therefore, the paper generates a new conceptual framework for splicing regulation in the rapid tuning of translation in the context of an overall physiologically relevant pathway through step-by-step rigorous testing of the molecular underpinnings of their hypothesis.

We have made some suggestions mostly on the Figures/Data:

- Figure 1A: It would be good to report the percent variance captured by the two principal components in the PCA plot, either in the legend or along the axis labels (e.g., "PC1 (60% variance)").
- Figure 1B: The reader can infer that the individual columns in the heatmap correspond to individual replicates, but it might be better to directly label them. It could also be beneficial to include a sentence stating that each row in the heatmap corresponds to a different intron retention event.
- Figure 1C: The amount of white space could be reduced in this panel by placing the two legends in the bottom right. The legend also says that the GO terms with "translation" are highlighted in blue. This sentence has an upside-down quotation mark before "translation" and an extra comma after "translation". It would also be more accurate to say "underlined in blue" rather than "highlighted in blue".
- Figure 1D-G: What are the middle bands in 1D,E,G?
- A plot comparing the distribution of all intron sizes and intron sizes for the IR events of interest would be useful to convey the point about short introns made towards the bottom of page 4 (possibly include the plot in the supplement).

- General comments for gels and blots: It would be good to label the sizes of the products / bands. This would make interpretation more straightforward.

Other comments

- Middle of first line on page 3: missing an apostrophe; "increasing the genomes coding capacity" should be "increasing the genome's coding capacity"
- On page 4 towards the bottom, should "as the vast majority of ~6.000 rMATS-quantified introns" be "as the vast majority of ~6,000 rMATS-quantified introns" (i.e., comma instead of period)? Or no punctuation?
- It's good that results are validated in other cell lines (e.g., 1G, 5F-G).

Referee #3:

This interesting manuscript from Drózdź, Heyd and colleagues describes a hitherto unknown gene regulatory mechanism to couple translational downregulation of the existing transcriptome with de novo transcription of Immediate Early Genes (IEG) upon T-cell activation. By carrying out chromatin associated RNA-Seq during T cell activation (0, 30 min, 150 min) the authors identified an Immediate Early Splicing (IES) response involving a set of short intron retention events focused upon genes encoding translation factors. Quantitative phosphoproteomics carried out over a similar timecourse identified transient phosphorylation of the known splicing regulator hnRNPC2, which shows a complete switch to the phosphoform within 15 minutes, which was fully reversed by 90 and 240 min.

The authors show that the IES, and hnRNPC2 phosphorylation, involves signalling via not only RAF/MEK/ERK, but also PKC. The latter requirement explains T-cell restriction of the IES response. The net result is transient downregulation of translation, with translational reactivation coinciding with availability of newly transcribed IEG transcripts.

The data supporting the existence of the IES, the signalling via MEK/ERK and the T-cell specific PKC, and the resultant transient (and complete) phosphorylation of hnRNPC2 are strong. The data concerning the role of hnRNPC and in particular hnRNPC2 phosphorylation as a key mediator of signal-induced IES, and the functional consequences of the IES programme, are less clearcut.

Overall, in my opinion the manuscript provides sufficient evidence to support the broad outline of an interesting and important new splicing/translation-dependent mechanism in T-cell activation. Upon this basis, it provides significant new insights that will be of interest to the wider readership of EMBO J.

However, the data supporting some aspects of the model are weaker, and the manuscript could be improved either by additional experiments or by clarifying the conclusions that can be drawn from the data presented. In my opinion these are relatively minor details in the context of the broader new findings reported.

Specific points

Major

1. The manuscript shows that a set of intron retention (IR) events are focused upon genes encoding translation factors. However, there is no attempt to systematically predict the functional effects. What proportion introduce Premature Termination Codons? For those that are in-frame ("exitrons"), can any predictions be made about their effects on protein function?
2. Related to 1, little attention is paid to the absolute change in percent spliced in (PSI). The two model events used throughout the manuscript (RPL10 and eIF5A) are not switch-like (ie showing a switch from complete splicing to complete intron retention). Even in chromatin associated RNA the largest changes appear to be $PSI \sim 20-30\%$, but in total RNA they are 5-10 fold lower. Would individual changes of this magnitude be more likely to provide an effective IES if they were PTC/NMD linked or via coding (potentially dominant negative) protein isoforms?
Some of the events in Fig 1B, do appear to be switch-like (blue at 0 and 150 min, red at 30 min), and so could have major functional consequences. Are any of these events relevant to translation?
3. Related to 1 and 2. The data in Fig 6D-F address the role of the eIF5A IR event. As noted in the text, the levels of IR achieved by the antisense morpholino are much higher than were observed in any of the previous experiments. A demonstration of a significant effect on translation with levels of intron retention (titration of eIF5A IR MO) closer to those observed in earlier experiments (perhaps in combination with MOs targeting other IR events) would strengthen the argument that this event contributes to the IES.
4. The model presented for hnRNPC2 mediation of IES (as I understand it) is that hnRNPC2 activates splicing of IES introns, and that hnRNPC2 phosphorylation neutralizes this activity (loss of function) due to impaired RNA binding, leading to an increase in intron retention. However, much of the data presented is not consistent with this model:

i) on p7, referring to data in Supp Fig 2H the authors state:

"Thus, hnRNPC1 and hnRNPC2 isoforms appear to fulfill similar and redundant functions, and differences become only apparent upon hnRNPC2-specific phosphorylation, as this reduces binding to selected RNAs and controls alternative splicing."

Given that hnRNP C1 is much more abundant than hnRNPC2, it is very difficult to see how a loss of function by hnRNPC2 phosphorylation could be responsible for the splicing phenotype if C1 and C2 are functionally redundant.

ii) Why do the hnRNPC cells not show constitutively elevated levels of intron retention if hnRNPC2 (and C1) are necessary for efficient splicing?

iii) In Fig 2G, why is IR increased with the C2 MO? The model would predict decreased IR in unstimulated conditions due to the elevated levels of hnRNPC2, which is proposed to bind to the introns and promote splicing.

Given that this is a first manuscript on the IES, I do not think that it is reasonable to expect all details of the mechanism to be characterized in detail. Nevertheless, it would help if the manuscript presented a model that is more consistent with the data presented.

Minor

5. It would be better if all splicing data were quantitated simply as Percent Spliced In, rather than normalized to fold-change (e.g. Fig 1D, Supp Fig 1D,E).

6. Quantitation of hnRNPC2 (e.g. Figs 3G, 5C): why is hnRNPC2 phosphorylation quantitated relative to the much more abundant hnRNPC1 rather than to non-phosphorylated hnRNPC2? It would be better to quantitate as % phosphorylation of hnRNPC2 ($100 * (C2P / (C2P + C2))$).

Minor point: the y-axis on these graphs should indicate what is being quantitated, as well as the arbitrary units.

7. Strictly speaking the legend of Fig 3 is incorrect. Reduced RNA binding was observed with a phosphomimetic mutant, not upon phosphorylation. Related to point 4 above, it would be useful if data for hnRNPC1 could be included in this Figure if available.

We would like to thank the reviewers for their insightful and constructive comments and specific suggestions to increase the quality of our manuscript. Please find below a detailed point-by-point answer, reviewer comments are in black, answers in green.

Referee #1:

Immediate early splicing upon hnRNPC2 phosphorylation controls translation after T cell activation by Drozd et al

Drozd et al propose a model for immediate early splicing (IES), i.e., phosphorylation-dependent alternative splicing changes in response to T cell stimulation, in analogy to the well-studied transcriptional induction of immediate early genes (IEGs). Using RNA-seq on chromatin-associated RNA, the authors observe the transient regulation of alternative splicing, most prominently intron retention, in PMA-stimulated JURKAT cells and further validate exemplary events in stimulated primary T cells. Using phospho-proteomics, they observe transient phosphorylation of the longer hnRNPC isoform hnRNPC2. Indeed, increasing the relative hnRNPC2 abundance with a splice-switching morpholino leads to higher intron retention. In vitro binding assays with wt and phosphomimetic hnRNPC2 and inhibitor studies indicate that the phosphorylation results in decreased hnRNPC2 binding and is catalyzed by ERK kinase. The authors move on to show that the cell type-specific regulation by hnRNPC is achieved through the PKC β ; kinase, which is restricted to T cells and may set a priming phosphorylation on hnRNPC2. Consistently, IES can be recapitulated in HEK293 cells through the ectopic overexpression of PKC β ; . Finally, the authors show that IES IR in eIF5A mediates a global reduction in de novo protein synthesis.

The study of alternative splicing programs in immunology is both timely and relevant. The authors provide a detailed mechanistic dissection of the proposed IES response, offering convincing evidence from orthogonal assays to support their conclusions.

Comments:

1) Regarding the observation of increased intron retention upon 30 min of treatment, I wondered whether this could also be due to an increase in transcription. A burst in transcription could lead to an increase in yet unspliced nascent RNA. Have the authors checked this? Also related to this, could it be that splicing is rather slowed down at these introns and that they will eventually still be spliced?

To address the possibility that the observed increase in intron retention after 30 minutes of treatment could be due to increased transcription, we analyzed differential gene expression in chromatin-associated RNA-seq samples using GetMM and DESeq. Our differential expression analysis shows a slight, transient increase in total reads across genes exhibiting increased IR at 30 minutes. Interestingly, the few genes with increased splicing efficiency show a decrease in total reads (Fig. 1A, below). This mild increase in transcription (seen in *eIF5A* and *RPL10*) or a stronger burst (seen in *TRAF4*) could reflect transient transcriptional upregulation (Fig. 1B, below). However, this effect could also be the result of chromatin retention of not fully spliced, intron-containing transcripts. In fact, we also observe examples of genes with strong transcriptional induction but minimal changes in splicing (Fig. 1C, below), and overall, splicing changes do not strongly correlate with changes in gene expression (Fig. 1D, below). Therefore, we do not consider a transcriptional burst to be the primary driver of increased intron retention. We further explored this by performing RT-qPCR on total RNA samples from Jurkat T cells stimulated with PMA for 0, 30, and 150 minutes. Our results (presented in new Appendix Fig. 1B) show that the three target genes analyzed throughout our study were not upregulated after stimulation (Fig. 1E, below). These data suggest that the mild increase in chromatin-associated

transcripts is the result of intron-containing transcripts remaining associated with the chromatin, rather than a significant transcriptional increase. Some exceptions, like NFKB2 and TRAF4 (highlighted in Fig. 1D), might indeed reflect transcriptional upregulation.

Whether these intron-containing transcripts are eventually spliced, degraded, or exported to the cytoplasm remains an open question. However, we are confident that the example of EIF5A demonstrates that intron-retaining transcripts can reach the cytoplasm (Fig. EV5A, D) and function as protein-coding isoforms (Fig. EV5B). It remains to be determined how common this phenomenon is, as intron retention is generally thought to antagonize nuclear export. Given that the effect on our target genes in the RNA-Seq analysis is very modest and may be caused by retention of partially spliced RNA on the chromatin rather than transcriptional changes, and given that this effect is not observed in qPCR using total RNA, we would not show the analysis of the RNA-Seq data in the manuscript. We are of course happy to include these data if you consider it helpful for the overall understanding.

Fig. 1 (for reviewer information, only panel E is included in the manuscript (Appendix Fig. 1B): Gene expression changes associated with splicing changes. A. Heat map showing gene expression changes after 0, 30 and 150 minutes of PMA stimulation. Relative gene expression levels were calculated using

GetMM and were plotted as log₂-fold-change relative to time point 0 (red: higher expression at 30 or 150 minutes). The first column shows the DPSI value of all significant introns comparing 0 and 30 minutes (green: higher intron retention at 30 minutes). **B.** Increased gene expression is reversible. Box-Whisker plot showing GetMM derived relative gene expression levels for EIF5A, RPL10 and TRAF4 (on the left, black). For comparison, rMATS derived PSI values are shown on the right (pink). Statistical significance was analyzed by unpaired t-tests and is indicated by asterisks (*p<0.05; **p<0.01; ***p<0.001). Line represents mean, whiskers min to max, all individual data points are shown. **C.** Examples for genes with a transient increase in GE lacking strong splicing changes. Data are presented as in B. **D.** Linear regression fit of changes in GE (log₂-fold-change of 0 vs 30 minutes) on the y-axis and intron retention (DPSI of 0 vs 30 minutes). TRAF4 and NFKB2 are highlighted as two examples with high IR and GE changes. **E.** Constant expression of target genes upon PMA stimulation. Jurkat T cells were stimulated with PMA for the indicated time points, after which total RNA was extracted. Target gene expression was determined by RT-qPCR. mRNA expression levels are quantified relative to hHPRT (fold change normalized to 0 min, mean ± SD, n = 3).

2) Regarding the binding of hnRNPC2 to U tracts (Fig. 3G), I think that from the presented data it cannot be concluded that there is no difference in binding between wt and the phosphomimetic mutant since binding is already saturated in the lowest protein concentration used (2 micromolar). The assay should be repeated with lower concentrations of hnRNPH2.

Thank you for pointing this out. We have repeated the experiment with lower hnRNPC2 concentrations (0-2000 nM). There is no difference in binding between hnRNPC2 wt and the phosphomimetic mutant to a poly-U stretch. We have added this graph together with the corresponding quantification to Fig. 3G.

Fig. 2 (Fig. 3G in the updated manuscript): No difference in binding between hnRNPC_WT and a phosphomimetic S115D mutant to poly-U. EMSA was performed using 10 pmol of 45 nts poly-U RNA. Increasing amounts of either hnRNPC2_WT or S115D were complexed with radioactively labeled RNA. Left: representative native gel. Data are representative of at least three independent experiments. Right: quantification of EMSA (mean ± SD, n = 3).

3) In the authors' model, phosphorylation leads to reduced RNA binding of hnRNPH2. Does this loss of binding result in a loss of function or can it also have a gain-of-function effect? For example, how would hnRNPH2 phosphorylation affect the binding of the hnRNPH tetramer? In this context, I wondered whether the authors have tested overexpression of wt and phosphomimetic hnRNPH2 in cells. It will be interesting to show and discuss how splicing is affected in these experiments.

Thank you for this comment. We have cloned the hnRNPC2 variant with a single mutation, S115→D, to generate a phosphomimetic version of the protein. We then overexpressed this variant or hnRNPC2 WT as control in HEK293 cells, extracted chromatin-associated RNA, and performed radioactive,

splicing-sensitive RT-PCR. As shown in Figs. 3-B, (Figs. EV3F-G in the updated manuscript), phosphomimetic hnRNP2 increases intron retention in *eIF5a* and *RPL10*, indicating a critical role of hnRNP2 phosphorylation at Ser115 in IES. As these experiments were performed in the presence of the endogenous wt protein, the results suggest that the S115D mutant can act in a dominant negative manner. Whether this is in the context of the hnRNP2 tetramer or in another complex remains an open question. We think that this finding substantially strengthens our conclusions.

Fig. 3 (Figs. EV3F-G in the updated manuscript): phosphomimetic hnRNP2 increases intron retention in *RPL10* and *eIF5A*. hnRNP2 WT and its phosphomimetic version S115D were overexpressed in HEK293 cells. After 48h, cells were harvested, and chromatin-associated RNA was extracted. *RPL10* (left) and *eIF5A* (right) IR were analyzed by radioactive, splicing-sensitive RT-PCR and quantified (% IR; student's unpaired t-test; mean \pm SD, n = 3, *p<0.05, **p<0.01).

4) The requirement of a priming phosphorylation for IES is not very well supported. Have the authors experimentally approached this? Are there alternative explanations? Does the priming need to be on HNRNPH2 or could it be on an adapter protein mediating the ERK-mediated phosphorylation of Ser115?

Thank you for this question. The initial idea was that PKC θ installs a priming phosphorylation on hnRNP2, which then allows ERK-mediated phosphorylation at Ser115 in hnRNP2. According to previous work, hnRNP2 can be phosphorylated by PKC, which then results in changes in alternative splicing [1]. Data in this study suggested that the PKC target site within hnRNP2 could be S260, which is why we generated a S260D mutation. This mutated version, by acting as an analog of PKC θ -mediated phosphorylation, could allow Ser115 phosphorylation independent of PKC θ and as a consequence would result in PMA-induced IES in HEK293 cells. We observed a trend towards more IR in *eIF5a* in hnRNP2 S260D overexpressing PMA-stimulated HEK293 cells, which, however, was not significant (see below, Fig. 4). We therefore can't rule out an involvement of S260 phosphorylation as a priming site, but very likely other mechanisms play the dominant role. We would prefer not to show these data, as it is not fully conclusive and instead discuss different possible mechanisms in more detail in the manuscript.

Fig. 4: hnRNPC2 WT and a phosphomimetic mutant (S260D) were overexpressed in HEK293 cells. After 48h, cells were harvested, and chromatin-associated RNA was extracted. eIF5A IR was analyzed by radioactive, splicing-sensitive RT-PCR (left) and quantified (right) (% IR; student's unpaired t-test; mean \pm SD, n = 3, ns: non-significant).

Other:

I find the use of "PSI (percent spliced in)" in combination with intron retention confusing. Maybe better to use "isoform frequency" instead?

Thank you for the suggestion. We have changed 'PSI' to '% IR' throughout the manuscript.

Figure 2C, F and G: More meaningful labelling of the y-axis for the bar diagrams would be helpful. What was quantified in the bar chart? For instance, in panel G, is this phosphorylated hnRNPC2 over total hnRNPC2 or over total hnRNPC?

In Fig. 2F, the y-axis represents hnRNPC2 over hnRNPC1. In other Western Blots we quantitated the percentage of hnRNPC2 phosphorylation (% C2-P) by using the following equation: $100 * (C2P / (C2P + C2))$. This is now indicated in the "Method" section, as follows:

"Western blots were quantified using the ImageQuant TL software, shown is the percentage of phosphorylated hnRNPC2 over total hnRNPC2, using the following equation $100 * (C2P / (C2P + C2))$ ".

This sentence is complicated to read: "For all three investigated target introns IES occurred upon knock down of U2AF1 or hnRNPK, two other candidate trans-acting factors with transient phosphorylation upon T cell activation (U2AF1) or another polyC binding protein (hnRNPK) (Figs. 3A, B, Supp. Fig. 2D), showing specificity for hnRNPC."

Thank you, we have rephrased:

"For all three investigated target introns, knock down of U2AF1 or hnRNPK did not interfere with IES, indicating specificity for hnRNPC. U2AF1 and hnRNPK were chosen as specificity control as U2AF1 exhibits transient phosphorylation upon T cell activation and hnRNPK functions as another polyC binding protein (Figs. 3A, B, Fig. EV2E)."

The following paper would be good to cite:

Regulation of alternative pre-mRNA splicing by the ERK MAP-kinase pathway

<https://www.ncbi.nlm.nih.gov/pmc/articles/PMC149173/>

Thank you, the citation is included (#42 in the reference list).

Figure 1A: Colors of 0 and 150 min are indistinguishable

Thank you for pointing this out. Our idea was to indicate the similarities of these two conditions (0 min and 150 min) in terms of splicing pattern by using the same color. We would prefer to leave it that way but have used different shapes for the two timepoints in Fig. 1A.

Figure 1D, E: If the Y-axis shows "fold change (rel to 0 min)", why are the 0 min values not at 1? The authors should ensure that the quantification and normalization strategies are mentioned in the methods.

Thank you for this comment. We corrected and clarified the labelling of graphs in Fig. 1D and E.

Referee #2:

This manuscript reports a fascinating new phenomenon that coordinates splicing programs with T cell activation through a phosphorylation cascade has the potential for broad applicability to cell cycle and related phenomena. Specifically, a specific kinase in model T cells (Jurkat) targets the hnRNPC2, which then induces through alternative splicing a palette of protein isoform changes relevant to translation. This coordinated use of an alternative splicing pathway - called by the authors immediate early splicing (IES) by analogy to the classical term immediate early genes (IEG) - can yield transient intron retention and is an exciting new function for alternative splicing in general. In Jurkat cells, the effect of IES is to reduce translation. In some ways the most important finding in the paper is that the cellular events including global inhibition of translation can be phenocopied by ASO-mediated induction of intron retention in the eIF5A pre-mRNA, and this is basically a sanity check. The strength of the paper is the rigorous testing along the arc of the paper providing clear evidence of eIF5A and RPL10 intron retention at 30 min of PMA or anti-CD3 stimulation, direct effects on RNA binding and this AS event by hnRNPC2 phosphorylation, identification of the kinase, and reliance on hnRNPC2 expression and activity of the kinase in vivo. therefore, the paper generates a new conceptual framework for splicing regulation in the rapid tuning of translation in the context of an overall physiologically relevant pathway through step-by-step rigorous testing of the molecular underpinnings of their hypothesis.

We have made some suggestions mostly on the Figures/Data:

Figure 1A: It would be good to report the percent variance captured by the two principal components in the PCA plot, either in the legend or along the axis labels (e.g., "PC1 (60% variance)").

Thank you, we have included this information (see updated Fig. 1A). Importantly, PC1, which shows the strong difference of the 30-minute samples and the clustering of 0 and 150 minute samples, reflects 77% of the variance. Therefore, the majority of splicing variance is indeed reversible.

Figure 1B: The reader can infer that the individual columns in the heatmap correspond to individual replicates, but it might be better to directly label them. It could also be beneficial to include a sentence stating that each row in the heatmap corresponds to a different intron retention event.

Thank you, we have labelled as follows:

'Each individual column corresponds to individual replicates. Each row corresponds to a different intron retention event.'

Figure 1C: The amount of white space could be reduced in this panel by placing the two legends in the bottom right. The legend also says that the GO terms with "translation" are highlighted in blue. This sentence has an upside-down quotation mark before "translation" and an extra comma after "translation". It would also be more accurate to say "underlined in blue" rather than "highlighted in blue".

Thank you for these suggestions, they are all included.

Figure 1D-G: What are the middle bands in 1D,E,G?

The middle bands in Figures 1D-E follow the intensity of the intron retention bands and are likely degradation products of the IR product. The band is now labeled with an asterisk and an explanation is included in the figure legend.

A plot comparing the distribution of all intron sizes and intron sizes for the IR events of interest would be useful to convey the point about short introns made towards the bottom of page 4 (possibly include the plot in the supplement).

To better characterize features of differentially spliced introns, we have performed an analysis of intron length and 5' and 3' splice site strength of introns retained or spliced more efficiently after 30 minutes (significant relative to either 0 or 150 minutes) and compared this to all introns quantified by the rMATS software. This analysis is summarized in the new Fig. EV1D. Interestingly splice site strength of all groups is identical, but the length of introns spliced more efficiently after 30 minutes of stimulation is much longer than for the other two groups. Introns that are retained 30 minutes post stimulation are thus shorter than the ones spliced more efficiently but are not shorter on average than all introns quantified by rMATS. It is likely that rMATS preferentially quantifies small introns, resulting in a biased comparison group, but based on this analysis, target introns are on average not shorter than all rMATS-quantified introns. We discuss this in the manuscript.

Fig. 5 (Fig. EV1D in the updated manuscript): Introns spliced more efficiently after 30 minutes of stimulation are longer. Box-whisker plots comparing intron length (left), MaxEntScan 5'ss (middle) and 3'ss (right) for introns retained after 30 minutes (IR_30), introns more efficiently spliced after 30 minutes (spliced_30) and all introns quantified in rMATS (all introns). Statistical significance was analyzed by unpaired t-tests and is indicated by asterisks (ns: non-significant, ***p<0.001). Line represents median, whiskers min to max.

General comments for gels and blots: It would be good to label the sizes of the products / bands. This would make interpretation more straightforward.

Thank you for the suggestion. This is corrected in the revised version.

Other comments

Middle of first line on page 3: missing an apostrophe;

"increasing the genomes coding capacity" should be "increasing the genome's coding capacity"

Thank you, we have changed it.

On page 4 towards the bottom, should "as the vast majority of ~6.000 rMATS-quantified introns" be "as the vast majority of ~6,000 rMATS-quantified introns" (i.e., comma instead of period)? Or no punctuation?

Thank you, this is corrected.

Referee #3:

This interesting manuscript from Drozd, Heyd and colleagues describes a hitherto unknown gene regulatory mechanism to couple translational downregulation of the existing transcriptome with de novo transcription of Immediate Early Genes (IEG) upon T-cell activation. By carrying out chromatin associated RNA-Seq during T cell activation (0, 30 min, 150 min) the authors identified an Immediate Early Splicing (IES) response involving a set of short intron retention events focused upon genes encoding translation factors. Quantitative phosphoproteomics carried out over a similar timecourse identified transient phosphorylation of the known splicing regulator hnRNPC2, which shows a complete switch to the phosphoform within 15 minutes, which was fully reversed by 90 and 240 min.

The authors show that the IES, and hnRNPC2 phosphorylation, involves signalling via not only RAF/MEK/ERK, but also PKC ϵ . The latter requirement explains T-cell restriction of the IES response. The net result is transient downregulation of translation, with translational reactivation coinciding with availability of newly transcribed IEG transcripts.

The data supporting the existence of the IES, the signalling via MEK/ERK and the T-cell specific PKC ϵ , and the resultant transient (and complete) phosphorylation of hnRNPC2 are strong. The data concerning the role of hnRNPC and in particular hnRNPC2 phosphorylation as a key mediator of signal-induced IES, and the functional consequences of the IES programme, are less clearcut.

Overall, in my opinion the manuscript provides sufficient evidence to support the broad outline of an interesting and important new splicing/translation-dependent mechanism in T-cell activation. Upon this basis, it provides significant new insights that will be of interest to the wider readership of EMBO J.

However, the data supporting some aspects of the model are weaker, and the manuscript could be improved either by additional experiments or by clarifying the conclusions that can be drawn from the data presented. In my opinion these are relatively minor details in the context of the broader new findings reported.

Specific points

Major

1. The manuscript shows that a set of intron retention (IR) events are focused upon genes encoding translation factors. However, there is no attempt to systematically predict the functional effects. What proportion introduce Premature Termination Codons? For those that are in-frame ("exitrons"), can any predictions be made about their effects on protein function?

Thank you for your insightful suggestion. It is indeed crucial to consider how often intron retention (IR) events might alter the encoded proteins. Our analysis shows that 169 of the 173 IR events occur in protein-coding genes, with 140 of these situated within the open reading frame (ORF) (see

Fig. 6 and graph in new Fig. EV1F). The 29 IR events located outside the ORF may influence the translation efficiency of the primary isoform through altering their UTRs. Of the 140 introns within the ORF, 136 introduce stop codons, either through frameshifts (93 cases) or by directly introducing a stop codon (43 cases). In 10 of these cases, the stop codon occurs in the last intron of the ORF, potentially producing an alternative C-terminus, as previously described in [2]. In the remaining 126 cases, the introns would encode premature termination codons (PTCs). This leaves 4 cases of traditional "exitrons" that maintain in-frame sequences, including the functionally significant event in the translation initiation factor EIF5A. The other three genes with exitrons are IRF3 (Interferon Regulatory Factor 3), MFSD10 (Major Facilitator Superfamily Domain-containing Protein 10), and TNFRSF1A (Tumor Necrosis Factor Receptor Superfamily Member 1A). Notably, IRF3, a key regulator of interferon alpha/beta transcription and other interferon-induced genes, has the alternative intron close to its DNA-binding domain, which might affect its transcriptional activity. The IR isoform in TNFRSF1A could potentially alter its TNF-alpha binding domain. In summary, while protein-coding IR events are relatively rare, as expected in humans, they may significantly impact T cell physiology by altering the function of critical proteins. However, the majority of IR events are more likely to reduce the expression of the affected genes by NMD and may also impact on protein function by producing truncated proteins in the first round of translation. These data are included and discussed in the revised version of the manuscript.

Fig. 6 (Fig. EV1F in the updated manuscript): Protein coding potential of IR isoforms. Last introns encoding a premature termination codon (PTC) are highlighted as having coding potential.

2. Related to 1, little attention is paid to the absolute change in percent spliced in (PSI). The two model events used throughout the manuscript (RPL10 and eIF5A) are not switch-like (ie showing a switch from complete splicing to complete intron retention). Even in chromatin associated RNA the largest changes appear to be dPSI ~ 20-30%, but in total RNA they are 5-10 fold lower. Would individual changes of this magnitude be more likely to provide an effective IES if they were PTC/NMD linked or via coding (potentially dominant negative) protein isoforms? Some of the events in Fig 1B, do appear to be switch-like (blue at 0 and 150

min, red at 30 min), and so could have major functional consequences. Are any of these events relevant to translation?

This is an interesting question, and we agree that large splicing changes are more likely to have functional significance. However, we also believe that the cumulative effect of multiple smaller changes could be biologically meaningful. It is also true that stable protein isoforms produced during the narrow window of IR isoform production are more likely to play a role beyond that period. Regarding the magnitude of splicing changes, we find that most events do not exhibit a switch-like behaviour. The majority show moderate PSI changes, typically between 15-25%, with fewer cases displaying more substantial shifts (see Fig. 7, below, also included in the manuscript). In our analysis, we identified 173 events with a PSI change greater than 15%, though it's likely that there are more events with smaller shifts. The strongest changes in PSI are not specifically associated with translation. Some of the most affected genes include *CHPF*, *HLA-B*, *RELB*, *VLDLR*, *WAC*, *P2RX4*, *HLA-A*, *DCAF13*, and *TPM4*. Therefore, we propose that the impact of hnRNPC2-mediated IR on global translation dynamics is likely due to the cumulative reduction in several translation-associated transcripts (via nuclear degradation or NMD) and/or may be primarily mediated through (potentially dominant negative) *EIF5A_IR*. Additionally, the most strongly affected genes could influence other processes related to T cell activation.

Regarding the four protein-coding events, each shows an approximately 25% change in PSI (in chromatin associated RNA). This level of change could generate sufficient amounts of a protein variant that acts dominant-negative (as demonstrated for *EIF5A_IR*) or displays different functionality than the canonical version.

Fig. 7: Frequency distribution of PSI changes (Fig. EV1E in the updated manuscript). To summarize the magnitude of splicing changes we have calculated the maximal PSI change (0 vs 30 or 30 vs 150 minutes) for all 173 significant introns. The histogram summarizes the frequency of PSI changes in Bin's differing by 0.05 PSI change (the number on the x-axis represents the center of each Bin).

3. Related to 1 and 2. The data in Fig 6D-F address the role of the *eIF5A* IR event. As noted in the text, the levels of IR achieved by the antisense morpholino are much higher than were observed in any of the previous experiments. A demonstration of a significant effect on translation with levels of intron retention (titration of *eIF5A* IR MO) closer to those observed in earlier experiments (perhaps in combination with MOs targeting other IR events) would strengthen the argument that this event contributes to the IES.

Thank you for this suggestion. We transfected the increasing concentrations of EIF5A_IR_MO and CTRL_MO in Jurkat T cells. First, we tested the effects of these MOs on splicing of *eIF5A*. Increasing EIF5A_IR_MO concentrations led to increasing levels of IR in *eIF5A* (Fig. 8A; Fig. EV5E in the revised version of the manuscript). Subsequently, we performed SuNSET assays using the same conditions to assess the effect on *de novo* translation. We observed a progressive reduction in global translation with increasing concentrations of EIF5A_IR_MO, while increasing concentrations of CTRL_MO did not result in a significant change (Fig. 8B, Fig. EV5F in the revised version of the manuscript).

Fig. 8 A: Titration experiment with increasing concentrations of EIF5A_IR_MO. (Fig. EV5E in the revised manuscript) Jurkat cells were electroporated with increasing concentrations of EIF5A_IR_MO (from 0.3 nmol to 4.5 nmol). Left: representative gel from radioactive, splicing-sensitive PCR; Right: Quantification of analysis from samples as shown on the left (%IR, student's unpaired t-test; mean \pm SD, n = 3, ns: not significant, *p<0.05; **p<0.01). **B. Titration experiment with increasing concentrations of EIF5A_IR_MO** (Fig. EV5F in the revised manuscript). Jurkat cells were electroporated with increasing concentrations of EIF5_IR_MO as in E. After 48 hours, cells were treated with 10ug/ml of puromycin 10 minutes before harvesting. Total protein was extracted. Left: A representative blot of the WB-SuNSET experiment. hnRNPL serves as loading control. Right: Quantification of analysis from samples as shown on the left; fold change normalized to 0.3 nmol; mean \pm SD, n = 3, *p<0.05).

Furthermore, we compared the magnitude of *eIF5A* intron retention upon 30 minutes of PMA stimulation with the effect upon transfection of 3 nmol EIF5A_IR_MO. Although these conditions cannot be directly compared as morpholino treatment continuously induced the IR isoforms while PMA stimulation induced this isoform only in a short time window, we find that the change in intron retention is similar (Fig. 9A). In the same condition we observed a similar impact on *de novo* translation (Fig. 9B). These data suggest that levels of EIF5A_IR produced during PMA stimulation could alone be sufficient to reduce translation efficiency.

Fig. 9 (provided for reviewer information): The graph compares the change of eIF5a IR upon A) PMA stimulation (30 min) and transfection of 3 nmol EIF5A_IR_MO and B) the impact on *de novo* protein synthesis. Both are in a similar order of magnitude, suggesting that eIF5A IR observed upon PMA stimulation can alone cause a decrease in global translation.

4. The model presented for hnRNPC2 mediation of IES (as I understand it) is that hnRNPC2 activates splicing of IES introns, and that hnRNPC2 phosphorylation neutralizes this activity (loss of function) due to impaired RNA binding, leading to an increase in intron retention. However, much of the data presented is not consistent with this model:

This is exactly the model we suggest. We think that all data are consistent with this model, see comments below.

i) on p7, referring to data in Supp Fig 2H the authors state: "Thus, hnRNPC1 and hnRNPC2 isoforms appear to fulfill similar and redundant functions, and differences become only apparent upon hnRNPC2-specific phosphorylation, as this reduces binding to selected RNAs and controls alternative splicing."

Given that hnRNP C1 is much more abundant than hnRNPC2, it is very difficult to see how a loss of function by hnRNPC2 phosphorylation could be responsible for the splicing phenotype if C1 and C2 are functionally redundant.

This is a good point and in part relates to point 7 (also see below). We think that phosphorylated hnRNPC2 may act in a dominant negative manner in a hnRNPC1:C2 complex, in a stoichiometry where one hnRNPC2 can inactivate several hnRNPC1 molecules. This is a hypothesis, reasonable but it remains to be shown experimentally.

ii) Why do the Δ hnRNPC cells not show constitutively elevated levels of intron retention if hnRNPC2 (and C1) are necessary for efficient splicing?

We assume that Δ hnRNPC refers to the cell line lacking the hnRNPC2 isoform (Δ hnRNPC2). This cell line does contain hnRNPC1, as only the splice site specific for hnRNPC2 is destroyed. We think that this does not alter overall hnRNPC levels but simply results in a shift to the C1 isoform. As we think that this is also active in promoting splicing of our target introns, we do not expect a change in basal IR.

iii) In Fig 2G, why is IR increased with the C2 MO? The model would predict decreased IR in unstimulated conditions due to the elevated levels of hnRNPC2, which is proposed to bind to the introns and promote splicing.

We think that the C2 MO does not alter total hnRNPC levels but shifts the C1:C2 balance to more C2. As we think that both C1 and C2 are active in promoting splicing, the altered C1:C2 ratio in the basal condition does not change splicing efficiency. But in the activated state, more C2 is phosphorylated, does not bind anymore and therefore there is an increase in intron retention.

We apologize if the model was not clear and have added some more explanation to the manuscript.

Given that this is a first manuscript on the IES, I do not think that it is reasonable to expect all details of the mechanism to be characterized in detail. Nevertheless, it would help if the manuscript presented a model that is more consistent with the data presented.

Minor

5. It would be better if all splicing data were quantitated simply as Percent Spliced In, rather than normalized to fold-change (e.g. Fig 1D, Supp Fig 1D,E).

Thank you, we have changed the representation of splicing changes. Splicing data show the percentage of retained introns (% IR) now.

6. Quantitation of hnRNPC2 (e.g. Figs 3G, 5C): why is hnRNPC2 phosphorylation quantitated relative to the much more abundant hnRNPC1 rather than to non-phosphorylated hnRNPC2? It would be better to quantitate as % phosphorylation of hnRNPC2 ($100 * (C2P / (C2P + C2))$). Minor point: the y-axis on these graphs should indicate what is being quantitated, as well as the arbitrary units.

Thank you, we have changed the quantification of hnRNPC2 phosphorylation by using the suggested strategy and indicate this on the y-axis.

7. Strictly speaking the legend of Fig 3 is incorrect. Reduced RNA binding was observed with a phosphomimetic mutant, not upon phosphorylation. Related to point 4 above, it would be useful if data for hnRNPC1 could be included in this Figure if available.

Thank you for this comment. We have changed the figure legend. We agree that characterizing the role of hnRNPC1 would be very interesting, for example the question whether phosphorylated hnRNPC2 can act in a dominant negative manner by sequestering hnRNPC1. The effect of hnRNPC2 S115D expression on intron retention in the presence of endogenous hnRNPC1/2 suggests a dominant negative effect (see reviewer 1, point 3). However, as we have focused our manuscript on hnRNPC2 we have not purified hnRNPC1, as addressing this question in detail would/will likely provide enough data for an entire new manuscript.

References:

[1] Martino F, Varadarajan NM, Perestrelo AR, Hejret V, Durikova H, Vukic D, Horvath V, Cavalieri F, Caruso F, Albihlal WS, Gerber AP, O'Connell MA, Vanacova S, Pagliari S, Forte G (2022) The mechanical regulation of RNA binding protein hnRNPC in the failing heart. *Sci Transl Med.* 14(672):eabo5715.

[2] Preussner, M., Gao, Q., Morrison, E., Herdt, O., Finkernagel, F., Schumann, M., Krause, E., Freund, C., Chen, W., and Heyd, F (2020). Splicing-accessible coding 3'UTRs control protein stability and interaction networks. *Genome Biol* 21, 186.

Dear Dr. Heyd,

We have now received re-review reports from three referees, which I have included below. As you will see, you have addressed their concerns satisfactorily; however, I would like you to carry out and discuss the brief analysis suggested by referee #2. Before I can finally accept the manuscript, there are some remaining editorial points which need to be addressed. In this regard would you please:

- acknowledge the following funding in our online submission system: LMU - Munich's Institutional Strategy LMUexcellent within the framework of the German Excellence Initiative,
- remove DOIs from all published articles that are listed in the reference section,
- ensure that all datasets are publically available upon acceptance and adapt the data availability section accordingly,
- include a completed author checklist (a template is available on our website),
- rename Table EV1 and Table EV2 as Dataset EV1 and Dataset EV2 (as they each have multiple columns), updating their source file names, titles in our online submission system and manuscript callouts,
- correct the nomenclature of appendix figures to "Appendix Figure S1", etc. (ie adding an 'S') add manuscript callouts and include a table of contents with page numbers on the appendix title page,
- upload Source Data as one folder per figure - each folder should have separate subfolders for each panel, please include source data for figure panel 6F,
- reduce the size of the synopsis image to exactly 550 pixels wide and between 300-600 pixels high,
- remove the R&T table from the manuscript and upload it separately,
- rename the 'Materials and Methods' section the 'Methods' section,
- ensure that, where $n=2$, data points are plotted without statistical tests; the range in values may be indicated with a bar if this is explicitly stated in the figure legend, and
- state exact p values in the legends of figures 1d-g; 2a, c, e-i; 3a-d; 4e-f; 5a-g; 6a-b, d-f; EV 2c-f; EV 3f-k; EV 5d-f.

I am looking forward to receiving your revised manuscript.

EMBO Press is an editorially independent publishing platform for the development of EMBO scientific publications.

Best wishes,

William

William Teale, PhD
Editor
The EMBO Journal
w.teale@embojournal.org

We realize that it is difficult to revise to a specific deadline. In the interest of protecting the conceptual advance provided by the work, we recommend a revision within 3 months (2nd Apr 2025). Please discuss the revision progress ahead of this time with the editor if you require more time to complete the revisions. Use the link below to submit your revision:

Referee #1:

The authors have addressed all my concerns. Thank you for the clear and detailed point-by-point response. Congratulations on this exciting work!

Referee #2:

The manuscript has been improved by the authors, responding to more extensive comments from reviewers 1 and 3. We (reviewer 2) had focused on data, analysis and presentation issues. The authors responded nicely to our suggestions, and we feel these aspects have been improved. One nagging issue is raised by the authors' response to our question about intron lengths, to which the authors responded by creating a new figure in Extended figure EV1D. This brought to light that, either because of the tool the authors are using to analyze their data (rMATS) or because they have a somewhat low number of mappable read counts in their dataset, they are for some reason only analyzing the behavior of small introns to their treatments. the median size of "all" introns in their analyzed set of introns is 300nt long, when the median annotated intron size is much longer (~2,000nt). In our lab, we also use rMATS and, in a recent analysis, the analyzed set of introns were longer than the median of annotated introns in that species. Thus, we would like to ask the authors if they could

- use different tools (e.g. Whippet, MAJIQ) to analyze their datasets. If a larger proportion of introns would be available for analysis, this would be of interest to report in parallel (perhaps in the supplement). We would not expect the existing results concerning hnRNP C's affects on small introns to be affected by the identification of new introns that may potentially be longer. These results would enable the authors to modify their text to make more expansive comments and generalizations.
- Whether or not that avenue is productive, we feel the authors must state the median intron length for all annotated introns, explicitly define the length range they are analyzing, and make their interpretations in the context of their knowledge that they may not be comprehensively analyzing alternative splicing but rather a subset of possible events.

Referee #3:

The authors have engaged thoughtfully with all of the issues I raised at the first review. As I stated in my initial review: "Overall, in my opinion the manuscript provides sufficient evidence to support the broad outline of an interesting and important new splicing/translation-dependent mechanism in T-cell activation. Upon this basis, it provides significant new insights that will be of interest to the wider readership of EMBO J."

The manuscript has been further improved during revisions and I am happy to recommend publication.

Answers to reviewer comments on EMBOJ-2024-118552R

Immediate early splicing mediated by hnRNPC2 phosphorylation controls translation after T cell activation

Referee #1:

The authors have addressed all my concerns. Thank you for the clear and detailed point-by-point response. Congratulations on this exciting work!

Thank you very much for your very helpful feedback and comments!

Referee #2:

The manuscript has been improved by the authors, responding to more extensive comments from reviewers 1 and 3. We (reviewer 2) had focused on data, analysis and presentation issues. The authors responded nicely to our suggestions, and we feel these aspects have been improved. One nagging issue is raised by the authors' response to our question about intron lengths, to which the authors responded by creating a new figure in Extended figure EV1D. This brought to light that, either because of the tool the authors are using to analyze their data (rMATS) or because they have a somewhat low number of mappable read counts in their dataset, they are for some reason only analyzing the behavior of small introns to their treatments. The median size of "all" introns in their analyzed set of introns is 300nt long, when the median annotated intron size is much longer (~2,000nt). In our lab, we also use rMATS and, in a recent analysis, the analyzed set of introns were longer than the median of annotated introns in that species. Thus, we would like to ask the authors if they could use different tools (e.g. Whippet, MAJIQ) to analyze their datasets. If a larger proportion of introns would be available for analysis, this would be of interest to report in parallel (perhaps in the supplement). We would not expect the existing results concerning hnRNP C's effects on small introns to be affected by the identification of new introns that may potentially be longer. These results would enable the authors to modify their text to make more expansive comments and generalizations. Whether or not that avenue is productive, we feel the authors must state the median intron length for all annotated introns, explicitly define the length range they are analyzing, and make their interpretations in the context of their knowledge that they may not be comprehensively analyzing alternative splicing but rather a subset of possible events.

We appreciate the reviewers' thoughtful feedback regarding the analysis of intron lengths and the tools used in our study. We find that the median intron length of 312nt in our rMARS analysis is in a similar range as previously found in an analysis of human RNA-seq data comparing different analysis tools (<https://pubmed.ncbi.nlm.nih.gov/36369064/>, David et al., Genome Biology, 2022). Similarly, in our own prior analyses of human RNA-seq samples using rMATS (e.g. <https://pubmed.ncbi.nlm.nih.gov/36867703/>), we observed a median intron length in a similar range, excluding a sample-specific effect.

Following the reviewer's suggestions, we performed an additional analysis using Whippet as an alternative tool. Consistent with our original findings, we observed a substantial increase in intron retention following 30 minutes of PMA stimulation (Fig. 1A), reinforcing the robustness of this result across different analysis pipelines. However, the median length of introns analyzed using Whippet was

even shorter (267 nt) compared to rMATS (Fig. 1B). Both tools also show that retained introns tend to be slightly shorter than introns that are more efficiently spliced 30 min after stimulation although this trend is not significant in Whippet.

While the Whippet analysis aligns well with our findings, we suggest to have only the rMATS analysis in the manuscript. As reviewer #2, we would have expected a longer mean intron length for humans. As this is not the case in different datasets using different analysis pipelines, and this was independently observed by others (<https://pubmed.ncbi.nlm.nih.gov/36369064/>), we decided to simplify the respective paragraph and avoid a statement that directly connects short introns with intron retention after stimulation:

Interestingly, introns that are more efficiently spliced 30 min after stimulation are longer than retained introns pointing to intron size as one factor that plays a role in determining splicing outcome. Splice site strength of affected introns was not different between the differentially regulated introns (Fig. EV1D), suggesting that other features are involved in controlling retention/splicing early after activation.

We believe these revisions address the very valid comment and adequately reflect the data.

Fig. 1 Analyzing alternative splicing with Whippet.

A. For this analysis duplicate RNA-seq samples from non-stimulated and 30-min PMA stimulated Jurkat cells were aligned and quantified using whippet-quant followed by a differential splicing analysis using whippet-delta. Events were considered significant when a PSI change $> \pm 0.15$ with a probability > 0.95 was observed. This analysis reveals substantially more IR after 30 minutes as the strongest change in alternative splicing.

B. Summary of intron length. The length of all introns quantified using whippet (whippet introns, total of 8070 introns) was compared to changed introns with increased (IR_30, 314 introns) or decreased (spliced_30, 45 introns) IR after stimulation. Statistical significance was determined by unpaired t-test and reveals a slight reduction in intron length in introns retained after stimulation ($*p < 0.05$).

Referee #3:

The authors have engaged thoughtfully with all of the issues I raised at the first review. As I stated in my initial review: "Overall, in my opinion the manuscript provides sufficient evidence to support the broad outline of an interesting and important new splicing/translation-dependent mechanism in T-cell activation. Upon this basis, it provides significant new insights that will be of interest to the wider readership of EMBO J." The manuscript has been further improved during revisions and I am happy to recommend publication.

We thank the reviewer for their comments and suggestions to improve the quality and impact of our manuscript.

Dear Florian,

I am pleased to inform you that your manuscript has been accepted for publication in the EMBO Journal.

Congratulations!

Yours sincerely,

William

William Teale, PhD
Editor
The EMBO Journal
w.teale@embojournal.org
